# Computer Agent Arena: Toward Human-Centric Evaluation and Analysis of Computer-Use Agents

**Bowen Wang**[*h] **Xinyuan Wang**[*h] **Jiaqi Deng**[*h] **Tianbao Xie**[h] **Ryan Li**[s] **Yanzhe Zhang**[s]

**Junli Wang**[h] **Dunjie Lu**[h] **Zicheng Gong**[h] **Gavin Li**[s] **Toh Jing Hua**[h] **Ion Stoica**[b] **Wei-Lin Chiang**[b]

**Diyi Yang**[s] **Yu Su**[o] **Yi Zhang**[a] **Zhiguo Wang**[a] **Victor Zhong**[w] **Tao Yu**[†h]

[h] The University of Hong Kong    [s] Stanford University    [o]The Ohio State University
[b]University of California, Berkeley    [a]AWS AI    [w]University of Waterloo

## Abstract

As Computer-Use Agents (CUAs) proliferate and grow increasingly capable, evaluation has become more challenging: static, manually curated benchmarks are narrow in domain, contamination-prone, and environment-heavy, and they diverge substantially from user-driven, real-world evaluation. We present COMPUTER AGENT ARENA, an open-source platform for head-to-head CUA evaluation and a dynamic methodology that converts human preferences into structured feedback in realistic environments. The system (i) simulates real-world computer use via cloud-hosted, diverse, and dynamic environment initializations and customizations; (ii) ensures authentic, fair comparison by faithfully reproducing open-source CUAs and executing anonymously in matched, controlled environments; and (iii) extends evaluation beyond pairwise preference and correctness to capability- and behavior-oriented signals. Across 2,201 high-quality votes over 12 agents—spanning multi-app interactions, ambiguous instructions, and open-ended queries—we observe striking ranking reversals relative to static benchmarks. Further analysis shows that overall correctness mainly drives human preference; beyond that, agent-human interaction and self-correction boost user preference, even when overall task completion is comparable. Our error analysis reveals agent behavior errors, such as long-horizon memory and fine-grained action failures that static benchmarks fail to evaluate. We also contrast pure GUI agents with universal digital agents capable of tool use and coding, and discuss the trade-offs of these different design philosophies. We open source the full platform, collected dataset, and code of COMPUTER AGENT ARENA to support future research on the evaluation and development of CUA.

🏠   **Website:** https://arena.xlang.ai
🔾   **Code:**   https://github.com/xlang-ai/computer-agent-arena
🗄   **Dataset:** https://huggingface.co/xlang-ai/computer-agent-arena

## 1 Introduction

Recent advancements in Large Language Models (LLMs) and Vision-Language Models (VLMs) have demonstrated significant potential for building Computer-Use Agents (CUAs) (Anthropic, 2025c; OpenAI, 2025d; Qin et al., 2025; Guo et al., 2025; Team et al., 2025). Such agents can perform various computer-use tasks, from web browsing to professional applications (Xu et al., 2024b; Xie et al., 2024a; Li et al., 2025a; Drouin et al., 2024; Wang et al., 2025). These systems are increasingly positioned for real-world deployment, making human-centric evaluation grounded in user preferences, safety, and reliability in open-ended settings a prerequisite rather than an afterthought.

Currently, CUA evaluations predominantly rely on static online and offline benchmarks (Xie et al., 2024a; Zhou et al., 2024; Xue et al., 2025; He et al., 2024; Xie et al., 2025b) containing human-

---

*Equal contribution. †Corresponding author.

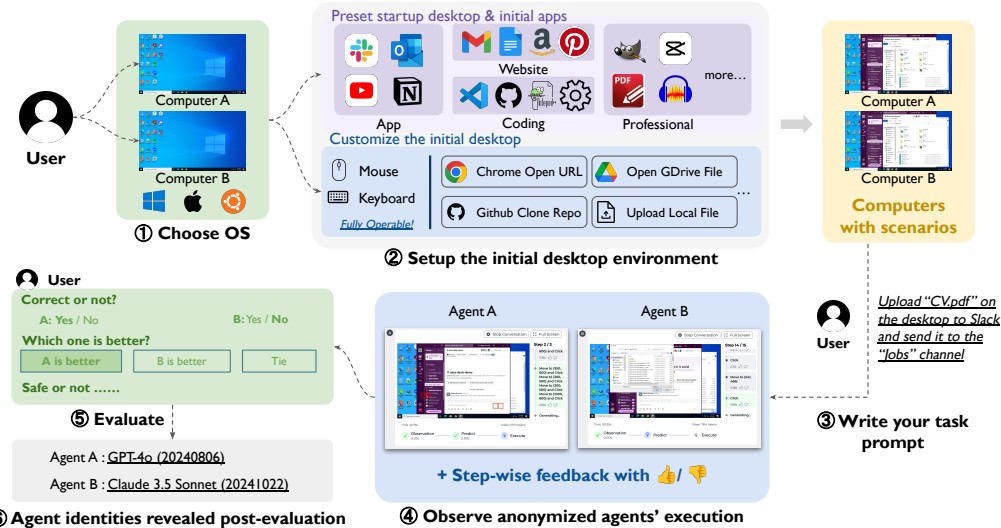

Figure 1: COMPUTER AGENT ARENA evaluation workflow: Users select an operating system (Windows/Ubuntu) and initialize the environment via preset or custom scripts. Two anonymized CUAs execute tasks simultaneously in parallel virtual machines, with their recorded visual trajectories presented for user evaluation. User preferences and correctness judgments are collected to generate global agent rankings.

written computer tasks with manually designed reward functions. However, these benchmarks are increasingly unable to accurately assess CUA capabilities and systematically neglect human- and real-world-centric evaluation. The static and limited task domain and evaluation environment leave them vulnerable to contamination/overfitting and unable to capture real-world dynamism and open-ended objectives. They also ignore personalization (different users value different outcomes and interaction styles), underestimate safety/privacy risks, and lack robustness to environment drift (software updates, network variability, unseen apps). Finally, their design often trades off authenticity for reproducibility, offering little guidance on fair head-to-head comparisons or scalable human-centric feedback collection.

To address these limitations, we introduce COMPUTER AGENT ARENA, a novel evaluation frame-work allowing evaluating CUAs on authentic computer tasks from real users and assessing agent performance through direct human feedback. To ensure diversity and authenticity, we deploy a cloud-hosted VM infrastructure with hundreds of prebuilt setups (e.g., preopened apps/web pages and files) and support user-defined initializations and customizations to mimic realistic computer use. To ensure fair comparison, two anonymous CUAs execute the user-proposed instruction in parallel within identical environments, rendering a side-by-side execution trajectory for pairwise evaluation. The accumulated preferences converge to a stable leaderboard through Bradley–Terry ranking model (Bradley & Terry, 1952), following the Chatbot Arena (Chiang et al., 2024). In parallel, optional step-wise evaluations of grounding errors, privacy violations, and self-correction behaviors are also collected to comprehensively evaluate CUAs capabilities.

We collected 2,201 filtered, high-quality votes from 1,058 users across 12 CUA models. The resulting leaderboard diverges sharply from static benchmarks—most notably, several top performers on OSWORLD (Xie et al., 2024b) are inverted in our setting. Through analysis, we find this divergence stems from (i) a broader, more heterogeneous task distribution and environment and (ii) human-centric evaluation criteria that emphasize process quality in realistic environments. We first did rigorous error study identifies systematic failure modes that static suites underexpose: (1) tool-integrated agents, despite excelling on scripted benchmarks, often underperform in real-user tasks due to tool-selection and tool-use errors; (2) long-horizon memory lapses and plan drift; (3) insufficient information seeking and underuse of clarification; and (4) fine-grained grounding/action-precision

errors. Beyond agent performance, preference analysis from a human-centric perspective shows that users evaluate the execution process, not just outcomes: agents earn preference through thoughtful planning, meaningful partial progress, error recovery, responsiveness, and judicious CALL_USER queries, even when full completion is not achieved. These effects are especially salient in open-ended or subjective tasks, where correctness is inherently ambiguous and closely tied to process quality. Together, the findings surface alignment signals beyond outcome correctness and clarify why agents can both fail more often and still (or fail to) earn human preference.

In summary, COMPUTER AGENT ARENA establishes a human-centric methodology for evaluating computer-use agents by converting real-world tasks and real-user preferences into structured signals and stable rankings. Our *rigorous error study* diagnoses current agent shortcomings (e.g., tool-selection/use errors, long-horizon memory failures and plan drift, and fine-grained action errors), while our *human-centric preference analysis* identifies what users actually care about in deployment—process quality, responsiveness, judicious clarification queries, recovery from errors, and privacy awareness. Beyond revealing ranking reversals relative to scripted benchmarks, we open-source the platform, reference implementations, and a large, multimodal, human-labeled preference dataset to facilitate CUA research. Rather than replacing benchmark-driven evaluation, our work provides a complementary human-centric lens that should not be overlooked in agent design and deployment.

## 2 COMPUTER AGENT ARENA SYSTEM DESIGN

We design the COMPUTER AGENT ARENA evaluation system as shown in Figure 1. In this section, we detail the platform infrastructure, agent execution interface, and the ranking system that transforms pairwise evaluations into a global leaderboard.

### 2.1 PLATFORM IMPLEMENTATION

Scalable online CUA evaluation requires infrastructure that (i) elastically serves many concurrent sessions, (ii) presents diverse, realistic software and web contexts, and (iii) guarantees *fair* comparison by running agents in matched environments. COMPUTER AGENT ARENA meets these requirements with a cloud-based stack that exposes fully interactive desktops to general users via a web interface.

**Scalable.** We extend OSWorld (Xie et al., 2024b) by packaging a standardized AMI and deploying it on AWS EC2 behind a dedicated backend service. The service provides on-demand provisioning of preconfigured virtual machines (VMs), parallel allocation for crowd evaluations, and low-latency startup through a managed pool. Each session streams a native desktop via a VNC window in the browser, enabling real-time interaction without client-side installation (Figure 1, Step (1)).

**Diverse and open-domain.** To replicate authentic computer use, we curated 600+ distinct initializations spanning both software and the web. Concretely, we sample popular sites from SimilarWeb and expand coverage through popular subdomains; we install mainstream applications from Microsoft Store and Snapcraft; and we preload 100+ heterogeneous files (e.g., .docx, .py) to instantiate realistic workflows (Figure 1, Step (2)). To reduce overfitting to fixed contexts, file-system contents are periodically refreshed. *To further support customization*, we provide quick-start tools that let users configure environments with minimal friction—e.g., uploading files, pre-opening specific websites, cloning GitHub repositories, and applying initialization recipes (packages, datasets, browser profiles) in one click. These initializations and user-defined customizations keep scenarios flexible and aligned with real user needs.

**Fair.** For head-to-head comparisons, two anonymized CUAs are instantiated in *identical* environments: same AMI, software versions, initialization recipe, and seeded configuration. The backend records an environment fingerprint (AMI ID, package hashes, and initialization spec) with each trial to support reproducibility checks. Agents execute *in parallel* to avoid temporal drift; their full trajectories are captured via built-in OBS services and rendered as synchronized replays on the evaluation interface. Users then submit pairwise preferences and structured labels (e.g., correctness, safety, efficiency), producing comparable feedback across matched conditions (Figure 1, Step 4).

**Observable and user-friendly.** All interactions are logged at step level (screens, actions, timestamps) and compiled into visual summaries for rater inspection. The browser-based workflow lowers the

barrier for non-technical participants while preserving a native desktop experience, enabling scalable, diverse, and fair human-centric evaluation at crowd scale.

## 2.2 AGENT IMPLEMENTATION

CUAs interact with computers through a unified action space and API service to ensure cross-model compatibility (Xu et al., 2024c; Bai et al., 2025; Guo et al., 2025; Xu et al., 2024b; Zhang et al., 2025; Wang et al., 2025). At each timestep, the agent receives a 1280×720 desktop screenshot and outputs a structured function call representing actions such as mouse movement, clicks, keyboard typing, scrolling, or special signals (`DONE`, `FAIL`, `CALL_USER`). Each session begins with a natural language instruction and proceeds as a trajectory of state–action pairs until termination, with full action specifications in Appendix B.3. For model integration, we adopt official frameworks when available (e.g., OpenAI Operator (OpenAI, 2025d), Claude 3.7 Sonnet (Anthropic, 2025c)) and otherwise use a standardized baseline agent that handles screenshot ingestion, prompting, function-call formatting, and environment interaction. This unified pipeline isolates model behavior and ensures fair comparison under identical constraints, including fixed step limits, response windows, and access to prior CoT outputs. To ensure fairness and reproducibility, all open-source CUAs are instantiated verbatim from public repository, released checkpoints, default system prompts and tools, inference parameters (e.g., temperature, max-tokens), and tool schemas—so that differences in outcomes reflect model behavior rather than integration variance. This unified pipeline isolates the model, equalizes interfaces and resources, and enables apples-to-apples comparison under identical environments. The details of implemented CUA models are listed in Section 3.1.

## 2.3 AGENT RANKINGS

Each evaluation in COMPUTER AGENT ARENA results in a pairwise preference vote: two anonymized CUAs execute the same task in identically initialized cloud computers, and the user selects the better-performing agent or declares a tie. Following prior work on human preference evaluation (Chiang et al., 2024; Chi et al., 2025; Li et al., 2025b; Guertler et al., 2025), we aggregate these votes into a global leaderboard using an Elo ranking system derived from the Bradley–Terry model (Bradley & Terry, 1952).

Let $x_i = (m_i^{\mathsf{L}}, m_i^{\mathsf{R}}) \in [M]^2$ denote the agent pair in comparison $i$, and $y_i \in \{1, 0, \frac{1}{2}\}$ the corresponding user preference. Each agent $m$ is assigned a strength parameter $\beta_m$, and the probability that the left agent wins is modeled as:

$$\Pr(m^{\mathsf{L}} \succ m^{\mathsf{R}}) = \frac{\exp(\beta_{m^{\mathsf{L}}})}{\exp(\beta_{m^{\mathsf{L}}}) + \exp(\beta_{m^{\mathsf{R}}})}.$$

We optimize the log-likelihood of all votes to estimate $\boldsymbol{\beta}$, and convert scores to the standard Elo scale via:

$$E_m = 400 \log_{10}(e^{\beta_m}) + 1000.$$

To ensure leaderboard stability, we compute 95% confidence intervals via bootstrap and rank agents by the lower bound of their interval. The Appendix C shows full optimization and leaderboard details.

## 3 EXPERIMENTS

We present our experimental setup and leaderboard results, followed by analysis of task distribution, data validation and cross-benchmark comparisons.

## 3.1 EXPERIMENTAL SETUP

**Agent Models and Sampling.** We evaluate 12 publicly accessible CUAs spanning three representative groups. **(i) Proprietary computer-use agents** with strong benchmark performance: Claude 4

Sonnet (Anthropic, 2025b), Claude 3.7 Sonnet (Anthropic, 2025a), UI-TARS-1.5 (Qin et al., 2025), Operator (OpenAI, 2025d), and Claude 3.5 Sonnet (New) (Anthropic, 2024). **(ii) Open-source CUAs**: OpenCUA-32B (Wang et al., 2025), Qwen 2.5 VL 72B (Bai et al., 2025), and CoAct-1 (Song et al., 2025); notably, CoAct-1 is a universal digital agent integrating API tool calls and code execution. **(3) General foundation models.** We additionally include strong general-purpose multimodal models not previously tested in this setting (e.g., GPT-5 (OpenAI, 2025b), GPT-4.1 (OpenAI, 2025a), OpenAI o4-mini (OpenAI, 2025c), and Gemini 2.5 Pro (Google DeepMind, 2025)). All agents follow the unified protocol in Section 2.2 and are sampled with uniform probability during evaluation. For quality control, agents with $< 10\%$ correctness in their first 100 votes are removed to preserve data quality and user experience.

**Crowdsourcing Evaluation**    We collected evaluation data from two sources: (1) public users on the COMPUTER AGENT ARENA platform and (2) paid crowd workers on Prolific (Peer et al., 2021), a high-quality crowdsourcing platform. Prolific participants were pre-screened for prior experience with LLM tools (e.g., ChatGPT, Claude) with details in Appendix B.3.1. All users had to submit real computer-use tasks and evaluate anonymized agent trajectories via pairwise preference judgments and other labels. To ensure that the resulting evaluations reflect realistic usage patterns, we recruited annotators from diverse demographic backgrounds, spanning multiple countries, educational levels, and professions, detailed statistics are reported in Appendix B.3.1.

**Data Filtering**    To ensure data quality, we applied post-hoc filtering to remove duplicate instructions, off-topic queries (e.g., math problems, chit-chat), and tasks incompatible with GUI-based execution. For further validation of annotation consistency, we conducted an inter-annotator agreement study, sampling 100 human-labeled trajectories from the dataset and assigning them to three different annotators for independent labeling. Krippendorff's $\alpha$ scores were calculated for each label type, yielding values of $= 0.72$ for preferences, $= 0.78$ for correctness, $= 0.68$ for safety, and $= 0.70$ for efficiency. These results demonstrate a moderate-to-strong level of agreement among annotators, confirming the reliability and consistency of our labeled data. Full details of the filtering procedure, including these consistency results, are provided in Appendix B.3.2.

## 3.2    MAIN RESULTS

| Rank | Model | Elo | Votes | Correct Rate |
|------|-------|-----|-------|--------------|
| 1 | Claude Sonnet 4 | 1167 | 416 | 52.0% |
| 2 | Claude 3.7 Sonnet | 1140 | 507 | 52.3% |
| 3 | UI-TARS-1.5 | 1092 | 533 | 49.9% |
| 4 | Operator | 1064 | 511 | 37.4% |
| 5 | CoAct-1* | 1043 | 110 | 41.8% |
| 6 | OpenCUA* | 1023 | 109 | 38.5% |
| 7 | Claude 3.5 Sonnet | 1023 | 425 | 35.8% |
| 8 | GPT-5* | 1002 | 108 | 34.3% |
| 9 | o4-mini | 895 | 266 | 15.4% |
| 10 | Qwen 2.5 VL 72B Instruct | 895 | 504 | 15.9% |
| 11 | GPT-4.1 | 837 | 432 | 8.6% |
| 12 | Gemini 2.5 Pro | 829 | 377 | 11.8% |

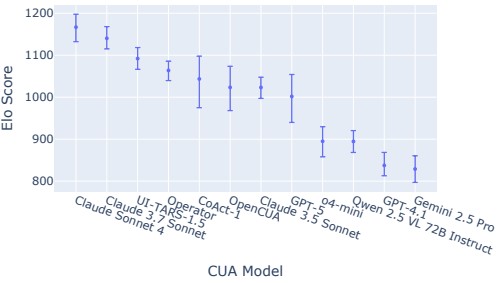

(a) COMPUTER AGENT ARENA leaderboard with Elo scores, vote counts, and correctness rates.

(b) Bootstrapped Elo scores with 95% confidence intervals.

Figure 2: Model performance on COMPUTER AGENT ARENA based on 2,201 pairwise user votes. (a) presents the leaderboard; (b) shows bootstrapped Elo score distributions. (* denotes models with limited votes; results will be updated in later versions.)

We collected a total of 3,418 evaluation votes, including 1,773 from public users and 1,645 from Prolific participants. After filtering invalid submissions and removing low-performing models, we retained 2,201 high-quality preference votes for analysis. In total, the platform engaged 1,058 unique users, comprising 821 Prolific participants and 237 public users.

**Leaderboard Results.**    Figure 2 shows the COMPUTER AGENT ARENA leaderboard based on over 2,201 pairwise votes from 1,058 users across 12 CUA models. Claude Sonnet 4 and Claude 3.7 Sonnet dominate the rankings with a clear Elo margin, followed by UI-TARS-1.5 and OpenAI Operator.

In contrast, general-purpose models like GPT-5 and Gemini 2.5 Pro rank lower, suggesting that strong multimodal capabilities do not necessarily translate to robust computer-use performance. We also compare the COMPUTER AGENT ARENA leaderboard with four public benchmarks: OSWorld, WebArena (Zhou et al., 2024), WebVoyager (He et al., 2024) and Online-Mind2Web (Xue et al., 2025). Figure 3a summarizes how existing models rank across all five leaderboards.

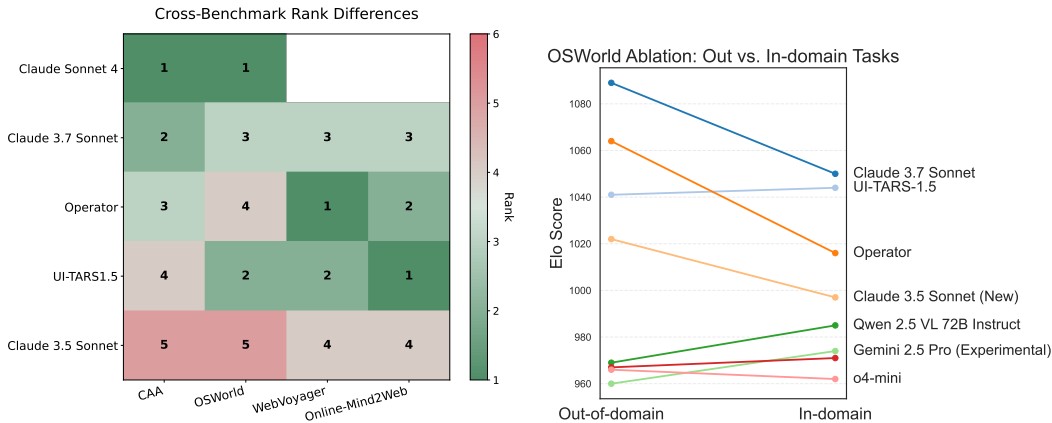

(a) Relative ranks of 12 models across COMPUTER AGENT ARENA and four public benchmarks. Green = higher (better); red = lower; blank = not evaluated.

(b) Elo scores on COMPUTER AGENT ARENA after splitting 1,000 tasks into OSWorld-In-domain (left) and OSWorld-OOD (right) subsets.

Figure 3: (a) Cross-benchmark comparison highlights rank inversions among top CUAs. (b) Ablation confirms rankings shift once evaluation moves away from OSWorld templates.

**Statistical Validation**   We validate that observed model differences are not artifacts of sampling variance using three complementary tests (details in Appendix D). Bootstrapped Elo scores provide narrow 95% confidence intervals. Permutation tests, implemented by repeatedly shuffling preference labels to form a null distribution, confirm that pairwise win-rate differences are highly significant ($p < 0.01$) with medium-to-large effect sizes (Cohen's $d > 0.5$). Power analysis, conducted by simulating effect detection under varying sample sizes, shows that with typical data ($>200$ votes per model pair) the probability of detecting medium effects ($\Delta$Elo $\approx 50$) exceeds 0.9. Together, these results demonstrate that the ranking gaps reported in Figures 10b, 11a, and 5 are statistically significant and robust.

**Task distribution strongly affects leaderboard rankings, revealing the fragility of static benchmarks.**   To examine how task distribution influences agent rankings, we perform an ablation study on the COMPUTER AGENT ARENA dataset. Using GPT-4o as a semantic classifier, we sampled 1,000 user-submitted tasks as either *in-domain* or *out-of-domain*. After manual verification, we recompute Elo scores for each subset separately. As shown in Figure 3b, leaderboard orders shift markedly: while Claude 3.7 Sonnet remains on top, models like UI-TARS-1.5 rise under in-domain tasks. These findings highlight that static benchmarks often overestimate agent performance by overfitting to narrow task distributions. In contrast, COMPUTER AGENT ARENA captures a broader task landscape through crowd-sourced inputs, which is a more reliable reflection of model robustness in real-world usage.

## 4   ANALYSIS

We analyze COMPUTER AGENT ARENA from four perspectives: task diversity, user preference, agent behavior, and error study. This allows us to characterize how CUAs are evaluated in real-world scenarios and to identify key divergences from static benchmarks.

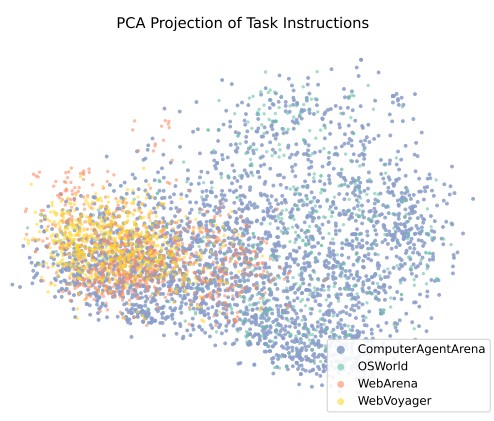

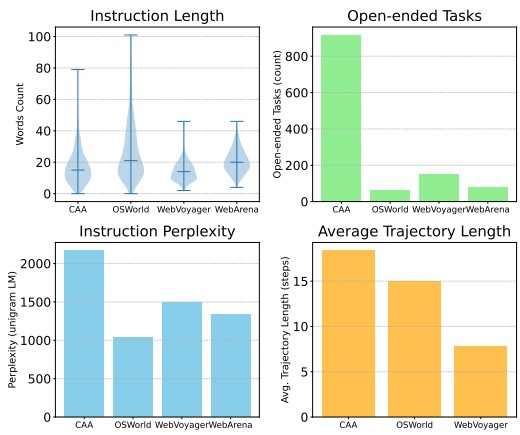

(a) PCA projection of task instructions across benchmarks.

(b) Instruction-level statistics, where "CAA" denotes COMPUTER AGENT ARENA.

Figure 4: Comparison of task distributions across four CUA benchmarks. Figure (a) shows the semantic projection of task instructions via PCA, highlighting coverage differences. Figure (b) shows instruction length, open-endedness, perplexity, and trajectory length across benchmarks, with COMPUTER AGENT ARENA ("CAA") exhibiting higher ambiguity and longer interaction sequences.

## 4.1 TASK ANALYSIS

**Tasks in COMPUTER AGENT ARENA feature broader semantic and domain coverage than prior benchmarks.** We analyze the semantic and linguistic characteristics of task instructions across COMPUTER AGENT ARENA, OSWORLD, WEBARENA, and WEBVOYAGER. As shown in Figure 9a, a PCA projection indicates that COMPUTER AGENT ARENA occupies a broader and less clustered semantic space, suggesting greater topical diversity. We further compare instruction-level features, including instruction length, open-endedness, unigram perplexity, and reference-trajectory length. Figure 9b shows that COMPUTER AGENT ARENA tasks are shorter on average (mean ≈ 17 words) yet substantially more ambiguous—reflected by nearly double the perplexity—and more likely to lack a canonical ground-truth answer. This pattern reflects a key trend: real users frequently issue concise but underspecified queries, which require agents to infer implicit intent and reason iteratively rather than converge on a fixed end state.

## 4.2 USER PREFERENCE ANALYSIS

We analyze behavioral features and outcome labels from COMPUTER AGENT ARENA to identify the factors that truly shape user preferences.

**Correctness is a leading predictor user preference, while execution length and latency have negligible impact.** Building on the prior observation that users value coherent execution, we next quantify how correctness and efficiency metrics correlate with user preference. Leveraging explicit correctness labels from our evaluation pipeline, we observe a strong linear relationship between model correctness and win rate (Figure 11a), indicating that task success is the most influential factor in shaping user judgments. To isolate the effect of execution efficiency, we analyze sessions where both agents were marked "correct" and compare their step counts and average latencies. As

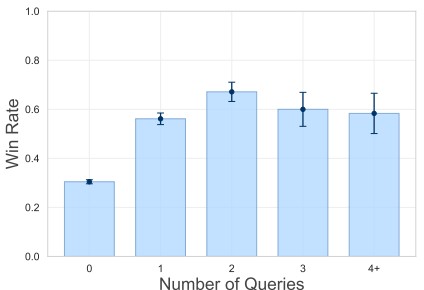

Figure 5: Win rate grouped by number of CALL_USER queries. Moderate querying (1–2 times) leads to higher user preference, while excessive or no querying lowers win rate.

shown in Figure 11b, neither factor shows a consistent influence on user preference—users do not systematically prefer faster or shorter executions when outcome quality is held constant. Overall, these

results suggest that while concise or responsive behavior may offer slight advantages, correctness remains the dominant signal guiding user preferences in current CUA systems.

**User preferences are shaped more by *turn-level* completeness rather than by *final state*.** We conduct a quantitative case study of 100 selected cases to better understand why top-performing agents diverge in real-world evaluation. Our analysis reveals that, unlike static benchmarks which assess agents solely based on final states, users on COMPUTER AGENT ARENA often base their preferences on holistic execution quality. As illustrated in Example 2 in Appendix J, agents that fail to complete the task may still be preferred if they demonstrate clear intent understanding, partial progress, or thoughtful error recovery and self-correction attempts. This preference is especially evident in open-ended tasks with no unique solution, where users value adaptive and coherent behavior over rigid outcome success. These findings highlight the importance of evaluating *how* an agent performs, not just *what* it accomplishes, emphasizing the need for turn-level assessment in realistic human–agent evaluation.

**Agent-human interaction improves agent preference, while excessive queries hinder performance.** Beyond execution correctness, human–agent interaction plays a key role in shaping preferences. Unlike prior benchmarks that treat CUAs as fully autonomous systems (Xie et al., 2024a; Zhou et al., 2024), COMPUTER AGENT ARENA enables an explicit human–agent interface via the `CALL_USER` action (Sec. 2.2). Figure 5 shows that *moderate* querying—typically one or two calls—correlates with the highest win rates, whereas both zero and *excessive* queries are associated with lower user preference. This inverted U-shaped pattern suggests that users value well-timed clarification, especially for underspecified tasks, yet tend to penalize over-reliance on human input. These results indicate that effective interaction is not merely about asking questions, but about demonstrating autonomy and judgment—traits users increasingly expect from capable CUAs.

## 4.3 TOOL-INTEGRATED VS. PURE GUI CUAS

Recent CUA work emphasizes *tool-augmented* architectures that invoke code interpreters or APIs to improve success on structured tasks. Although such agents often surpass GUI-only models on scripted benchmarks, our human-centric evaluation reveals a different pattern in real use. For example, CoAct-1 (Song et al., 2025)—a tool- and coding-integrated agent based on OpenAI o3 (OpenAI, 2025c)—reports state-of-the-art results on OSWORLD-VERIFIED (60.1% success), yet exhibits substantially lower performance in COMPUTER AGENT ARENA, particularly on non-technical tasks. This contrast underscores a benchmark–real-world gap: scripted suites tend to reward deterministic tool pipelines, whereas real user instructions are often underspecified and not always tool-benefiting.

The evaluation results of CoAct-1 show two salient characteristics in our evaluation: (1) On tasks naturally suited to tooling (e.g., code debugging, technical workflows), CoAct-1 is competitive—especially among tech-savvy raters. (2) On general-purpose tasks, *tool selection* frequently misfires, producing errors invisible in the GUI replay and leading to lower user preference. Notably, its successful trajectories are very short (mean ≈ 3 steps), suggesting that tool calls can solve narrow problems efficiently but generalize poorly across open-ended tasks. We trace the gap to two factors: (i) **Tool-selection bias**, where agents over-invoke coding tools on tasks better served by direct GUI actions; (ii) **Error amplification**, where tool calls produce opaque, non-surfaced failures that undermine interpretability and user trust.

Our findings indicate that adding more tools to CUAs does not inherently translate into better real-world performance; misapplied tooling can degrade both accuracy and user satisfaction. Two implications follow: (1) **Tool definition and selection matter.** Future designs should develop adaptive policies that decide *when, what, and how* to invoke tools—including abstaining, clarifying with the user, or falling back to direct GUI actions. (2) **Mind the benchmark–real-world gap.** Benchmarks that overweight tool-centric pipelines risk misrepresenting user needs. By exposing these discrepancies, COMPUTER AGENT ARENA offers a lens to study tool-use strategies and their usability impact in open-ended environments.

### 4.4 ERROR ANALYSIS

While quantitative metrics such as Elo and correctness provide a global view of model performance, they cannot reveal the nuanced behavioral failures that limit agent usability in realistic settings. A key advantage of COMPUTER AGENT ARENA is that human preference annotations surface diverse error modes that scripted benchmarks fail to capture. Many failed trajectories stem from common issues such as *grounding errors* (e.g., mis-clicking a button) and *planning errors* (e.g., lacking a clear execution strategy), which reflect base model limitations and can be mitigated with training. Beyond these expected cases, however, we identify three insightful error types that are subtler, more detrimental, and harder to expose with scripted tasks:

- **Long-horizon memory failures.** In tasks requiring long-horizon context or repetitive workflows, CUAs often lose track of key information after many steps (e.g., converting multiple files or compiling season-long statistics). Even state-of-the-art models such as Claude 4 Sonnet frequently drift or forget intermediate goals.

- **Information awareness.** Real-world queries are often underspecified, omitting details such as file paths or intention slots. Purely GUI-based agents typically issue speculative commands rather than clarifying uncertainties, leading to compounding errors. By contrast, models like Claude 4 Sonnet and Operator effectively invoke `CALL_USER` to resolve ambiguity, underscoring the importance of interactive mechanisms.

- **Fine-grained action failures.** Even when intent is clear, execution breaks down due to fine-grained control errors, including mishandled scrolling, clicks on non-interactive elements, or faulty text editing (e.g., appending instead of replacing). These mistakes, sitting between planning and grounding, often derail entire tasks and highlight the need for new training signals on low-level action precision.

Together, these categories show how COMPUTER AGENT ARENA functions as a systematic error discovery pipeline. By surfacing nuanced failure modes beyond correctness, our analysis provides actionable insights for improving memory, uncertainty handling, and fine-grained grounding in future CUAs.

## 5 RELATED WORK

**Computer-Use Agent Benchmarks**   Recent advancements have fostered diverse benchmarks and evaluation frameworks for assessing computer-use agents. Within GUI agent contexts, evaluations mainly target scenarios involving web navigation (Zhou et al., 2024; Deng et al., 2023; Koh et al., 2024; Drouin et al., 2024) and computer-use (Xie et al., 2024a; Bonatti et al., 2024; Davydova et al., 2025; Xie et al., 2025a). OS-level benchmarks, such as OSWorld-Verified (Xie et al., 2024a; 2025b) and Windows Agent Arena (Bonatti et al., 2024), provide rule-based evaluations of desktop agents performing various GUI tasks. Recent works like AgentCompany (Xu et al., 2024a), AndroidWorld (Rawles et al., 2024), Online-Mind2Web (Xue et al., 2025), AgentNetBench (Wang et al., 2025) extend evaluations toward longer horizons, leveraging interactive and multi-modal scenarios. Although these frameworks have significantly contributed toward standardized assessments, they remain largely script-based and static. Our platform, COMPUTER AGENT ARENA, further expands upon these ideas by crowd-sourcing authentic user-defined tasks and emphasizing dynamic, human-centric evaluations of computer-use agents.

**Human-centric Evaluation**   Human preference evaluation has become increasingly significant for tasks lacking clearly defined ground truths, such as dialogue agents and interactive applications. Platforms like Chatbot Arena (Chiang et al., 2024) popularized large-scale, pairwise human comparisons for evaluating language models, using Bradley–Terry–based Elo ranking systems. This paradigm was subsequently adapted to domains like programming assistants (Copilot Arena (Chi et al., 2025)), large audio models (Li et al., 2025b), and text-based gaming agents (TextArena (Guertler et al., 2025)). These arenas highlight the value of integrating human judgment into model assessment, enabling nuanced distinctions that conventional benchmarks or scripted evaluations might overlook. Despite their success, existing preference-based platforms primarily focus on isolated, context-independent interactions without grounding evaluations within users' real environments. By bridging the gap

between evaluation and deployment, our platform lays the groundwork for more ecologically valid assessments and provides a foundation for future research on agents operating in the wild.

## 6 CONCLUSION

We present COMPUTER AGENT ARENA, a human-centric framework for evaluating Computer-Use Agents (CUAs) that enables scalable, diverse, authentic, and fair head-to-head comparisons in realistic, dynamic environments. We collect 2,201 filtered, high-quality votes across specialized CUA models and general-purpose foundation models on diverse, open-ended tasks, uncovering insights that static, scripted suites overlook. In particular, our *error study* surfaces systematic shortcomings of current CUAs (e.g., tool-selection/use errors, long-horizon memory and plan drift, fine-grained action/grounding issues), while our *human preference analysis* shows that users value process quality, agent-human interaction, recovery from errors, and privacy awareness. We open-source the evaluation platform, reference agent implementations, and a large, multimodal, human-labeled preference dataset to facilitate future research on CUAs. Ultimately, we underscore the centrality of a *human-centric perspective* for AI agent design and deployment in real-world use.

## ACKNOWLEDGEMENTS

We thank Graham Neubig, Daniel Fried, Shuyan Zhou, Pengcheng Yin, Hao Zhang for their helpful feedback on this work. This project was supported by a gift funding from AWS AI and support from the ECS (27212023) and Areas of Excellence Scheme (AoE/E-601/24-N) from RGC of Hong Kong

## ETHICS STATEMENT

This work involves the collection of human preference data through both public participation and paid crowdworkers. All participants provided informed consent prior to participation: public users agreed to the consent form before using the COMPUTER AGENT ARENA platform, and paid annotators recruited via Prolific were required to read and accept an explicit consent statement before starting tasks. Paid annotators interacted only through anonymized accounts, and no personally identifiable information was collected or stored. For the final dataset release, we applied a sensitive data filtering process to remove submissions containing personal or inappropriate content, ensuring that all data remain anonymized and privacy-preserving.

We acknowledge that crowdsourced data may contain demographic and cultural biases. To assess fairness, we conducted a demographic study of annotators, which confirmed a broad distribution across countries, education levels, and professional backgrounds (Appendix B.3.1).

The dataset and platform are released solely for academic research purposes. They are intended to advance evaluation methodology and alignment for computer-use agents, and must not be used for surveillance, discriminatory decision-making, or other potentially harmful applications. This study was conducted in accordance with institutional ethical guidelines, and all authors confirm compliance with the ICLR Code of Ethics. We declare that there are no conflicts of interest or external sponsorships influencing this work.

## REPRODUCIBILITY STATEMENT

We are committed to ensuring the reproducibility and transparency of our work. All code for the COMPUTER AGENT ARENA platform, including frontend, backend, evaluation logic, and ranking scripts, will be released under an open-source license. To guarantee consistent runtime environments, we additionally provide pre-built Amazon Machine Images (AMIs) for both Windows and Ubuntu, and a plug-and-play agent hub for integrating external models and enabling third-party leaderboard participation (Sec. 2.2 and Appendix B.2). Detailed descriptions of data filtering, annotation protocols, and demographic studies are provided in Appendix B.3.2 and Appendix B.3.1, ensuring transparency in data collection and quality control. We also report evaluation cost and throughput analyses in the Appendix H, confirming that the framework is both affordable and scalable for community use. Together, these efforts provide a reproducible end-to-end pipeline for benchmarking, training, and behavioral diagnostics of computer-use agents in realistic environments.

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

# TABLE OF CONTENTS IN APPENDIX

## A    LLM USAGE STATEMENT

We provide a disclosure of how LLMs were used in the preparation of this work. LLMs were employed as a general-purpose assistive tool in the following ways:

- **Writing support and refinement:** LLMs were used to assist with phrasing, grammar checking, and stylistic polishing of the manuscript to improve clarity and readability. All technical content, claims, and arguments were conceived, verified, and finalized by the authors.
- **Data analysis assistance:** LLMs were occasionally used to help with exploratory data analysis (e.g., generating Python code snippets for statistical tests, summarizing results of bootstrapping or agreement studies), with all outputs verified and validated by the authors.

LLMs were not used for the ideation of research questions, experimental design, or the generation of novel scientific claims. All conceptual contributions, dataset collection, experiments, and final interpretations are the sole responsibility of the authors. We emphasize that LLMs are not contributors or authors, and that the authors take full responsibility for the correctness and integrity of all content reported in this paper.

## B    SYSTEM DETAILS

### B.1    PLATFORM IMPLEMENTATION DETAILS

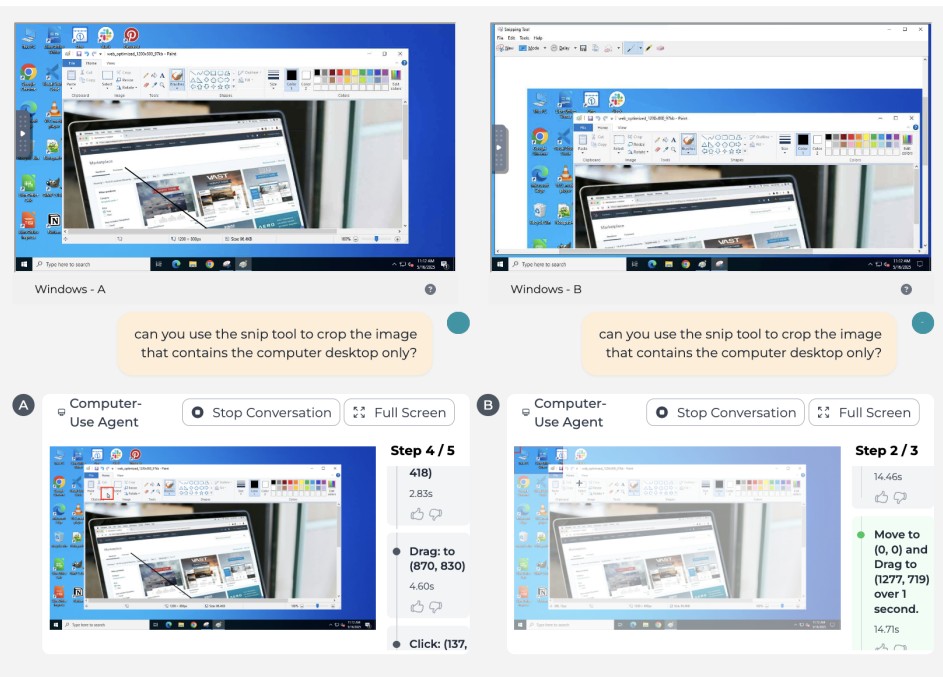

Figure 6: User interface of the COMPUTER AGENT ARENA platform. Users are presented with two side-by-side VNC desktops representing anonymized CUA executions. Task instructions appear at the top, while trajectory replays and voting options are shown below.

To support scalable evaluation of CUAs under realistic conditions, we extend the OSWorld image and build a private Amazon Machine Image (AMI) deployed on AWS EC2. This AMI serves as the base environment for launching on-demand virtual machines. Upon receiving a task execution request from the frontend, our backend service either allocates two pre-warmed instances or dynamically launches new EC2 instances depending on current load. Each evaluation session involves two such virtual machines to facilitate parallel execution and pairwise comparison.

For real-time human interaction, we enable WebSocket Secure (WSS) communication to expose VNC (Virtual Network Computing) interfaces. These interfaces are embedded in the frontend using an iframe-based VNC viewer, allowing users to directly observe and interact with the virtual desktop in real time. This mechanism ensures a high-fidelity, low-latency connection, even under concurrent usage. To ensure performance robustness, the infrastructure supports multi-region deployment and load balancing, which minimizes cold start latency and improves responsiveness during high-traffic periods.

Figure 6 provides a visual overview of the COMPUTER AGENT ARENA evaluation interface. On the frontend, we implement the entire user interface in React and TypeScript. We leverage built-in OBS (Open Broadcaster Software) services installed within the VM to record agent trajectories. Specifically, we capture key frames before and after each agent-issued action (e.g., mouse click, keystroke, scroll) and compile these frames into animated `.gif` files. These gifs are rendered sequentially on the user interface to form an interpretable execution timeline. Each frame is annotated with overlays (e.g., bounding boxes for clicks, keyboard icons for typing) to help users understand the agent's interaction logic at each step.

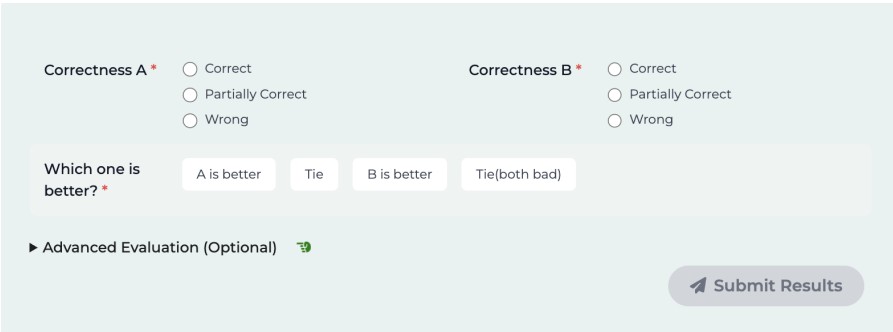

Figure 7: Evaluation form interface shown after users review the agent trajectories. Users select a preferred agent (or tie), mark correctness labels for each agent, and can optionally provide comments.

To support pairwise evaluation, the two anonymous agent trajectories are rendered side-by-side. When agents invoke the `CALL_USER` action during execution, the frontend renders a user-facing message box to present the agent's clarification query. After watching the side-by-side execution trajectories, users are directed to the evaluation form (Figure 7). This form collects pairwise preference judgments by asking users to indicate which agent performed better or whether both agents were equally good or bad. Additionally, users assign correctness labels to each agent (*Correct*, *Partially Correct*, or *Wrong*), allowing for fine-grained accuracy analysis. Optional text fields also enable users to explain their decisions or report abnormal behavior. All evaluation artifacts, including task metadata, screen recordings, and user responses, are logged for subsequent analysis and benchmarking.

## B.2 INITIAL SETUP DETAILS

To ensure realistic diversity across evaluation environments and reduce duplication, we implement a structured pipeline for configuring initial virtual machine states. Following the design philosophy of OSWorld (Xie et al., 2024a), we predefine a large number of environment setups through scripted configuration files. When a user initiates an evaluation session, the backend randomly selects and applies one of these predefined scripts to both virtual machines, ensuring a synchronized and comprehensive starting state.

**Website Configuration.** We curate a diverse pool of popular websites by crawling the top 100 high-traffic domains from SimilarWeb (Similarweb Ltd., 2025). After manual filtering to exclude login-gated or geoblocked domains, we expand the pool by collecting up to 10 subdomains per top-level site. These subdomains are injected into the browser history or opened as startup tabs within the VM, providing agents with richer content structures and deeper navigation targets during execution.

Table 1: Summary of Initial Environment Setup Components

| Category | Source | Example Types | Count |
|---|---|---|---|
| Websites | SimilarWeb | Wikipedia, YouTube, GitHub | 89 |
| Subdomains | Manual Expansion | en.wikipedia.org, studio.youtube.com | 472 |
| Applications | Microsoft Store & Snapcraft | LibreOffice, VLC, VS Code | 12 |
| Files | Synthetic Corpus | .docx, .xlsx, .pptx, .pdf, .py, .md | 97 |

**Application Configuration.** To simulate realistic usage patterns, we preload the virtual machines with over 20 frequently used desktop applications. These are sourced from the Microsoft App Store (Microsoft Corporation, 2025) and Snapcraft (Canonical Ltd., 2025), spanning categories such as document editing, media playback, messaging, and terminal utilities. Each application is installed system-wide and accompanied by prewritten startup scripts to ensure reproducibility across instances.

**File System Setup.** We populate the virtual machine file system with over 100 heterogeneous files, covering common formats such as `.docx`, `.xlsx`, `.pptx`, `.pdf`, `.py`, and `.md`. These files simulate a working desktop environment and provide agents with realistic artifacts to manipulate during task execution. To avoid agent overfitting, we update the file content monthly and apply randomized filenames and folder structures at each session launch.

**User Customization Interfaces.** Beyond static setups, we implement a suite of "quick-start" APIs that allow users to tailor the VM environment. Specifically:

- `upload_file` allows users to upload local files directly to both VMs.
- `open_websites` enables users to preload specific URLs into browsers.
- `clone_repo` automates the cloning of GitHub repositories via a single command.

These tools enhance fidelity and flexibility in modeling users' native computing environments, making the evaluation experience more aligned with realistic desktop usage scenarios.

### B.3 AGENT IMPLEMENTATION DETAILS

**Interaction Protocol.** We model the agent-computer interaction as a sequential decision process:

$$(I, \langle s_0, a_0 \rangle, \langle s_1, a_1 \rangle, \dots, \langle s_t, a_t \rangle) \tag{1}$$

Here, $I$ is the initial task instruction provided by the user; $s_i$ is the screenshot of the desktop at step $i$; $a_i$ is the action taken by the agent at that step. The agent conditions its decisions on $I$, the current observation $s_t$, and a history buffer of $n$ past state-action pairs ($n = 5$ by default), allowing limited memory and context tracking.

**Observation.** Each observation $s_t$ is a full-color PNG screenshot of the entire virtual desktop, captured at a fixed resolution of $1280 \times 720$. These raw pixels are encoded and sent to the agent model without any additional metadata or structured UI representations.

We do not provide DOM trees, accessibility layers, or pre-parsed UI elements, due to their inherent latency and inconsistency across software stacks. By enforcing a vision-only interface, we ensure that agents are robust to general GUI variation and layout changes.

**Action Space.** Our platform adopts a structured action format modeled after the PyAutoGUI API. The action space includes:

- **Mouse Interactions:** `click(x, y)`, `double_click(x, y)`, `move(x, y)`, `scroll(amount)`
- **Keyboard Interactions:** `type(text)`, `press(key)`, `hotkey([key1, key2, ...])`

- **Meta Actions:** DONE (terminate when task is complete), FAIL (terminate if task is un-achievable), CALL_USER (request user feedback or clarification)

All actions are serialized into a JSON-compatible function-call format that supports easy logging, replay, and debugging.

**Action Execution Workarounds.** To ensure consistent and reliable agent behavior across diverse operating systems and software environments, we introduce two key code-level workarounds that address common compatibility and reliability issues in action execution.

On some systems, the pyautogui.write and typewrite functions suffer from unpredictable behavior—especially for non-English input or special characters. To improve compatibility and ensure multi-language text entry behaves consistently, we transform these commands into clipboard-based pasting operations. Specifically, we use the pyperclip module to set clipboard contents and then simulate a ctrl+v hotkey to inject the text. The original clipboard contents are saved and restored to minimize side effects. The transformation logic is shown below:

```python
def workaround_pyautogui_write(original_code: str) -> str:
    import re
    lines = original_code.splitlines(keepends=True)
    pattern = re.compile(
        r'^(\s*)pyautogui\.(write|typewrite)\s*\(\s*(["\'])(.*?)\3\s*(.*)\)\s*$'
    )
    has_write = any(pattern.match(line) for line in lines)
    if has_write:
        transformed_lines = ["import pyperclip\n",
                             "original_clipboard = pyperclip.paste()\n\n"]
        for line in lines:
            match = pattern.match(line)
            if match is None:
                transformed_lines.append(line)
            else:
                indent, _, _, text, _ = match.groups()
                transformed_lines.append(f"{indent}pyperclip.copy({repr(text)})\n")
                transformed_lines.append(f"{indent}time.sleep(0.2)\n")
                transformed_lines.append(f"{indent}pyautogui.hotkey('ctrl','v')\n")
                transformed_lines.append(f"{indent}pyperclip.copy(original_clipboard)\n")
        return "".join(transformed_lines)
    return original_code

def workaround_pyautogui_scroll(original_code: str) -> str:
    import re
    lines = original_code.splitlines(keepends=True)
    pattern = \
        re.compile(r'^(\s*)pyautogui\.scroll\s*\(\s*(\d+)\s*,\s*(\d+)\s*,\s*(\d+)\s*\)\s*$')
    has_scroll = any(pattern.match(line) for line in lines)
    if has_scroll:
        transformed_lines = ["import pyautogui\n"]
        for line in lines:
            match = pattern.match(line)
            if match is None:
                transformed_lines.append(line)
            else:
                indent, clicks, x, y = match.groups()
                transformed_lines.append(f"{indent}pyautogui.moveTo({x}, {y})\n")
                transformed_lines.append(f"{indent}pyautogui.scroll({clicks})\n")
        return "".join(transformed_lines)
    return original_code
```

Listing 1: PyAutoGUI Workaround

In certain environments, pyautogui.scroll behaves inconsistently when used with three arguments (e.g., scroll(clicks, x, y)), especially on Windows or within virtual desktops. To ensure coordinate-aware scroll behavior works uniformly, we split the scroll into two atomic operations: first, moving the mouse to the intended location, then performing the scroll. The adjusted transformation is shown below:

These workarounds enable robust agent behavior under diverse conditions and reduce errors arising from OS-specific inconsistencies. The transformation logic is applied automatically to all agent-generated code prior to execution within the virtual environment.

**Execution Interfaces.** Agents interact with the virtual machine through two API calls:

- `get_observation()` → Returns a base64-encoded PNG screenshot
- `step(action)` → Executes the given action.

The interface is stateless and HTTP-based, allowing deployment across different VM backends and containerized setups.

**Model Integration.** We support two modes of model integration:

1. **Official Agent Implementation:** For models such as OpenAI Operator or Claude-Sonnet-CU, the agent logic is implemented and maintained by the model provider. We interface with their inference server and wrap the input/output into our standardized protocol, translating their tool calls into a unified action format.
2. **Baseline Agent Implementation:** For open-source models or checkpoints without official implementations, we offer a default baseline wrapper. This wrapper is designed to handle screenshot ingestion, instruction prompting, function call parsing, step-wise planning, and tool-use execution across Ubuntu and Windows environments. The logic also includes configurable termination criteria, retry-on-failure, and error reflection.

All logs, actions, screenshots, and intermediate CoTs are saved for every session to support evaluation and visualization. The framework is open-sourced and modular, allowing easy integration of new models.

**Baseline Prompt Implementation.** The default inference prompt used in our platform is structured as follows:

```
System Prompt for Baseline Agent - Ubuntu

You are an agent performing desktop tasks as instructed, with
    knowledge of computers and internet access. Your tool calls will
     control mouse and keyboard actions on a computer.

### Task Parameters:
- **Instruction**: {task_instruction}
- **Resolution**: {resolution}
- **Platform**: Ubuntu
- **System Password**: 'password' (for sudo rights if needed)

### Ubuntu-Specific Instructions:
- **Desktop Path**: `/home/user/Desktop`

### Observation Information:
Each step provides an observation that includes a screenshot with
    these characteristics:
- If previous mouse actions didn't achieve the expected result, do
    not repeat them, especially the last one - adjust the coordinate
     based on the new screenshot
- Do not predict multiple clicks at once. Base each action on the
    current screenshot; do not predict actions for elements or
    events not yet visible in the screenshot.
- Launching applications may take some time to appear on the
    desktop. If the screenshot indicates that the correct
    application has already been clicked, do not click it again-wait
     for it to open instead.

### Tool Call Information:
```

```
You are provided with computer-use tools which are defined in the
    tools clearly, you can use them to perform the task.
- REQUIRED: Accurate Positioning - Base each click/move/drag as
    precisely as possible on the screenshot coordinates. Use visual
    cues to approximate the exact location of the target.
- REQUIRED: Wait between multiple actions.

### Response Structure:
- Current observation analysis
- Results of any previous actions
- Any adjustments needed based on feedback
- The tool call you will use

### Requirements:
- YOU MUST USE THE PROVIDED TOOLS IN EVERY RESPONSE TO PERFORM THE
    TASK IN COMPUTER. ONLY SKIP TOOL IF YOU HAVE FINISHED THE TASK.
- You are a computer-use agent - please keep executing on the
    computer until the user's query is completely resolved, before
    ending your turn and yielding back to the user. Only terminate
    your turn when you are sure that the problem is solved.
- You MUST plan extensively before each function call, and reflect
    extensively on the outcomes of the previous function calls. DO
    NOT do this entire process by making function calls only, as
    this can impair your ability to solve the problem and think
    insightfully.
```

The system prompt implementation for Windows platform is similar to the Ubuntu platform, only replacing 'Platform-Specific Instructions'.

### System Prompt for Baseline Agent - Windows

```
You are an agent performing desktop tasks as instructed, with
    knowledge of computers and internet access. Your tool calls will
     control mouse and keyboard actions on a computer.

### Task Parameters:
- **Instruction**: {task_instruction}
- **Resolution**: {resolution}
- **Platform**: Windows

### Windows-Specific Instructions:
- **Desktop Path**: `C:\\Users\\Administrator\\Desktop`
- **Open Terminal Command**: `Win+R`, then type `cmd` and press `
    Enter`
- **Application Launch**: Desktop applications require a double-
    click to open

### Observation Information:
Each step provides an observation that includes a screenshot with
    these characteristics:
- If previous mouse actions didn't achieve the expected result, do
    not repeat them, especially the last one - adjust the coordinate
     based on the new screenshot
- Do not predict multiple clicks at once. Base each action on the
    current screenshot; do not predict actions for elements or
    events not yet visible in the screenshot.
- Launching applications may take some time to appear on the
    desktop. If the screenshot indicates that the correct
    application has already been clicked, do not click it
     a g a i n wait for it to open instead.
```

```
### Tool Call Information:
You are provided with computer-use tools which are defined in the
    tools clearly, you can use them to perform the task.
- REQUIRED: Accurate Positioning - Base each click/move/drag as
    precisely as possible on the screenshot coordinates. Use visual
    cues to approximate the exact location of the target.
- REQUIRED: Wait between multiple actions.

### Response Structure:
- Current observation analysis
- Results of any previous actions
- Any adjustments needed based on feedback
- The tool call you will use

### Requirements:
- YOU MUST USE THE PROVIDED TOOLS IN EVERY RESPONSE TO PERFORM THE
    TASK IN COMPUTER. ONLY SKIP TOOL IF YOU HAVE FINISHED THE TASK.
- You are a computer-use agent - please keep executing on the
    computer until the u s e r s query is completely resolved, before
     ending your turn and yielding back to the user. Only terminate
    your turn when you are sure that the problem is solved.
- You MUST plan extensively before each function call, and reflect
    extensively on the outcomes of the previous function calls. DO
    NOT do this entire process by making function calls only, as
    this can impair your ability to solve the problem and think
    insightfully.
```

### Function Call Tools Definition for Baseline Agent

```
tools = {
    "type": "function",
    "function": {
    "name": "desktop_automation",
    "description": "Perform desktop automation actions like mouse
        movements, clicks, keyboard input, and more.",
    "parameters": {
        "type": "object",
        "properties": {
            "action": {
                "type": "string",
                "enum": [
                    "key",
                    "hold_key",
                    "type",
                    "mouse_move",
                    "left_mouse_down",
                    "left_mouse_up",
                    "left_click",
                    "left_click_drag",
                    "right_click",
                    "middle_click",
                    "double_click",
                    "triple_click",
                    "scroll",
                    "wait",
                    "fail",
                    "done",
                    "call_user"
                ],
```

```
                    "description": "The action to perform. Available
                        actions include keyboard input, mouse movements,
                         clicks, and system operations. call_user is
                        used to call the user for further query."
                },
                "coordinate": {
                    "type": "array",
                    "items": {
                        "type": "integer"
                    },
                    "description": "The (x, y) pixel coordinates for
                        mouse actions. Required for mouse_move and
                        left_click_drag and scroll actions"
                },
                "duration": {
                    "type": "integer",
                    "description": "Duration in seconds for hold_key
                        and wait actions."
                },
                "scroll_amount": {
                    "type": "integer",
                    "description": "Number of 'clicks' to scroll.
                        Required for scroll action."
                },
                "scroll_direction": {
                    "type": "string",
                    "enum": ["up", "down", "left", "right"],
                    "description": "Direction to scroll. Required for
                        scroll action."
                },
                "start_coordinate": {
                    "type": "array",
                    "items": {
                        "type": "integer"
                    },
                    "description": "Starting (x, y) coordinates for
                        drag actions. Required for left_click_drag."
                },
                "text": {
                    "type": "string",
                    "description": "Text input for type, key, and
                        hold_key actions. Can also be used with click or
                         scroll actions to hold down keys while
                        performing the action."
                }
            },
        "required": ["action"]
    }
    }
}
```

**Agent Sampling**    To ensure a meaningful and balanced evaluation of CUA performance, we curated a pool of 12 models from two categories: (1) public CUA implementations that have reported results on existing computer-use benchmarks, and (2) general-purpose vision-language models (VLMs) with sufficient instruction-following capabilities, included to assess their zero-shot utility in realistic computer environments.

During each evaluation session, we sample two agents uniformly at random from the available pool. To prevent sampling bias, newly added agents are temporarily upweighted until their total vote count reaches parity with existing models. This ensures fair exposure in pairwise comparisons and stabilizes Elo ranking calculations.

To safeguard the quality of user experience and filter out underperforming models, we apply a lightweight correctness-based screening policy: if any model receives fewer than 10% correctness ratings over a window of 100 user votes, it is automatically retired from the pool and excluded from further evaluation. This policy mainly targets general VLMs lacking GUI grounding, whose performance may be unreliable in visual desktop environments.

Through this hybrid sampling and filtering strategy, we maintain both the diversity and reliability of agent comparisons while preserving fairness across the evaluation process.

### B.3.1 CROWD-SOURCING DATA COLLECTION DETAILS

**Consent and Ethical Compliance**    Before participating, all users—both Prolific-based and public—were required to review and agree to a digital consent form that outlined the nature and scope of data collection. The consent form clearly specified:

- The purpose of the study: to evaluate AI agents in realistic computer-use scenarios.
- The types of data collected: task instructions, screenshots, agent actions, user votes, and optional comments.
- The intended use of the data: for academic research and potential release under a permissive license (e.g., CC-BY 4.0).
- Voluntary participation and withdrawal: participants could exit at any time without providing a reason, and without loss of compensation or access.
- Anonymity and privacy: participants were instructed not to include any personally identifiable information in their submissions.

All study procedures were reviewed and deemed exempt by our institutional ethics board. For the public deployment, the terms of service shown on the platform homepage served as a binding usage agreement. The ToS reiterated the research purpose, prohibited inappropriate use, and informed users that submitted data may be reused under open academic licenses.

**Evaluation Crowdsourcing**    To collect human preference data for evaluating CUAs, we deployed our study on two channels: (1) unpaid public users on the COMPUTER AGENT ARENA platform, and (2) paid participants recruited via the Prolific platform (Peer et al., 2021), a reputable crowdsourcing service widely used for academic research.

For the Prolific-based study, we designed a structured evaluation task in which participants were asked to: (i) submit real-world computer-use tasks they personally care about, and (ii) evaluate anonymized trajectories from two competing agents using a pairwise preference interface. Before starting, all participants were required to review and agree to a digital consent form clearly describing the purpose of the study, data collected (task instructions, votes, optional comments), and their right to withdraw at any time without penalty.

Participants were pre-screened through Prolific's internal filters to ensure prior familiarity with large language models (e.g., ChatGPT, Claude), ensuring informed judgments. In accordance with the Prolific platform's fair pay policy, we paid all participants with the price of £6/hour, which meets or exceeds the legal minimum payment requirement on the platform.

In parallel, unpaid users accessing the public version of COMPUTER AGENT ARENA were offered free usage of the platform and invited to contribute evaluation data on a voluntary basis. These users also interacted with the same anonymized evaluation interface and followed the same task submission and voting protocols.

In total, we collected 682 valid evaluation votes from Prolific, contributed by 481 unique participants who successfully completed at least one voting task. These users spanned a range of demographic backgrounds and were geographically distributed across English-speaking regions. Their contributions accounted for approximately one-third of the final evaluation dataset and provided a reliable baseline of high-quality, vetted user judgments.

To quantify the resulting diversity, we conducted a post-task demographic survey of our Prolific annotators, with the following results:

- Participants spanned 20+ countries across North America, Europe, and Asia
- 47% held a bachelor's degree or higher, with the rest covering high school and associate-level education
- Employment status included full-time (32%), part-time (21%), and student (26%) populations
- Age distribution was broad (mean $\approx$ 33.2 years, SD $\approx$ 9.6)
- Ethnic backgrounds included White (54%), Asian (13%), and Black (8%)

These findings suggest that our annotator pool reflects a wide range of real-world user perspectives. We believe the resulting benchmark captures realistic and representative computer-use behaviors across a broad population.

### B.3.2 EVALUATION DATA FILTERING PROCEDURE

To ensure the reliability and relevance of the evaluation data, we performed a multi-stage post-hoc filtering process combining automated heuristics with manual review. The goal was to retain only high-quality, executable, and task-relevant sessions for leaderboard ranking and agent behavior analysis.

**Automated Filtering**    We first applied a set of automatic filters to remove:

- **Duplicate or near-duplicate task instructions**, based on normalized string matching and embedding similarity using SentenceTransformers (cosine similarity > 0.98).
- **Off-topic or irrelevant queries**, such as:
  - Non-computer-use tasks (e.g., "What is 2 + 2?", "Tell me a joke")
  - Chit-chat or open-ended dialogue prompts
  - Content requesting hallucinated data (e.g., "Give me a fake email address")
- **Tasks incompatible with GUI-based execution**, including those requiring internet accounts, payment, or system-level changes (e.g., rebooting).

**Manual Review**    To further improve data quality, we conducted a manual verification pass over the filtered dataset:

- **3 human reviewers** (including one of the authors) independently reviewed randomly sampled evaluation sessions (totaling $\sim$600) to ensure task executability and validity.
- Sessions with technical problems—such as browser rendering failures, agent output truncation, or corrupted trajectory logs—were flagged and removed.
- Votes with both agent outputs missing, or showing formatting artifacts (e.g., empty responses, code-only dumps without interaction), were discarded.

**Resulting Dataset**    After filtering, we retained 2,201 high-quality pairwise votes, each associated with a valid computer-use task and two successfully executed agent trajectories.

## C   ELO RANKING METHOD

To construct a dynamic and interpretable leaderboard based on human preference votes, we adopt an Elo-style ranking system grounded in the Bradley–Terry model (Bradley & Terry, 1952), as widely used in other pairwise evaluation settings (Chiang et al., 2024; Chi et al., 2025).

Each evaluation session results in a comparison outcome between two anonymized agents $m_i^{\mathsf{L}}$ and $m_i^{\mathsf{R}}$. We denote the $i$-th pairwise comparison result as:

$$x_i = (m_i^{\mathsf{L}}, m_i^{\mathsf{R}}), \quad y_i \in \{1, 0, \tfrac{1}{2}\},$$

where $y_i = 1$ indicates the left agent wins, 0 the right wins, and $\tfrac{1}{2}$ denotes a tie (including both "tie" and "tie (both bad)" responses, which make up roughly 33% of all votes).

We assign each agent $m$ a real-valued strength score $\beta_m$, and model the win probability of the left agent as:

$$\Pr(m_i^{\mathsf{L}} \succ m_i^{\mathsf{R}}) = \frac{\exp(\beta_{m_i^{\mathsf{L}}})}{\exp(\beta_{m_i^{\mathsf{L}}}) + \exp(\beta_{m_i^{\mathsf{R}}})}.$$

We then compute $\boldsymbol{\beta} \in \mathbb{R}^M$ by maximizing the (regularized) log-likelihood over all recorded votes:

$$\mathcal{L}(\boldsymbol{\beta}) = \sum_{i=1}^{n} \log \Pr(y_i \mid x_i; \boldsymbol{\beta}),$$

where ties are modeled as the average of the two win probabilities.

For interpretability, we convert the strength scores to standard Elo ratings via:

$$E_m = 400 \log_{10}(e^{\beta_m}) + 1000,$$

such that all agents begin with an Elo score of 1000 and adjust as more evidence accumulates.

To stabilize the leaderboard and report uncertainty, we compute 95% confidence intervals using bootstrapping (1,000 resamples over vote logs). Agents are ranked by the lower bound of their Elo confidence interval to ensure conservative movement and mitigate volatility when differences are not statistically significant.

To better contextualize the Elo leaderboard results, we include several supplementary visualizations that capture vote distribution, model correctness, and head-to-head performance dynamics. As shown in Figure 14, each CUA model received a reasonably balanced number of votes, with newly added models temporarily boosted in sampling frequency to ensure coverage. Figure 15 reports the user-assessed correctness rate for each agent, reflecting how often their outputs were deemed valid across all tasks.

To examine inter-model interactions in more detail, Figure 16 presents a pairwise heatmap of vote counts between agent pairs, illustrating the evaluation density across matchups. Building on this, Figure 17 shows the empirical win rates between agents, highlighting not just the aggregate strength but also performance asymmetries in specific matchups—for instance, some models consistently outperform certain others while struggling elsewhere.

Lastly, Figure 18 visualizes the pairwise win probabilities predicted from our fitted Elo model, offering a smooth, interpretable estimate of how likely each agent is to win in head-to-head comparisons. These analyses collectively validate the robustness of our ranking method and reveal fine-grained interaction patterns that go beyond overall Elo scores.

# D   STATISTICAL VALIDATION OF RANKINGS

To ensure that the ranking differences reported in COMPUTER AGENT ARENA are robust and not artifacts of sampling variance or annotator subjectivity, we conducted a series of statistical validation experiments. Below we detail the methodology, results, and conclusions.

## D.1   BOOTSTRAP CONFIDENCE INTERVALS

We applied bootstrap resampling (1,000 iterations) over the collected pairwise preference votes. Elo scores were recomputed in each resample, and we report bias-corrected and accelerated (BCa) 95% confidence intervals. These intervals, visualized in Figure 2b, quantify uncertainty in model rankings. Notably, top-ranked models (e.g., Claude Sonnet 4 vs Claude 3.7 Sonnet) maintain non-overlapping intervals, supporting the reliability of observed differences.

## D.2   PERMUTATION TESTS AND EFFECT SIZES

To evaluate whether observed pairwise win-rate differences could arise by chance, we performed permutation tests with 5,000 random shuffles of preference labels. This generated a null distribution of Kendall's $\tau$ correlations and pairwise win rates. The observed global ranking achieved $\tau = 1.0$ compared to a null mean of 0.004 (std 0.276), yielding $p = 2 \times 10^{-4}$. In addition, we conducted

two-sided $z$-tests on all model pairs using bootstrap-estimated Elo and standard errors. After Holm–Bonferroni correction, 95% comparisons remained significant at $\alpha = 0.05$. We also report effect sizes using Cohen's $d$, which consistently indicated medium-to-large effects in the most critical comparisons (e.g., Claude 3.7 Sonnet vs GPT-4.1).

### D.3 POWER ANALYSIS

We performed a power analysis to assess whether the collected number of votes is sufficient to detect meaningful effects. Assuming a conservative medium effect size ($\Delta$ Elo = 50, corresponding to 57% win rate), our dataset with $n = 1,661$ votes provides $\approx 0.90$ power to detect differences at $\alpha = 0.05$. This analysis confirms that the study is well-powered for medium effects, though smaller differences between closely-ranked models may require additional votes.

### D.4 INTER-ANNOTATOR AGREEMENT (IAA)

To quantify annotation reliability, we randomly sampled 100 trajectories and had them independently labeled by three annotators for multiple attributes: preferences, correctness, safety, and efficiency. Agreement was measured using Krippendorff's $\alpha$:

- Preferences: $\alpha = 0.72$
- Correctness: $\alpha = 0.78$
- Safety: $\alpha = 0.68$
- Efficiency: $\alpha = 0.70$

These results indicate moderate-to-strong agreement even for open-ended GUI tasks, validating the consistency of human judgments.

### D.5 NOISE SENSITIVITY ANALYSIS

To examine robustness under noisy annotations, we randomly perturbed 10–30% of votes and recomputed Elo scores. Kendall's $\tau$ correlation with the original ranking remained high: $\tau = 0.87$ (10% noise), $\tau = 0.76$ (20%), and $\tau = 0.61$ (30%). This suggests that rankings remain stable under moderate subjectivity, though extreme noise can affect closely ranked models.

### D.6 STEP-LIMIT ABLATION STUDY

Since agent trajectories are capped at 50 steps, we investigated whether this limit biases rankings against long-horizon strategies. We truncated 100 pairwise trajectories to the first 15 steps and recollected votes from three annotators. The recomputed Elo scores correlated with the original ranking at $\rho = 0.71$ (Spearman), indicating moderate sensitivity to trajectory depth. This suggests that step limits may underrepresent agents with strong long-term planning ability.

### D.7 CONCLUSION

Together, these analyses provide convergent evidence that our reported ranking differences are statistically significant, robust to sampling variance, and supported by consistent human judgments. While sensitivity analyses highlight areas for future work—particularly around closely ranked models and long-horizon tasks—the overall findings are reliable within the current evaluation design.

## E ANALYSIS DETAILS

### E.1 SETUP DISTRIBUTION ANALYSIS.

To better understand how *real users* define computer-use tasks, we analyzed the distribution of initial environment setups selected on COMPUTER AGENT ARENA. The most frequently chosen setups involved Chrome browsers, LibreOffice applications, and custom file uploads. A closer inspection of Chrome-based sessions further reveals that users commonly visited domains related to web search

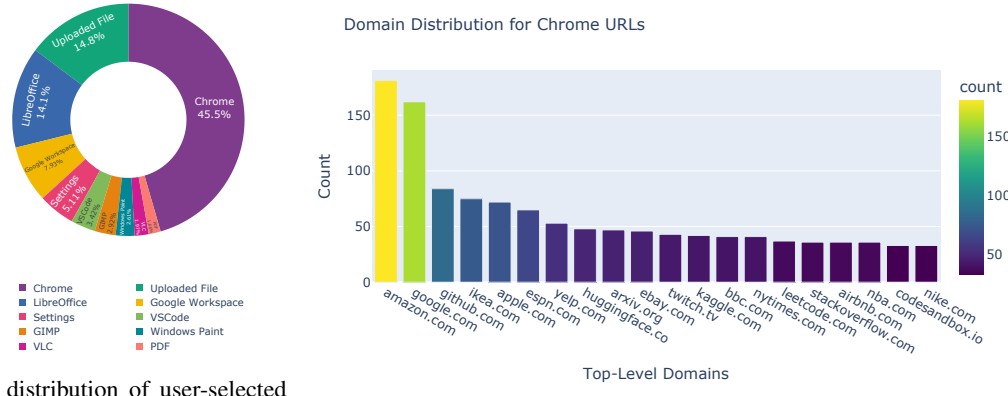

(a) distribution of user-selected environment setups.

(b) Top domain categories accessed under Chrome setups.

Figure 8: User behavior analysis on COMPUTER AGENT ARENA. (a) Environment setup distribution. (b) Domain categories in web browsing tasks.

(e.g., Google, Bing), e-commerce (e.g., Amazon, eBay), and technical resources (e.g., GitHub, Stack Overflow). These patterns indicate that real users tend to evaluate CUAs on utility-driven workflows—such as web browsing, information retrieval, and document editing—rather than on creative or leisure-oriented applications, which highlights a practical focus in current CUA expectations: users prioritize performance in high-frequency, productivity-centric tasks, while evaluation in casual or exploratory domains remains underexplored.

## E.2 GENERALIZATION ANALYSIS

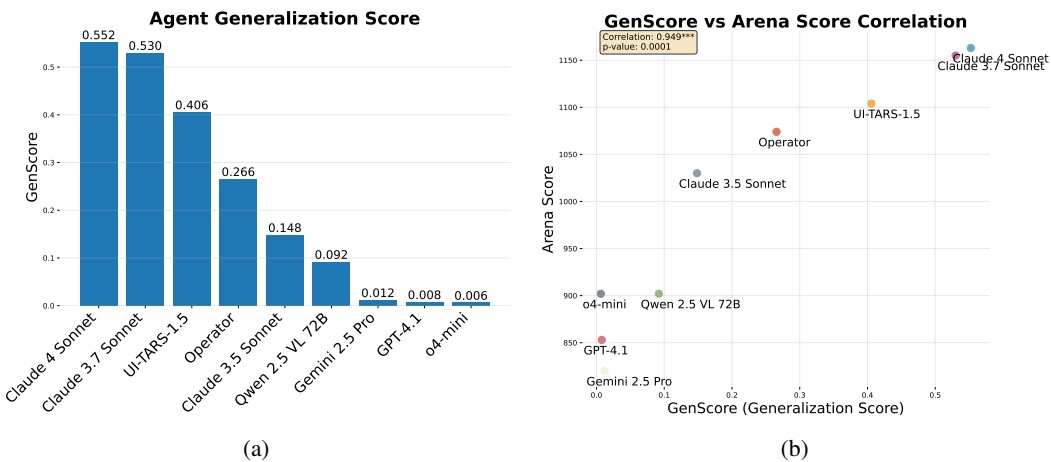

(a)

(b)

Figure 9: **Generalization across topics.** (a) Bar plot of *GenScore* for the top agents (sorted descending). A higher GenScore indicates both strong mean correctness and low variance across task categories. (b) Scatter of GenScore versus the arena-wide Elo-based *Arena Score*; we observe a strong correlation ($r = 0.949$, $p = 0.0001$), suggesting that success on the crowdsourced platform tightly tracks an agent's cross-domain generalization ability.

While Elo captures pairwise preferences, it does not tell us whether a CUA is *consistently* good across heterogeneous task types. Because COMPUTER AGENT ARENA crowdsources tasks from real users, the resulting distribution is highly diverse (Sec. 10a). An agent that overfits a few niches will underperform in the arena as a whole. We therefore introduce a *generalization score* (GenScore) that rewards both high accuracy and uniformity across topic categories.

Using the topic taxonomy derived by our clustering pipeline (Fig. 10a), we compute, for each agent $a$ and topic $t \in \mathcal{T}$, a correctness rate

$$r_{a,t} = \frac{\text{\# correct tasks of } a \text{ in } t}{\text{\# total tasks of } a \text{ in } t}.$$

**GenScore definition.** Let $\mathbf{r}_a = \{\hat{r}_{a,t}\}_{t \in \mathcal{T}}$ be the smoothed rate vector of agent $a$ across $K = |\mathcal{T}|$ topics. We compute:

$$\mu_a = \frac{1}{K} \sum_t \hat{r}_{a,t},$$

$$\sigma_a = \sqrt{\frac{1}{K} \sum_t (\hat{r}_{a,t} - \mu_a)^2}, \quad \text{CV}_a = \frac{\sigma_a}{\mu_a},$$

$$\text{HM}_a = \frac{K}{\sum_t 1/\hat{r}_{a,t}},$$

$$m_a = \min_t \hat{r}_{a,t}.$$

We then define

$$\text{GenScore}_a = \text{HM}_a \times \max\big(0,\, 1 - \text{CV}_a\big) \times \frac{m_a}{\mu_a}.$$

This multiplicative form (i) uses HM to punish low-performing topics, (ii) rewards uniformity through $(1 - \text{CV})$, and (iii) explicitly guards the worst-case via $m_a/\mu_a$.

Figure 9a shows GenScore for top agents; Fig. 9b visualizes its relationship with the platform-wide Elo-based Arena Score. We observe a strong Pearson correlation ($r = 0.949$, $p = 0.0001$) between GenScore and Arena Score. This suggests that success on COMPUTER AGENT ARENA is tightly linked to an agent's ability to generalize across many real-world task types, rather than excelling in a small subset. Concretely, models such as *Claude 4 Sonnet* and *Claude 3.7 Sonnet* achieve both high mean correctness and low variance across topics, yielding the highest GenScores. In contrast, lower-ranked models show not only lower average correctness but also high dispersion—indicating pronounced weaknesses on certain categories (e.g., spreadsheet or document-editing tasks).

COMPUTER AGENT ARENA's crowdsourcing pipeline naturally yields a broad, utility-driven task mix, agents are implicitly required to be robust generalists. GenScore makes this requirement explicit and provides a compact diagnostic: models that score well on the platform also maintain balanced performance across categories. Going forward, we argue that reporting such generalization metrics—alongside global Elo—better reflects what users actually need from CUAs in the wild.

**CUA performance varies substantially by task category, with open-ended tasks amplifying model differences.** To examine task-specific performance, we cluster user-submitted instructions into six semantic categories using the Cilo pipeline (Tamkin et al., 2024). We then recompute Elo scores per category across all models (Figure 10), and observe substantial Elo margins depending on task type. Open-ended categories like *Information Retrieval* and *Content Creation* yield the largest performance gaps, Conversely, while more deterministic categories like *UI Navigation* and *OS Operations* yield tighter Elo differences. These findings underscore the importance of evaluating CUAs across a broad spectrum of task types. Uniform benchmark scores may obscure large model disparities that only emerge under complex, interactive, and underspecified conditions—precisely the kinds of tasks real users often submit.

### E.3 OSWORLD ABLATION STUDY

To examine the impact of task distribution on agent performance, we randomly sampled 1,000 user-submitted tasks from COMPUTER AGENT ARENA and used GPT-4o-mini to classify them according to semantic similarity to OSWorld. We prompted the model with a binary choice prompt comparing each task to a curated set of OSWorld exemplars. As shown in Table 2, only 20.7% of tasks were deemed in-domain, while the majority (79.3%) were classified as out-of-domain, highlighting the broader topical spread of real user inputs in COMPUTER AGENT ARENA.

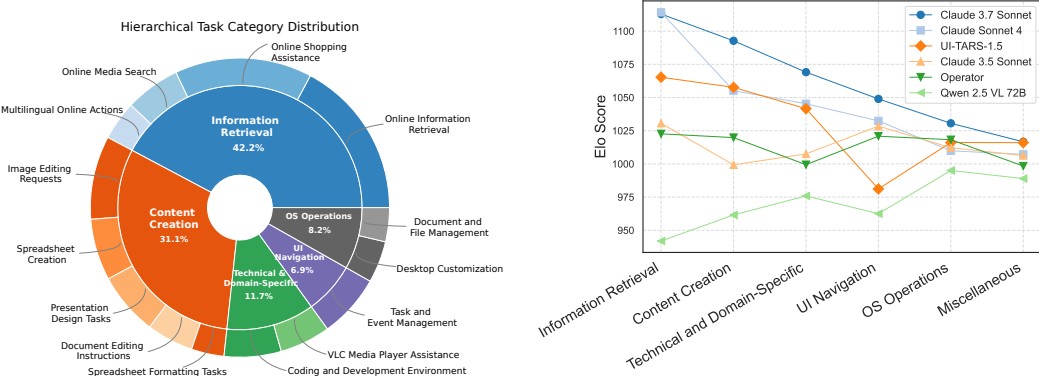

(a) Hierarchical topic clusters of user-submitted tasks.    (b) Elo scores of models across task categories.

Figure 10: Analysis of task taxonomy and model performance by category. Figure (a) shows a sunburst visualization of hierarchical task categories derived via topic modeling. Figure (b) presents model Elo scores grouped by task type, illustrating how task taxonomy amplifies performance differences across CUAs.

Table 2: Task distribution by semantic similarity to OSWorld (based on GPT-4o-mini classification).

| Category | Count | Percentage |
|---|---|---|
| In-Domain Tasks (similar to OSWorld) | 207 | 20.7% |
| Out-of-Domain Tasks (diverse / open-ended) | 793 | 79.3% |
| **Total** | 1,000 | 100.0% |

---

**OSWorld Ablation Study Classification Prompt**

```
Compare the following task with the set of tasks from OSWorld.

Task to analyze: "{task}"

Here is a whole set of OSWorld tasks:
{osworld_task_examples}

Rate the similarity of the task to analyze with the OSWorld tasks
    on a scale:
1 - Similar (the task is very similar to some of the tasks in
    OSWorld, maybe just a little different)
2 - Not similar (the task is different from tasks in OSWorld)

Provide only the number 1, 2 as your answer.
```

### E.4 TASK DISTRIBUTION ANALYSIS DETAILS

To compare task distributions across benchmarks, we sample task instructions from COMPUTER AGENT ARENA, OSWorld (Xie et al., 2024a), WebVoyager (He et al., 2024), and WebArena (Zhou et al., 2024). We embed all instructions using the text-embedding-3-small model and project them into a two-dimensional space via Principal Component Analysis (PCA) to visualize semantic coverage. This highlights the broader and less clustered semantic spread of real user instructions in COMPUTER AGENT ARENA.

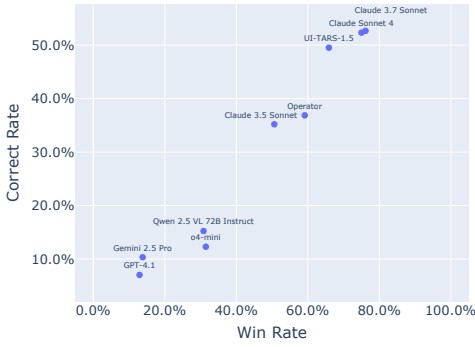
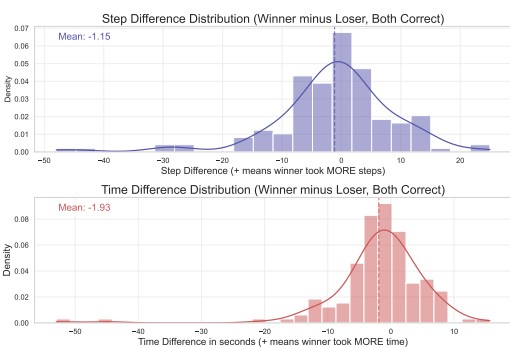

(a) Scatter plot of agent win rate and correctness rate.

(b) Distribution of step and latency differences for agent pairs with both marked correct.

Figure 11: User preference analysis based on agent behavior. Figure (a) shows a positive correlation between correctness rate and user win rate across agents. Figure (b) illustrates that differences in trajectory length and response latency do not significantly affect user preference in correctness controlled settings.

## E.5 INSTRUCTION AND TRAJECTORY METRIC COMPUTATION

To compare the complexity and ambiguity of user instructions across benchmarks, we compute four metrics uniformly for all tasks: instruction length, proportion of open-ended tasks, unigram perplexity, and average trajectory length.

**Instruction Length.**  We tokenize each task instruction using standard whitespace and punctuation rules and count the number of tokens. The results are aggregated to compute the average instruction length per benchmark.

**Open-Ended Task Detection.**  To estimate the proportion of open-ended or subjective tasks, we employ a GPT-4o-mini classifier using the following prompt:

---

**Open-ended Task Classification Prompt**

```
You are an expert at analyzing computer-use task instructions to
    determine if they are open-ended or subjective tasks.
A task is considered open-ended or subjective if:
1. It has multiple valid solutions or approaches
2. The success criteria are not strictly objective
3. It requires creative or subjective judgment
4. The outcome can vary based on personal interpretation
5. the answer cannot be evaluated by a simple rule-based judgement

Please analyze the given task instruction and respond in JSON
    format with:
{
    "is_open_ended": true/false,
    "confidence": 0-1,
    "explanation": "brief explanation of your reasoning"
}
```

---

For each task, we collect the binary open-endedness decision and compute the share of positive cases within each benchmark.

**Unigram Perplexity.** We use a smoothed unigram model to measure the lexical ambiguity of task instructions. The corpus-wide token distribution is first collected, and perplexity for each instruction is computed as:

$$\text{Perplexity}(x) = \exp\left(-\frac{1}{N}\sum_{i=1}^{N}\log\Pr(w_i)\right)$$

where $\Pr(w_i) = \frac{\text{count}(w_i)+1}{\text{total}+V}$, with $V$ denoting the vocabulary size. Higher perplexity indicates more unpredictable and context-dependent language, suggesting more ambiguous or under-specified input.

**Trajectory Length.** For COMPUTER AGENT ARENA, we compute the trajectory length of each agent-task interaction as the number of executed steps (actions). We include only trajectories marked as correct by human evaluators. For OSWorld and WebVoyager, we extract golden trajectory lengths as reported in their official benchmarks. WebArena is excluded from this comparison due to its lack of standardized trajectory ground-truth.

### E.6 SETUP DISTRIBUTION ANALYSIS.

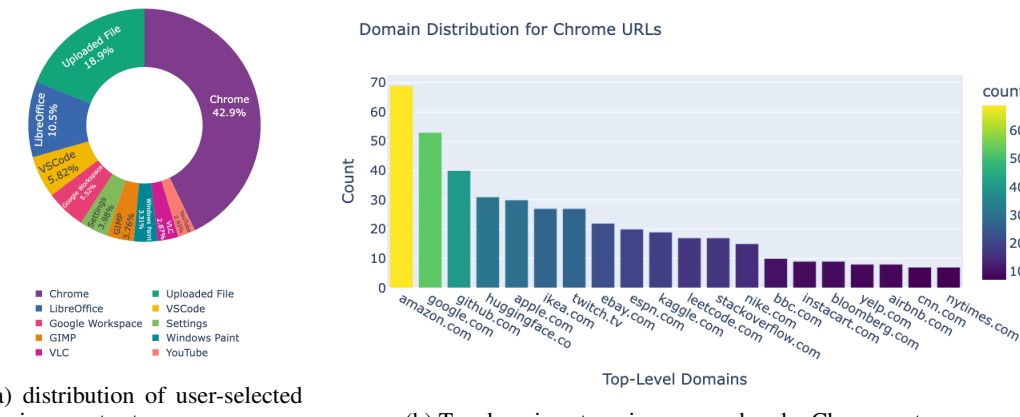

(a) distribution of user-selected environment setups.

(b) Top domain categories accessed under Chrome setups.

Figure 12: User behavior analysis on COMPUTER AGENT ARENA. (a) Environment setup distribution. (b) Domain categories in web browsing tasks.

To better understand how *real users* define computer-use tasks, we analyzed the distribution of initial environment setups selected on COMPUTER AGENT ARENA. The most frequently chosen setups involved Chrome browsers, LibreOffice applications, and custom file uploads. A closer inspection of Chrome-based sessions further reveals that users commonly visited domains related to web search (e.g., Google, Bing), e-commerce (e.g., Amazon, eBay), and technical resources (e.g., GitHub, Stack Overflow). These patterns indicate that real users tend to evaluate CUAs on utility-driven workflows—such as web browsing, information retrieval, and document editing—rather than on creative or leisure-oriented applications, which highlights a practical focus in current CUA expectations: users prioritize performance in high-frequency, productivity-centric tasks, while evaluation in casual or exploratory domains remains underexplored.

### E.7 TOPIC CLUSTERING DETAILS

To investigate how model performance varies by task type, we apply unsupervised topic modeling to group user-submitted instructions into semantic categories.

Following the Clio pipeline (Tamkin et al., 2024), we first encode all task instructions using Sentence-BERT embeddings. These embeddings are then projected using PCA for dimensionality reduction.

We perform initial coarse-grained clustering with KMeans, using the Elbow Method to select the optimal cluster count based on within-cluster variance. To refine results, we apply agglomerative hierarchical clustering over the KMeans centroids using Ward's linkage method, which enables the construction of a dendrogram for topic hierarchy visualization.

Each final cluster is then manually reviewed to assign interpretable labels. This process results in a six-category taxonomy:

- **Information Retrieval** (e.g., search, browse, summarize web content)
- **Content Creation** (e.g., writing, editing, preparing slides)
- **Technical and Domain-Specific Tasks** (e.g., code debugging, finance tools)
- **UI Navigation** (e.g., click, type, scroll workflows)
- **OS Operations** (e.g., file renaming, screenshot, task switching)

This hierarchical clustering approach provides a flexible way to capture semantic variation across real user instructions, allowing us to conduct category-specific evaluations and better understand the strengths and weaknesses of different CUAs across diverse task profiles.

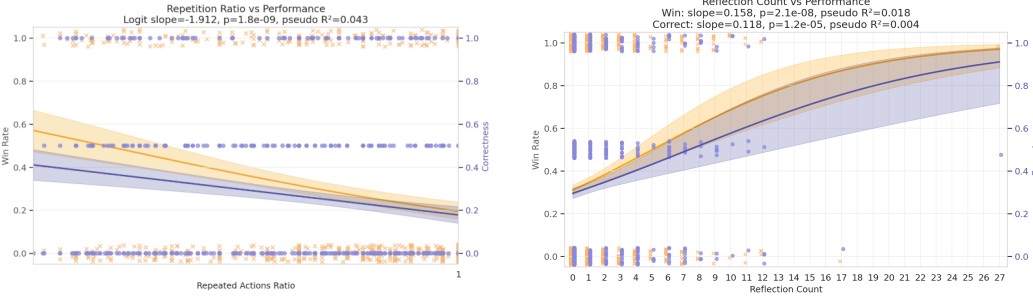

(a) Impact of repeated action ratio on performance.  (b) Impact of reflection frequency on performance.

Figure 13: Relationship between agent behavior patterns and performance metrics.

## F  AGENT BEHAVIOR ANALYSIS DETAILS

To better understand how users perceive different agent behaviors, we analyze two specific patterns observable from execution trajectories: **repetition** and **reflection**.

**Repetition Ratio.**  We define the repetition ratio as the proportion of actions within a trajectory that belong to contiguous segments of near-identical steps. Specifically, we scan the action sequence $(a_1, a_2, \ldots, a_T)$ and mark segments where three or more consecutive actions share high structural and parametric similarity—e.g., repeated scrolls or identical clicks. Repetition often reflects failure to adapt or unclear intent from the model.

**Reflection Frequency.**  We count the number of steps whose Chain-of-Thought (CoT) outputs contain introspective or self-corrective language (e.g., "think", "retry", "mistake", "adjust"). These steps are labeled as reflective, indicating the model is engaging in self-monitoring or explicit reasoning before issuing commands.

**Regression Analysis.**  For both behavioral metrics, we perform linear regression against model win rate and correctness. Repetition negatively correlates with user preference and correctness, confirming that users tend to penalize agents that exhibit stuck or redundant behavior. Conversely, reflection is positively associated with both outcomes, suggesting that agents that appear deliberate and thoughtful are better received, even if they do not fully solve the task.

This analysis complements the visual trends shown in Figure E.7 and supports the design implication that lightweight behavioral signals can serve as useful proxies for assessing agent robustness and user trust.

## G    SOFTWARE AND INFRASTRUCTURE RESOURCES

Our platform and experiments were built using a combination of open-source and proprietary software components. Below, we list the key software tools and dependencies used throughout the development and deployment of COMPUTER AGENT ARENA:

- **AWS EC2**: Hosts the virtual machines used for agent execution and screen recording.
- **AWS S3**: Stores large artifacts such as task logs, screen recordings, and agent-generated output traces.
- **OBS**: OBS software for recording agent trajectories.
- **Vision-Language Models (VLMs)**: Accessed via OpenAI, Google, Anthropic APIs; local inference used for selected open models (e.g., MiniGPT-4, LLaVA)
- **Frontend**: React (TypeScript), TailwindCSS for styling
- **Backend**: FastAPI + PostgreSQL for logging, orchestration, and user/session management
- **Data Analysis and Visualization**: Python (NumPy, pandas, matplotlib, seaborn, scikit-learn)

All custom components, orchestration scripts, and evaluation UIs will be released as part of our open-sourced codebase to facilitate reproducibility and community adoption.

## H    REPRODUCIBILITY AND COST ANALYSIS

**Full Reproducibility Commitment.**    We release all assets under an open-source license, including:

- End-to-end platform code (frontend/backend, evaluation logic, ranking scripts)
- Pre-built Amazon Machine Images (AMIs) for both Windows and Ubuntu, ensuring consistent virtual machine environments
- A plug-and-play agent hub for integrating external models and enabling third-party leaderboard participation

**Evaluation Cost and Throughput.**    We provide a detailed breakdown of the per-evaluation cost:

- **Compute:** Each comparison runs two agents on `t3.medium` EC2 instances for approximately 15 minutes, costing about $0.02 USD per session.
- **API usage:** Using Claude 3.7 Sonnet as a reference, each session consumes about $0.72 USD.
- **Annotation:** Paid annotators are compensated at $0.50 USD per qualified evaluation.

Therefore, the total estimated cost per evaluation is approximately $1.24 USD.

**Summary.**    We believe COMPUTER AGENT ARENA delivers a reproducible and affordable evaluation pipeline with high community utility for benchmarking, training, and behavioral diagnostics.

## I    ADDITIONAL RESULTS

## J    CASE STUDY ANALYSIS

To complement our quantitative evaluation, we present a qualitative case study showcasing how real users interact with agents in complex, multi-step computer-use tasks. The example below illustrates a scenario where the user requests the agent to encrypt a PDF document using a specific password. This task requires not only navigating a real desktop interface but also verifying the success of encryption through file reopening.

Battle Count for Each Model

Figure 14: Number of votes for each CUA model.

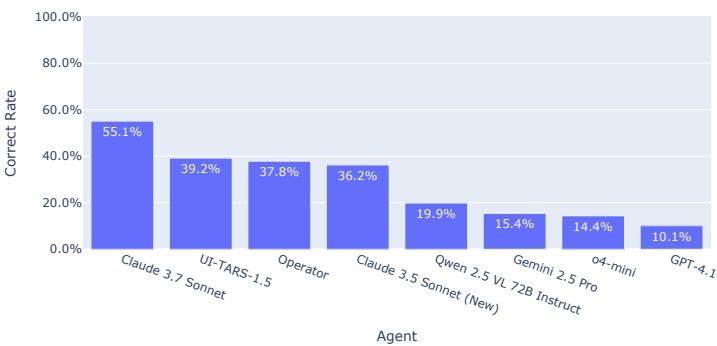

Figure 15: Aggregate correctness rate for each agent judged by users.

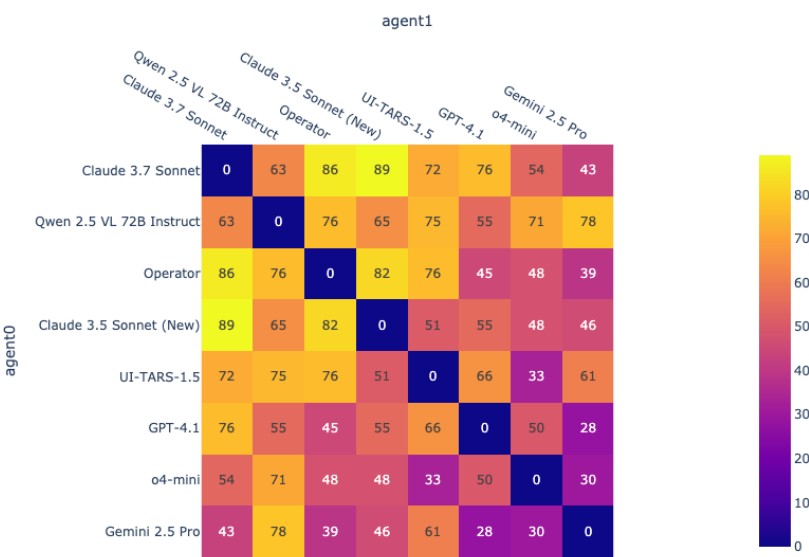

Figure 16: Pairwise battle count heatmap: each cell $(i, j)$ indicates the number of votes conducted between $\text{agent}_i$ and $\text{agent}_j$.

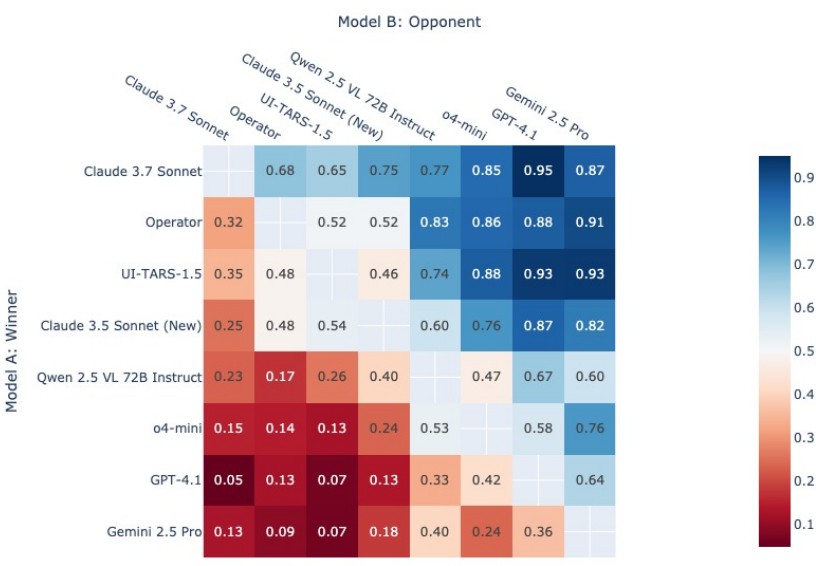

Figure 17: Pairwise win rate heatmap: each cell $(i, j)$ indicates the proportion of wins where Model A (row) outperformed Model B (column) based on user preferences.

Average Win Rate Against All Other Agents (Assuming Uniform Sampling and No Ties)

Figure 18: Average win rate of each agent against all others under uniform sampling and no ties.

The case highlights several key agent behaviors: interpreting user intent, identifying relevant UI elements within a real application (PDF-XChange Editor), and executing sequential steps with feedback loops (e.g., reopen to verify). Such real-world interactions are difficult to assess through scripted benchmarks but are essential for evaluating robustness and usability in practical settings.

This case exemplifies how our platform supports in-the-wild evaluation by embedding agent actions, user instructions, and full desktop context into a unified assessment flow.

> ### Example 1:
>
> Can you help me encrpyt this pdf file with password "jhsvcjvakj"? I want this contract file to keep a secret, please help me do that and PLEASE HELP ME VERIFY THE ENCRYPTION BY OPENING IT AGAIN.

| Agent 1: UI-TARS-1.5 (Lose) | Agent 2: Claude 3.7 Sonnet (Win) |
|---|---|
| **Agent 1: Partial Correct** | **Agent 2: Correct** |

**Step 1**

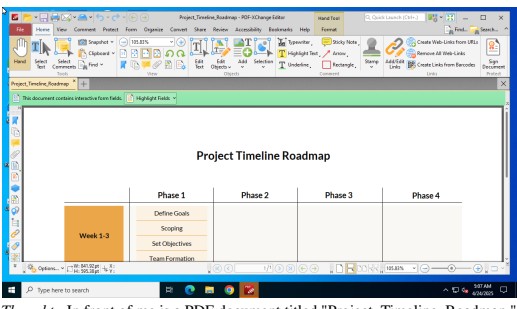

*Thought:* In front of me is a PDF document titled "Project_Timeline_Roadmap," and I need to password protect it. To begin, I should click on the File menu in the upper left corner, which is the first step in encrypting the file. From there, I will be able to find the options related to encryption settings.

**Action**
```
pyautogui.click(47.702, 55.385, button='left')
```

**Step 2**

**Step 1**

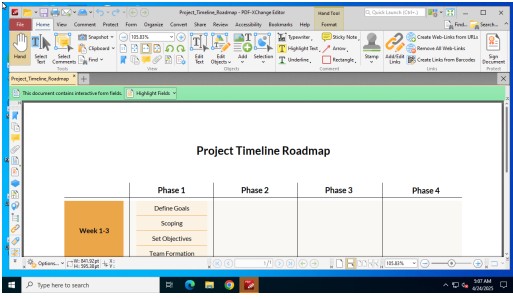

*Thought:* The user is asking me to encrypt a PDF file that appears to be a Project Timeline Roadmap with a password "jhsvcjvakj". They want me to verify the encryption works by reopening the file. Let me help with this task.

In PDF-XChange Editor, which is the application currently open, I need to: 1. Save the PDF with password encryption 2. Close the file 3. Reopen it to verify the encryption works

The encryption option should be available in the "Protect" menu or possibly under File > Save As with security options.

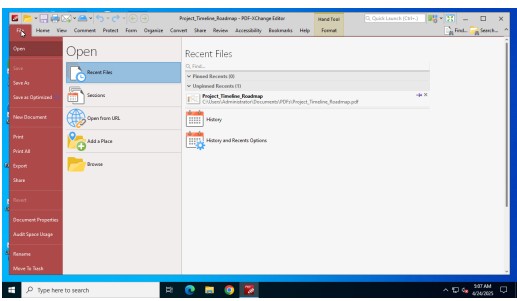

*Thought:* I just opened the File menu and I'm currently searching for the export option. Based on my experience, the encryption feature is usually tucked away within the export function, so I'm going to click on export in the left sidebar now.

**Action**
pyautogui.click(49.689, 393.626, button='left')

### Step 3

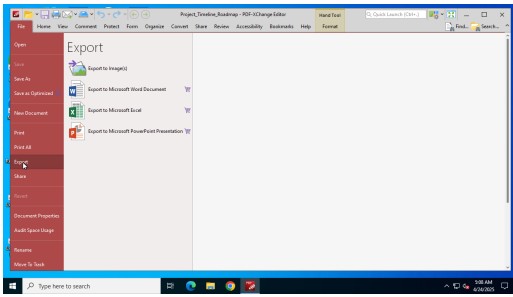

*Thought:* I took a close look at the contents of the export menu, and I realized that this option doesn't actually provide the PDF encryption feature we need. We need to rethink our approach, so I'll close the current export menu and continue searching for the correct encryption option in the File menu.

**Action**
pyautogui.click(626.087, 253.187, button='left')

### Step 4

(The screenshot is the same as the previous one.)

*Thought:* Let me summarize the current situation: I'm currently working in the PDF-XChange Editor, and I just spent some time looking for the encryption feature under the export menu, but it seems I was looking in the wrong place. Since I can't find the option in the File menu, I thought it might be a good idea to check the "Protect" button in the toolbar at the top of the interface. This button should help me find the functionality to add password protection to the PDF.

**Action**
pyautogui.click(269.317, 56.374, button='left')

### Step 5

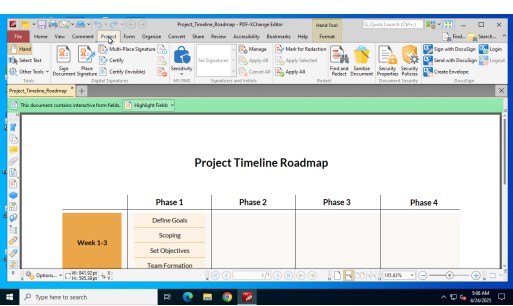

*Thought:* I looked through the File menu and didn't find the encryption option, but then I noticed a "Document Security Properties" button on the toolbar, which was exactly what I needed. Clicking on it opened the PDF encryption settings window, allowing me to continue configuring password protection for the document.

**Action**
pyautogui.click(964.969, 122.637, button='left')

### Step 6

I'll help you encrypt this PDF file with a password. I can see you have PDF-XChange Editor open with your Project Timeline Roadmap document. Let me guide you through the steps to encrypt it and then verify the encryption works. First, let's take a screenshot to see the current state:

**Action**
pyautogui.screenshot()

### Step 2

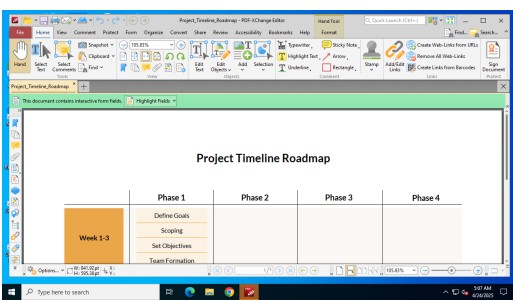

*Thought:* I notice there's a "Protect" tab in the menu bar. Let's click on that first to see the encryption options:

**Action**
pyautogui.click(270, 55, button='left')

### Step 3

*Thought:* Let's take another screenshot to see the protection options:

**Action**
pyautogui.screenshot()

### Step 4

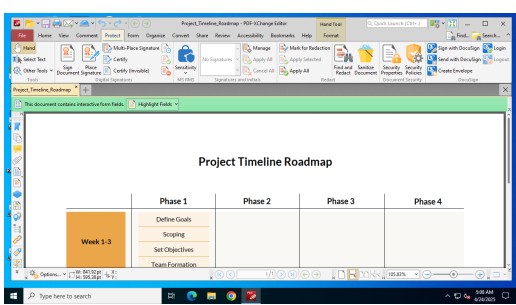

*Thought:* I see the Protect tab is now open, but I don't see a direct password encryption option in this ribbon. Let's try looking in the "Security Properties" section:

**Action**
pyautogui.click(963, 115, button='left')

### Step 5

*Thought:* Let's take another screenshot to see what options appear:

**Action**
pyautogui.screenshot()

### Step 6

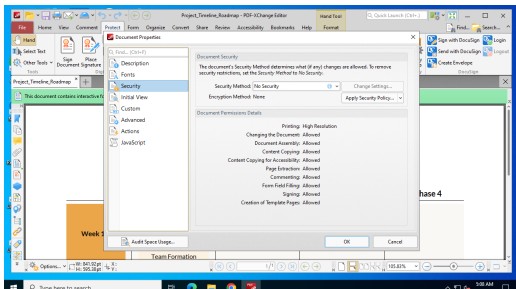

*Thought:* Great! I see we're in the Document Properties dialog box and "Security"

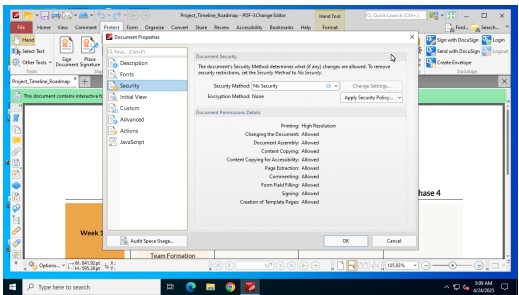

*Thought:* I can see the document properties window right now, and the crucial next step is to set the encryption method. To make this file more secure, I need to locate the dropdown menu labeled "Security Method" and then choose an appropriate encryption method from it. This way, I can effectively safeguard this important document.

**Action**
pyautogui.click(709.565, 200.769, button='left')

### Step 7

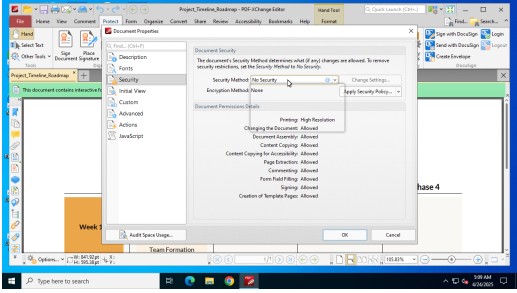

*Thought:* After going through all the setup menus, I've finally reached the crucial step of selecting the encryption method. In the dropdown list for Security Method, there are several encryption schemes available. Considering both security and compatibility, I believe the Password Security option is the most suitable choice, as it allows for precise control over access permissions. Next, I'll click on this option to proceed with setting up the password protection.

**Action**
pyautogui.click(636.025, 245.275, button='left')

### Step 8

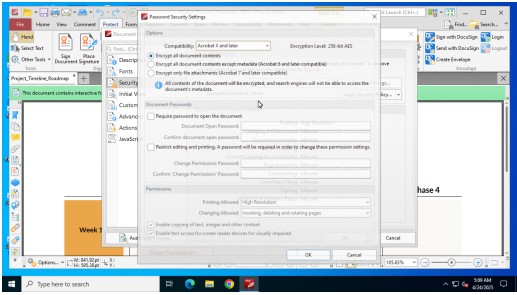

*Thought:* I noticed that the password security settings window is now open. I need to fill in the password "jhsvcjvakj" in the "Require password to open the document" section. This password is a crucial step for protecting the document, so it's important that I input it accurately. After that, I will also need to confirm the password again, as this will ensure that the document's encryption security is properly established.

**Action**
pyautogui.click(590.311, 333.297, button='left')

### Step 9

(The screenshot is the same as the previous one.)

*Thought:* I am currently in the process of setting a password, and the interface is waiting for me to input the password "jhsvcjvakj." The password box labeled "Require password to open the document" seems to be specifically designed for protecting document access. I am focusing on entering this password accurately, as it is crucial for the security of the document.

is already selected in the left panel. Now let's click on "Change Settings..." to set up password encryption:

**Action**
pyautogui.click(917, 200, button='left')

### Step 7

*Thought:* Let's see what security options we have now:

**Action**
pyautogui.screenshot()

### Step 8

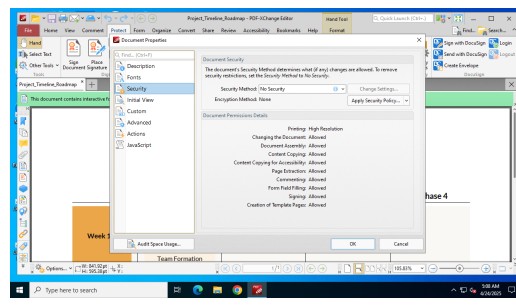

*Thought:* I notice that even though I clicked "Change Settings...", the dialog didn't change. Let's try clicking on the dropdown next to "Security Method:" where it currently says "No Security":

**Action**
pyautogui.click(827, 200, button='left')

### Step 9

*Thought:* Let's see what security options are available:

**Action**
pyautogui.screenshot()

### Step 10

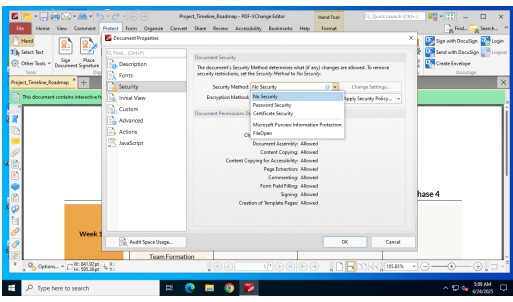

*Thought:* Perfect! I can see the dropdown menu with security options. Let's select "Password Security":

**Action**
pyautogui.click(671, 245, button='left')

### Step 11

*Thought:* Now let's see if we get a password dialog:

**Action**
pyautogui.screenshot()

### Step 12

**Action**

pyautogui.click(453.168, 392.637, button='left')

## Step 10

(The screenshot is the same as the previous one.)

*Thought:* It seems that I need to enter the password in the correct order. I noticed that the first password input box labeled "Document Open Password" is empty. I should click on this box and type in "jhsvcjvakj" as the document opening password. Once I've completed this step, I'll be able to move on to setting the permission password.

**Action**

pyautogui.click(476.025, 389.67, button='left')

## Step 11

(The screenshot is the same as the previous one.)

*Thought:* I just accidentally clicked in the wrong spot, and that's not good at all. Let me start over - I need to set the password in the "Document Open Password" box. This password will be used to open the document, so I should enter "jhsvcjvakj" in there. I need to follow the correct order, and once I finish this step, I can move on to setting the permission password.

**Action**

pyautogui.click(664.845, 392.637, button='left')

## Step 12

(The screenshot is the same as the previous one.)

*Thought:* It seems like there's a little hiccup in the password setup process. I'm staring at the empty input box on the screen, which is supposed to be for the document's opening password, but the system isn't responding at all. This might be due to some lag in the interface. I need to click on the "Document Open Password" input box again to make sure it's activated so I can continue entering the password.

**Action**

pyautogui.click(700.621, 309.56, button='left')

## Step 13

(The screenshot is the same as the previous one.)

*Thought:* The password input box is finally set up and ready to go. Now, I need to enter the specified password "jhsvcjvakj." It's important that I input it correctly since this is the first step in setting the password to protect the document.

**Action**

pyautogui.typewrite('jhsvcjvakj')

## Step 14

(The screenshot is the same as the previous one.)

*Thought:* Let me assess the situation here – I'm at the password security settings interface, and the input box is still greyed out and unresponsive. It seems that the system might be a bit slow, so I need to click on the "Document Open Password" input box again to activate it. Only then can I enter the password "jhsvcjvakj."

**Action**

pyautogui.click(663.851, 311.538, button='left')

## Step 15

(The screenshot is the same as the previous one.)

*Thought:* I've already entered the password input interface, and now I need to input "jhsvcjvakj" into the "Document Open Password" field. This step is crucial because this password will serve as the key to open the encrypted document later on. Let me focus on this task and make sure to enter each character accurately.

**Action**

pyautogui.typewrite('jhsvcjvakj')

## Step 16

(The screenshot is the same as the previous one.)

*Thought:* Hmm, something feels off here. The password box is right in front of me, but I've tried a few times and can't seem to get the password entered correctly. Upon closer inspection, it seems like the input field isn't fully activated. Let me click on the "Document Open Password" box once more to make sure it's in an input-ready state, and then I'll be able to enter the password "jhsvcjvakj" smoothly.

**Action**

pyautogui.click(663.851, 332.308, button='left')

## Step 17

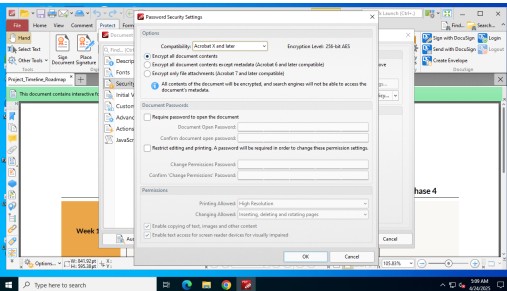

*Thought:* Excellent! Now we need to: 1. Check "Require password to open the document" 2. Enter the password "jhsvcjvakj" in the Document Open Password field 3. Confirm the password in the Confirm field Let's start by checking the "Require password" option:

**Action**

pyautogui.click(367, 282, button='left')

## Step 13

*Thought:* Now let's click in the Document Open Password field:

**Action**

pyautogui.click(755, 307, button='left')

## Step 14

*Thought:* Let's type the password "jhsvcjvakj":

**Action**

pyautogui.write("jhsvcjvakj")

## Step 15

*Thought:* Now let's click in the Confirm password field:

**Action**

pyautogui.click(755, 335, button='left')

## Step 16

*Thought:* And type the confirmation password:

**Action**

pyautogui.write("jhsvcjvakj")

## Step 17

*Thought:* Now let's click "OK" to apply the password security:

**Action**

pyautogui.click(762, 629, button='left')

## Step 18

*Thought:* Now let's click "OK" on the Document Properties dialog to apply all settings:

**Action**

pyautogui.click(851, 585, button='left')

## Step 19

*Thought:* Now that we've set the password security, let's save the document. Let's click on File menu:

**Action**

pyautogui.click(45, 55, button='left')

## Step 20

*Thought:* Let's take a screenshot to see the File menu options:

**Action**

pyautogui.screenshot()

## Step 21

(The screenshot is the same as the previous one.)

*Thought:* I found myself stuck in a cycle of repeatedly entering the password, and this was clearly pointless. Taking a step back, I realized that the current security settings window was quite cluttered and hard to navigate, making it difficult for me to operate effectively. Rather than wasting time here, it made more sense to close this window and reevaluate a clearer method for encrypting the PDF file. So, I decided to click the close button in the upper right corner and restart the entire encryption process.

**Action**
pyautogui.click(933.168, 33.626, button='left')

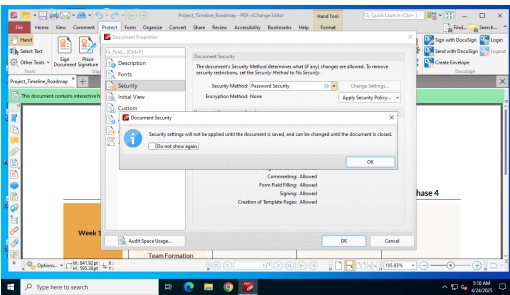

*Thought:* I see we have a notification that says "Security settings will not be applied until the document is saved." Let's click "OK" on this notification:

**Action**
pyautogui.click(917, 389, button='left')

### Step 18

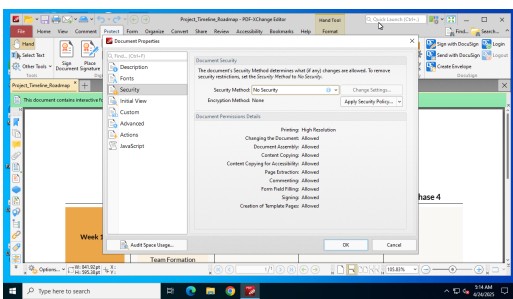

*Thought:* I just closed the previous window, and now I'm staring at the document properties interface. I took a quick glance at the dropdown menu labeled "Security Method," which should allow me to select the appropriate encryption option. After wasting time in the wrong place before, I want to make sure I don't choose incorrectly this time. So, I'll go ahead and click on that dropdown menu.

**Action**
pyautogui.click(709.565, 200.769, button='left')

### Step 19

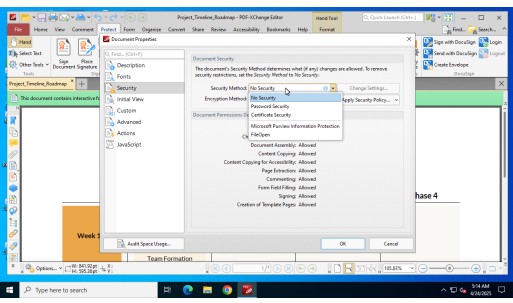

*Thought:* I just took a look around the properties window of the document and finally found the right place! In the dropdown menu under Security Method, there are several options, and now I need to select Password Security to add password protection to the PDF. This option is clearly visible, so I'm going to click on it right now to start configuring the encryption settings.

**Action**
pyautogui.click(677.764, 244.286, button='left')

### Step 20

### Step 22

*Thought:* Now let's click "OK" on the Document Properties dialog:

**Action**
pyautogui.click(851, 585, button='left')

### Step 23

*Thought:* Now let's try to save the document. Let's click on the File menu again:

**Action**
pyautogui.click(45, 55, button='left')

### Step 24

*Thought:* Let's take a screenshot to see the File menu:

**Action**
pyautogui.screenshot()

### Step 25

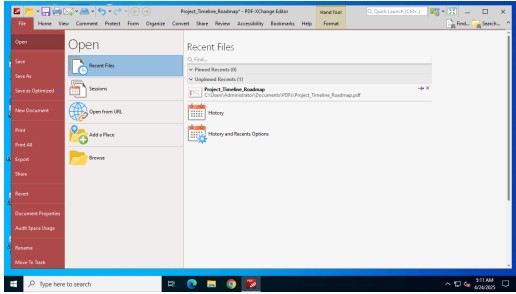

*Thought:* I see the File menu is open. Let's click on "Save" to save the document with the security settings:

**Action**
pyautogui.click(40, 149, button='left')

### Step 26

*Thought:* Let's take a screenshot to see if any confirmation dialog appears:

**Action**
pyautogui.screenshot()

### Step 27

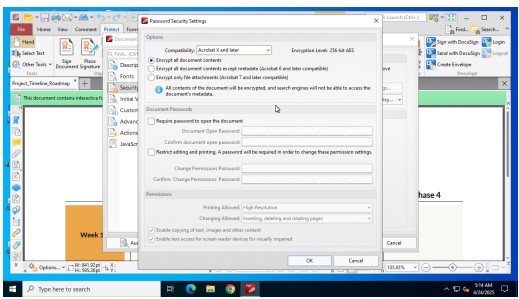 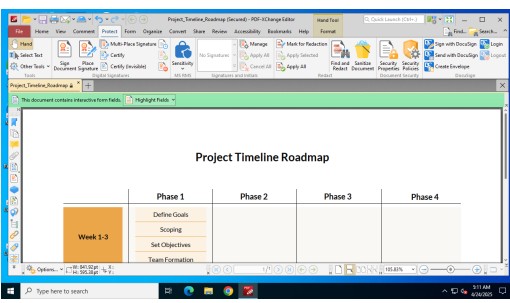

*Thought:* I'm now looking at the interface for password security settings, and it's time to enter the password "jhsvcjvakj." Let me first check the "Document Open Password" input box to make sure it's in a selected state, so that when I type on the keyboard, the characters go directly into the box. With this preparation, I'm confident that I won't encounter any issues like I did last time.

**Action**
pyautogui.click(700.621, 309.56, button='left')

### Step 21

(The screenshot is the same as the previous one.)

*Thought:* It looks like the password input field is all set up now. I'll go ahead and enter "jhsvcjvakj" as the opening password for the document, so I can continue with the encryption process. Since the previous steps have all been completed properly, this step should go smoothly as well.

**Action**
pyautogui.typewrite('jhsvcjvakj')

### Step 22

(The screenshot is the same as the previous one.)

*Thought:* After several attempts just now, I realized that simply retyping the password isn't working. I took a moment to calm down and analyze the situation: the input box is right in front of me, but it seems that the system hasn't registered my entries correctly. It looks like the issue is that the input box hasn't been activated properly. So, I should click on the "Document Open Password" input box first to make sure it's in a state ready for input, and then I can enter "jhsvcjvakj." This time, I need to be patient and make sure I get it right on the first try.

**Action**
pyautogui.click(663.851, 310.549, button='left')

### Step 23

(The screenshot is the same as the previous one.)

*Thought:* The password field is right in front of me, and after several attempts earlier, I've figured out the correct approach. Now, I need to enter the password "jhsvcjvakj" into the Document Open Password box, making sure to do it in one go. The system is just waiting for me to input it, so let me focus and complete this step accurately.

**Action**
pyautogui.typewrite('jhsvcjvakj')

### Step 24

(The screenshot is the same as the previous one.)

*Thought:* I have now arrived at the confirmation step for the password setup. It's a bit annoying that the password box is not responding. However, based on my previous experience, this issue is usually due to the input box not being properly activated. So, I need to click on the "Document Open Password" box to gain focus, which will allow me to enter the password. Once this step is completed, we can proceed with encrypting the document.

**Action**
pyautogui.click(364.72, 359.011, button='left')

### Step 25

(The screenshot is the same as the previous one.)

*Thought:* I see that the checkbox for restricting editing and printing is already checked, which is good. However, the main focus right now is the password input field, as the PDF encryption can't proceed without the password being set. Let me concentrate on the "Document Open Password" box and click on it so I can enter the password.

*Thought:* Great! I can see that the document has been saved and is now showing "(Secured)" in the title bar. This indicates that the password security has been successfully applied. Now, let's verify that the encryption works by closing and reopening the file. First, let's close the document by clicking the "X" on the document tab:

**Action**
pyautogui.click(192, 187, button='left')

### Step 28

*Thought:* Let's see if we need to confirm closing:

**Action**
pyautogui.screenshot()

### Step 29

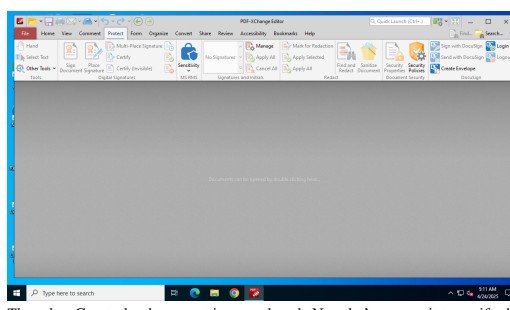

*Thought:* Great, the document is now closed. Now let's reopen it to verify the password protection works. Let's click on the File menu:

**Action**
pyautogui.click(45, 55, button='left')

### Step 30

*Thought:* Let's take a screenshot to see the File menu:

**Action**
pyautogui.screenshot()

### Step 31

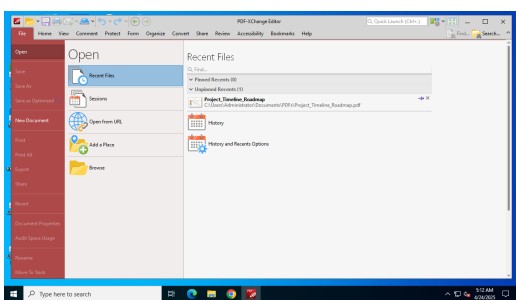

*Thought:* Perfect! I can see the document in the Recent Files list. Let's click on it to open it:

**Action**
pyautogui.click(673, 225, button='left')

### Step 32

*Thought:* Let's take a screenshot to see if a password prompt appears:

**Action**

pyautogui.click(663.851, 333.297, button='left')

## Step 26

(The screenshot is the same as the previous one.)

*Thought:* I'm currently setting up the confirmation password step, and the interface feels a bit sluggish. However, this reminds me that the key to encrypting a PDF lies in these two password inputs. Even though the system is responding slowly, I need to stay patient and complete the entry of "jhsvcjvakj" in the confirmation password field. Once this step is finished, the entire encryption process will be nearly complete.

**Action**

pyautogui.typewrite('jhsvcjvakj')

## Step 27

(The screenshot is the same as the previous one.)

*Thought:* Hmm, let me analyze the current situation. I'm stuck in the PDF encryption settings interface, where there's a password input box that I just can't seem to get right. I've realized that I've been stuck in a vicious cycle—just mindlessly repeating the input, but never actually getting anywhere. Now I've come to understand the problem: the input box is there, but it hasn't been activated properly, which is why my input isn't being registered. So here's what I'll do: I'll click on the "Document Open Password" box first to make sure it's ready to receive input, and then I'll enter "jhsvcjvakj" accurately. Sometimes, it really pays to be a bit patient and tackle problems step by step.

**Action**

pyautogui.click(685.714, 331.319, button='left')

## Step 28

(The screenshot is the same as the previous one.)

*Thought:* I've come to realize that I've been repeating the same mistake over and over again. Each time, I simply type in the password without giving much thought to the actual issue at hand. The problem is quite simple: the input box isn't activated! It's like talking to thin air; no matter how many times I try, it's all in vain. Right now, I need to pause for a moment and make sure to activate the "Document Open Password" input box properly. Only then can we really begin the password entry process. Let's take it step by step and first resolve the issue of the input box.

**Action**

pyautogui.click(663.851, 310.549, button='left')

## Step 29

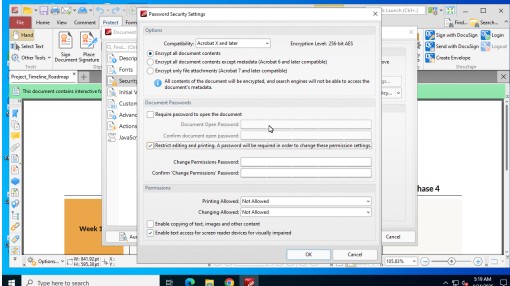

*Thought:* The previous steps were a bit chaotic, and after analyzing the situation, it seems I've fallen into a trap. I've been repeating the password entry process without addressing the root issue. Looking back, I realized that I never properly clicked on the "Document Open Password" input box. Now, the password security settings window is still open, and I need to click on the input box to activate it first. Only then will I be able to enter "jhsvcjvakj" as the password to enable the document encryption feature.

**Action**

pyautogui.typewrite('jhsvcjvakj')

---

### Example 2:

Due to work requirements, the following changes need to be made to the document. All "challenges" are changed to "difficulties", and the changed text is highlighted in red for my convenience to view.

Agent 1: o3 (Lose)

**Action**

pyautogui.screenshot()

## Step 33

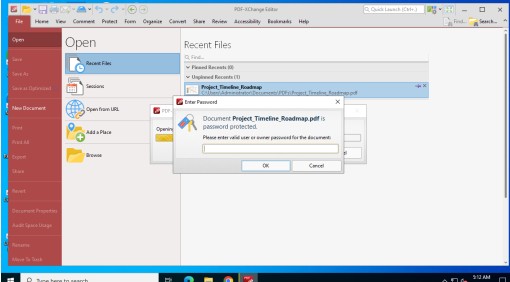

*Thought:* Perfect! We can see that the document is now password protected - there's a prompt asking for the password. Let's enter the password we set ("jhsvcjvakj"):

**Action**

pyautogui.click(670, 371, button='left')
pyautogui.write("jhsvcjvakj")

## Step 34

*Thought:* Now let's click "OK" to submit the password:

**Action**

pyautogui.click(658, 413, button='left')

## Step 35

*Thought:* Let's take a screenshot to see if the document opens:

**Action**

pyautogui.screenshot()

## Step 36

*Thought:* Perfect! The document has successfully opened after entering the correct password. We can see that it still shows "(Secured)" in the title bar, confirming that the password protection is working properly.

**Action**

Summary of What We Did:
1. We successfully added password protection to your Project Timeline Roadmap PDF with the password "jhsvcjvakj" 2. We saved the document with the security settings applied 3. We verified that the protection works by: - Closing the document - Reopening it, which prompted for a password - Successfully entering the password to access the protected document

Your PDF is now encrypted and can only be opened by someone who knows the password "jhsvcjvakj". The encryption uses 256-bit AES security, which is a strong encryption standard. Anyone trying to open this document will need to enter the correct password or they won't be able to view the contents.

Is there anything else you'd like me to help you with regarding this document or its security?

Agent 1: OpenAI Computer-Use Preview (Win)

## Agent 1: Partial Correct

### Step 1

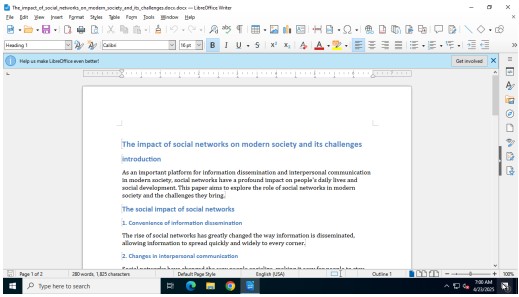

*Thought:* Planning find/replace task The user wants to replace all occurrences of "challenges" with "difficulties" in a LibreOffice Writer document, highlighting the changes in red. I'm thinking I'll use the Find & Replace dialog (Ctrl+H) and set it up: find "challenges," replace with "difficulties," and format the replacement text as red. Alternatively, I could find "difficulties" afterward and style it red. For now, I'll open Find & Replace and proceed from there. It sounds straightforward!

**Action**
import pyautogui import time pyautogui.keyDown('ctrl') pyautogui.keyDown('h') pyautogui.keyUp('h') pyautogui.keyUp('ctrl')

### Step 2

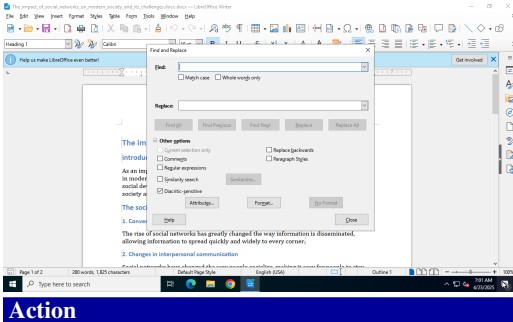

**Action**
import pyautogui import time pyautogui.typewrite('challenges', interval=0.01)

### Step 3

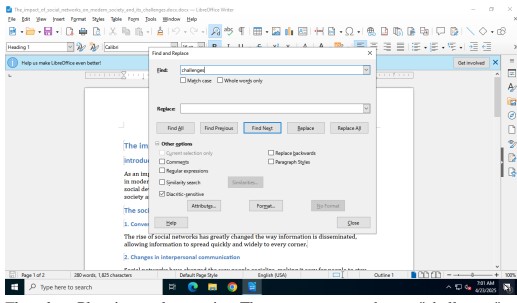

*Thought:* Planning replace action The user wants to change "challenges" to "difficulties" in LibreOffice Writer's Find and Replace dialog, then highlight the text in red. The immediate next step is to fill in the Replace field with "difficulties." I'll also open the Format option to set the text color to red, and then click "Replace All." Essentially, I'll need to click on the Replace entry box and type "difficulties," focusing on the approximate coordinates within the dialog. Let's proceed with that click action.

**Action**
import pyautogui import time pyautogui.click(501, 254)

### Step 4

## Agent 1: Partial Correct

### Step 1

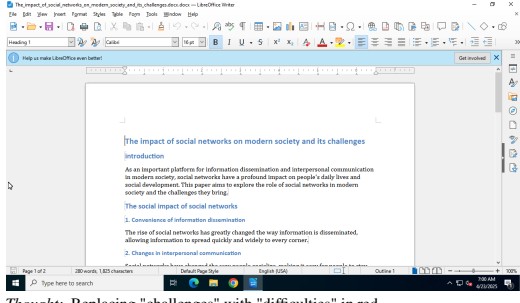

*Thought:* Replacing "challenges" with "difficulties" in red

**Action**
pyautogui.hotkey('ctrl', 'h')

### Step 2

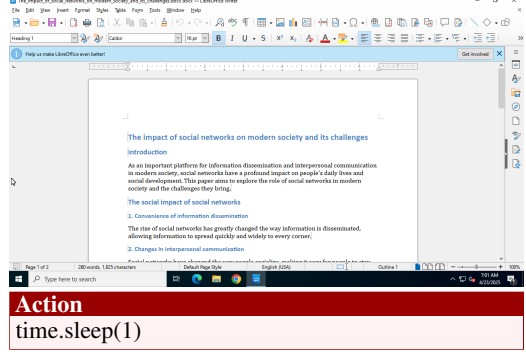

**Action**
time.sleep(1)

### Step 3

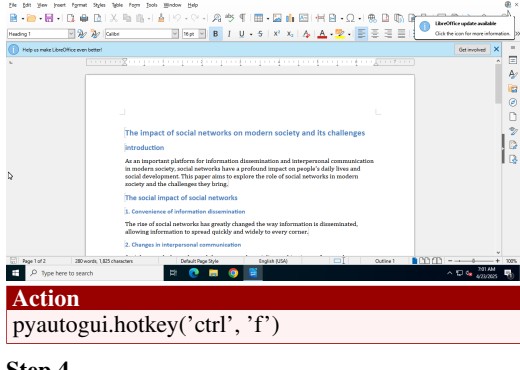

**Action**
pyautogui.hotkey('ctrl', 'f')

### Step 4

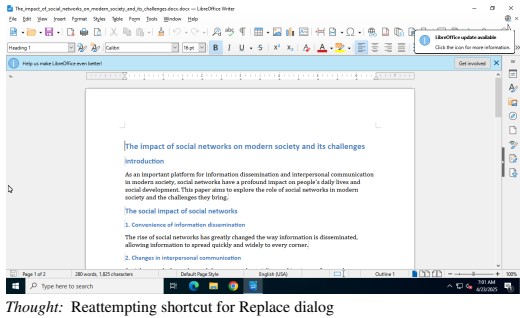

*Thought:* Reattempting shortcut for Replace dialog

**Action**
pyautogui.hotkey('ctrl', 'h')

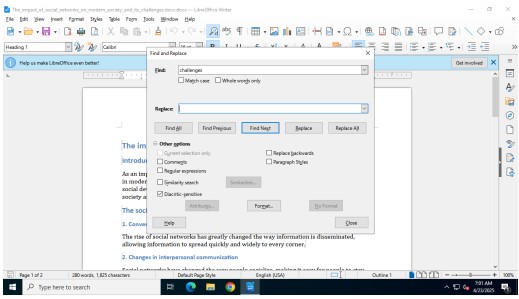

*Thought:* Executing find and replace I'm planning to replace all instances of "challenges" with "difficulties" and highlight the replacements in red. First, I'll use the Find & Replace dialog, ensuring "challenges" is in the Find field. For the Replace field, I'll type "difficulties." Next, I need to set the character color to red using the Format... button. After that, I'll click 'Replace All.' It's crucial to note it will only change "challenges" and not "challenge." Let's get started!Adjusting formatting settings In the screenshot, I noticed that the Format... button is not greyed out after all, so it's clickable! I'll start by typing "difficulties" into the Replace input. Then, I'll click the Format... button to choose Font Effects, where I'll change the Font color to red. After confirming, I'll hit Replace All to apply the changes. The cursor is visible in the Replace field, indicating it's active, so I can just go ahead and type "difficulties."

**Action**

import pyautogui import time pyautogui.typewrite('difficulties', interval=0.01)

## Step 5

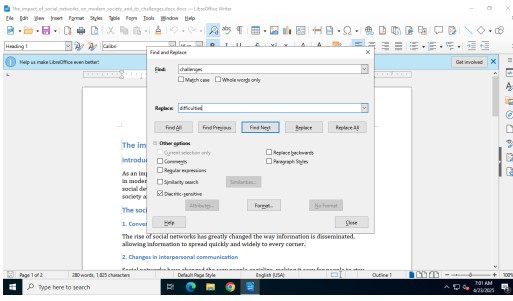

*Thought:* Checking formatting options I'm looking into how to use 'Search for Styles' and replace text with red font color. It seems like I need to change the replaced words to red instead of highlighting the background. I think I'll use the Find and Replace feature to set 'difficulties' in the 'Replace' field, then replace them all. After that, I can specify the font color red in formatting. Wait, the Format button for Replace looks greyed out. I need to clarify that for using the 'Format...' option.Planning text formatting I'm figuring out how to highlight changed text while replacing it. If the Format option is available, it makes sense to set it before doing the Replace All so I can do both simultaneously. The next step involves clicking "Format..." in the Replace field to set it to red. Looking at the screenshot, I'll need to estimate the coordinates for clicking the buttons. The Find/Replace window shows the format button around (620, 501), but I might need to adjust if it doesn't work right away. Let's stay flexible with the coordinates!

**Action**

import pyautogui import time pyautogui.moveTo(620, 500, duration=0.5)

## Step 6

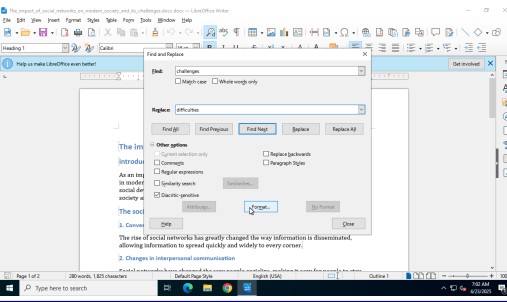

**Action**

import pyautogui import time pyautogui.click(640, 505)

## Step 7

## Step 5

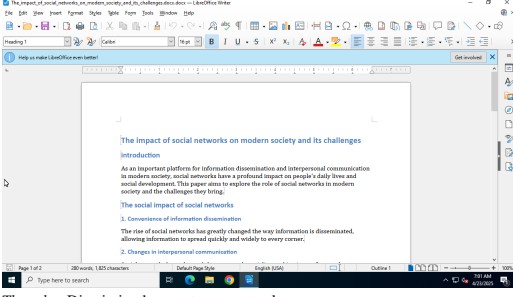

*Thought:* Dismissing banner to access tools

**Action**

pyautogui.click(1218, 143, button='left')

## Step 6

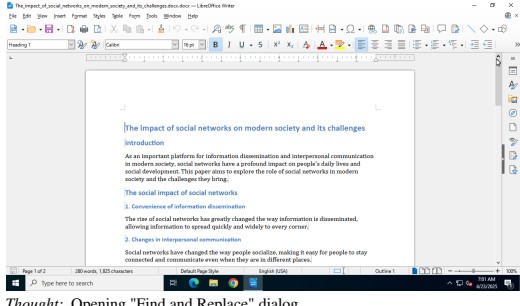

*Thought:* Opening "Find and Replace" dialog

**Action**

pyautogui.hotkey('ctrl', 'h')

## Step 7

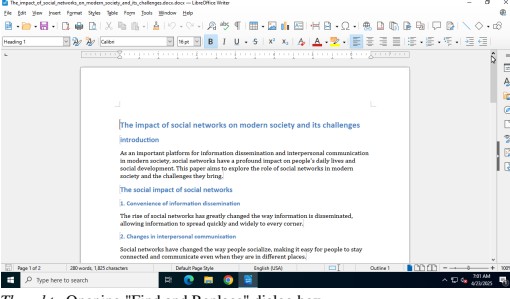

*Thought:* Opening "Find and Replace" dialog box

**Action**

pyautogui.click(57, 35, button='left')

## Step 8

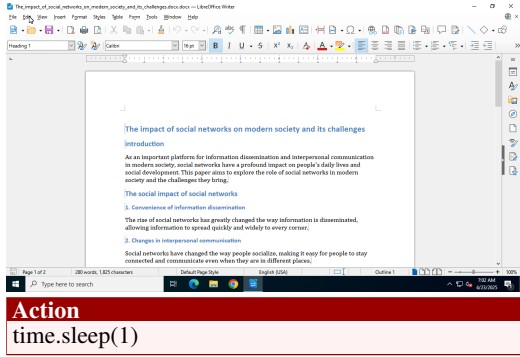

**Action**

time.sleep(1)

## Step 9

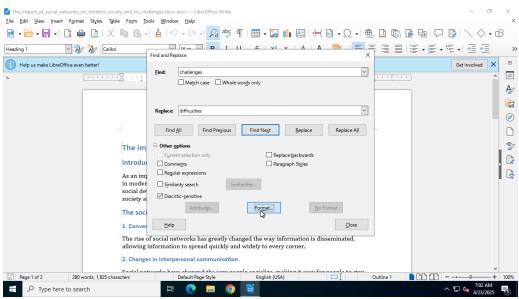

*Thought:* Executing find and replace The screenshot shows the 'Find and Replace' dialog in LibreOffice Writer. The user wants to change "challenges" to "difficulties" and highlight the changes in red. The Find and Replace fields are presumably set already. I need to configure the font color to red by clicking the 'Format' button. I'll approximate the button's coordinates in the dialog, which seems centered, around (618, 497), before clicking. Next, I'll proceed to click 'Replace All' after setting the format.

**Action**

import pyautogui import time pyautogui.click(617, 501)

**Step 8**

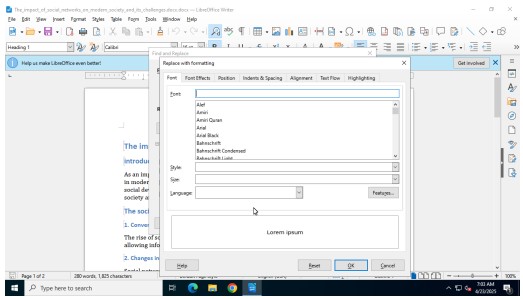

*Thought:* Adjusting Find & Replace settings It looks like I clicked on the wrong option when trying to open Find & Replace, ending up in a "Replace with formatting" dialog instead. My current goal is to replace all instances of "challenges" with "difficulties" and highlight them in red using LibreOffice Writer. I need to check if highlighting means changing the font color, and I'll set the font color to red in the format options. I'll set the "Replace with" field to "difficulties" and adjust the formatting accordingly.Setting formatting for replacement I have the "Replace with formatting" window open, which is helpful. Now, I need to set the Font Effects by choosing red for the font color. The next step is to click the "Font Effects" tab in this window. I'll navigate to the tab row at the top and click on the second tab. My mouse target will be around the "Font Effects" area, approximately at coordinates 450, 182. Let's proceed with that!

**Action**

import pyautogui import time pyautogui.moveTo(455, 181, duration=0.5)

**Step 9**

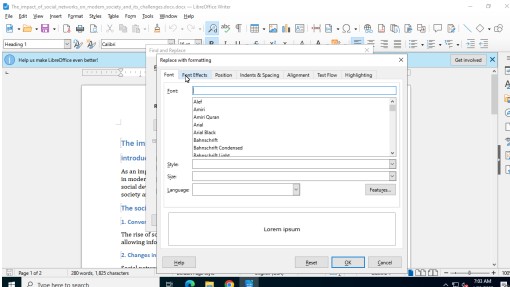

*Thought:* Adjusting text in LibreOffice To help the user replace "challenges" with "difficulties" and highlight those changes in red, I need to open the Find and Replace dialog with Ctrl+H. Right now, I see the formatting modal open, likely because the user double-clicked the "Format" button. First, I should close that modal to return to the main dialog, where I can set "Find" to "challenges" and "Replace" to "difficulties." I'll also check "Match case" and possibly "Whole words."Setting font color for changes To highlight the changed text in red, we need to adjust the Font Color under the "Replace" area. Right now, the Font Effects tab appears to be selected, but I'm seeing options that suggest the Font tab might be active instead. I need to check if the Font Effects tab is indeed active. The goal is to navigate to the Font Effects tab and set the Font Color to red. So, my next step is to click on the Font Effects tab.

**Action**

import pyautogui import time pyautogui.click(462, 181)

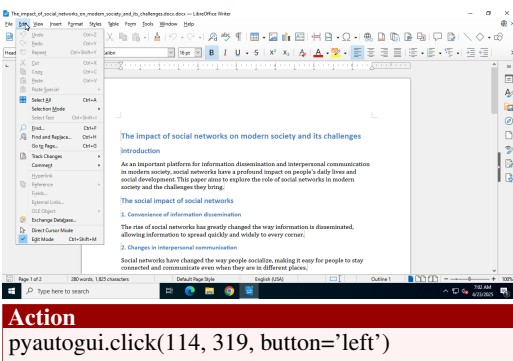

**Action**

pyautogui.click(114, 319, button='left')

**Step 10**

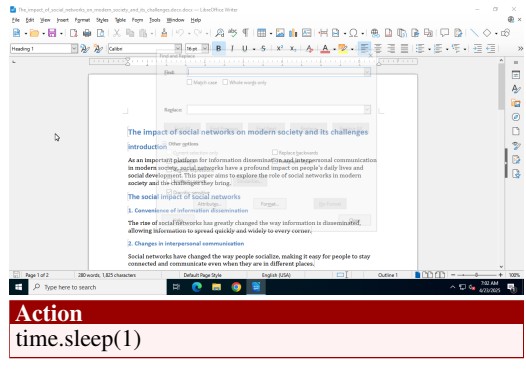

**Action**

time.sleep(1)

**Step 11**

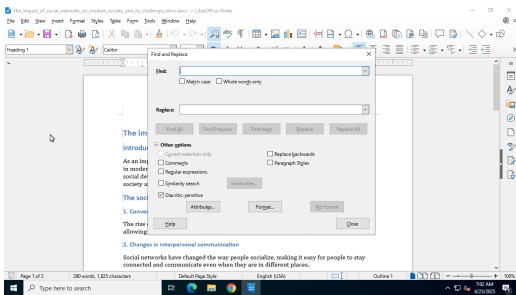

*Thought:* Inputting find and replace terms

**Action**

pyautogui.click(457, 159, button='left')

**Step 12**

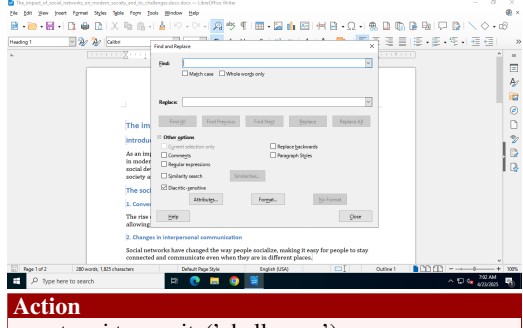

**Action**

pyautogui.typewrite('challenges')

**Step 13**

## Step 10

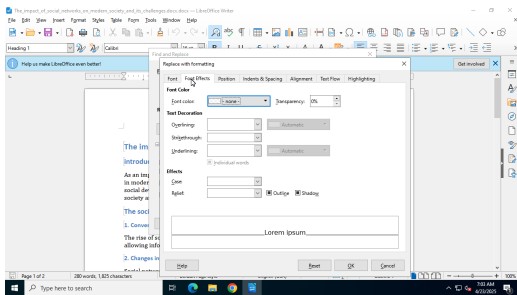

*Thought:* Planning text replacement in LibreOffice I need to help with the user's task of replacing all occurrences of "challenges" with "difficulties" in LibreOffice Writer and highlighting these changes in red. We've opened the Find & Replace dialog by pressing Ctrl+H, which shows the formatting details. The font color dropdown is currently set to none. My next step is to select red from that dropdown to ensure the replaced text appears in red. I need to carefully coordinate this action based on the screenshot's layout.Selecting font color in LibreOffice I'm estimating the font color dropdown's location roughly at coordinates (522, 238) on the screenshot. The arrow points to the top of the dropdown, which looks to be about 150 pixels wide. After this, I'll want to select the red color, which may require another click. So, the next step is to click that dropdown list based on the screenshot's resolution of 1280x720. The "Font color" field is almost centered, around 460 pixels wide. I'll coordinate my clicks accordingly.

**Action**

import pyautogui import time pyautogui.click(510, 238)

## Step 11

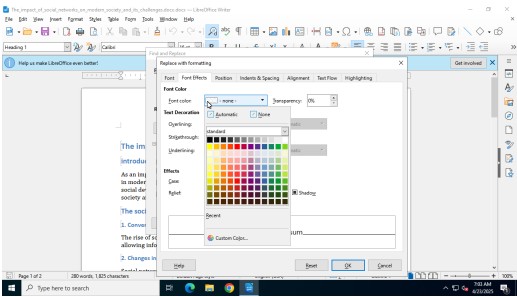

*Thought:* Inspecting grid coordinates I'm taking a look at the screenshot. In the grid, the third row looks to have yellowish colors, while the fourth has reds. I want to focus on a strong red cell, possibly around row 5, column 3. I'll estimate the coordinates, starting from the top left at about 514, 338. Since each cell is roughly 12x12 pixels, I calculate the coordinates for the red cell at row 3, column 6 to be (586, 374). Now, I'll plan the action to click at that point.

**Action**

import pyautogui import time pyautogui.click(586, 374)

## Step 12

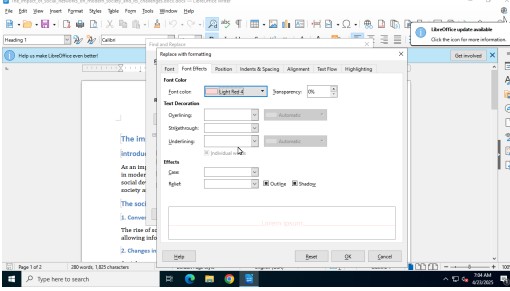

*Thought:* Replacing text in LibreOffice The user instructed me to replace "challenges" with "difficulties" and highlight the text in red. I've opened LibreOffice Writer and pressed Ctrl+H to access Find and Replace. It seems I'm in a "Replace with formatting" dialog, where I can customize the format for the replacement. The Font Effects tab shows Light Red 4, which is good for highlighting. I need to confirm this red color and return to set the Find and Replace fields accordingly.Confirming formatting changes I need to click the OK button in the bottom-right corner of the "Replace with formatting" window. It looks like there are buttons for "OK" and "Cancel." My cursor is probably in the Color field, so I just need to do a left click on OK to confirm the changes. This should allow me to finalize the formatting settings I'm adjusting. Let's get this done!

## Step 13

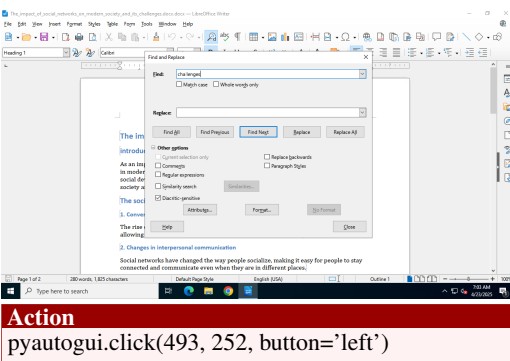

**Action**

pyautogui.click(493, 252, button='left')

## Step 14

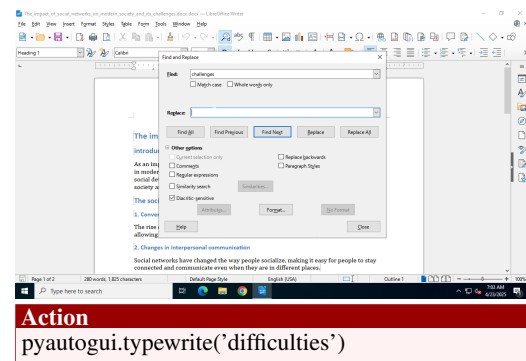

**Action**

pyautogui.typewrite('difficulties')

## Step 15

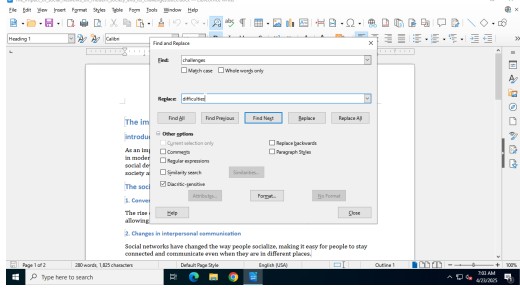

*Thought:* Clicking "Format" to set highlighting

**Action**

pyautogui.click(658, 497, button='left')

## Step 16

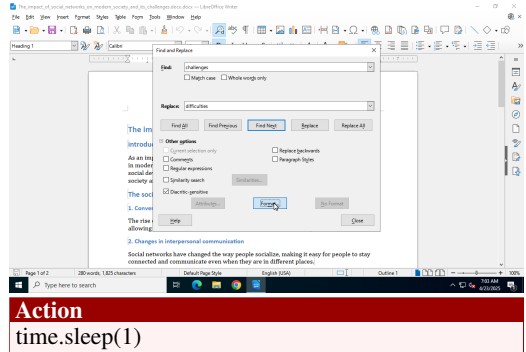

**Action**

time.sleep(1)

## Step 17

**Action**

import pyautogui import time pyautogui.click(859, 644)

## Step 13

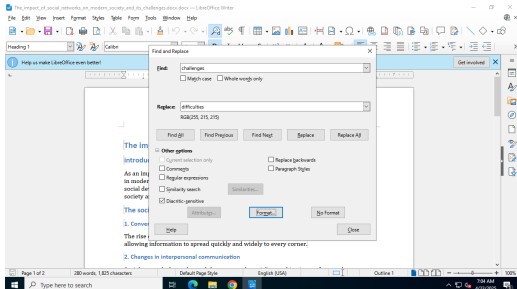

*Thought:* Setting up text replacement The user wants to replace every occurrence of "challenges" with "difficulties" and highlight the changes in red. I've opened the Find & Replace dialog and filled in the fields with the appropriate words. There's a "Format..." button in the dialog for setting the text color. I see it's highlighted but hasn't been configured yet. I need to click on this button to set the font color to red. After that, I can confirm the changes!Preparing to set formatting In the screenshot, I see the "Format..." button near the bottom right. To proceed, I need to click on this button, set the font color to red, and then confirm with OK before using "Replace All." I'm estimating the coordinates for the click since the screenshot size is 1280x720. The button seems to be around x 620, y 519 based on my estimation. The highlight indicates it's ready to be clicked, so I'll execute the left-click at those coordinates.

**Action**

import pyautogui import time pyautogui.click(631, 521)

## Step 14

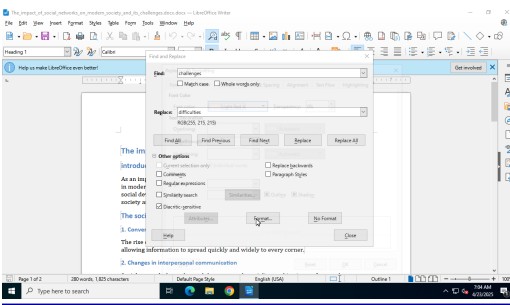

**Action**

import pyautogui import time pyautogui.click(630, 501)

## Step 15

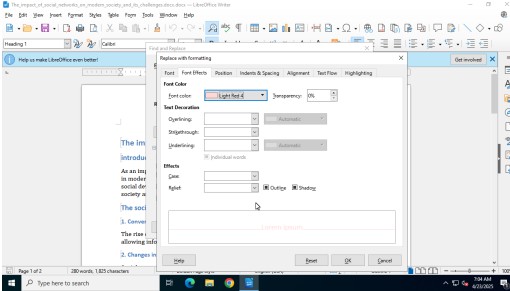

*Thought:* Executing text replacement I'm acting as a desktop automation agent, and the task is to change occurrences of "challenges" to "difficulties" while highlighting the changed text in red. When I triggered the Find & Replace dialog in LO Writer, I found that the "Replace with formatting" tab was open. Now, I need to specify "Find" and "Replace" fields and set the font color to Light Red 4. The next step is simply to click OK to confirm the formatting. After that, I can proceed to complete the replacement.Determining click coordinates I'm reviewing a screenshot that displays the "Replace with formatting" window, and I see the OK button located at the bottom right corner, around the coordinates x 853, y 646. I think this is a good approximation for the location since it looks reasonably accurate. My next step is to execute a left click at that position to confirm the formatting changes. It's a straightforward action, just following the visual cues from the screenshot.

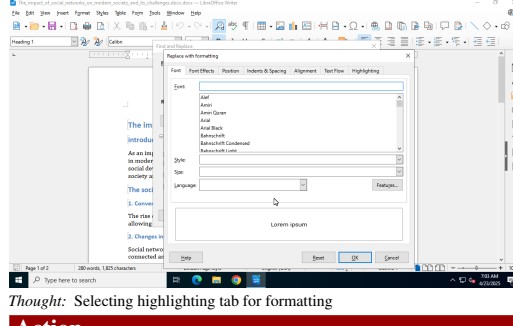

*Thought:* Selecting highlighting tab for formatting

**Action**

pyautogui.click(889, 182, button='left')

## Step 18

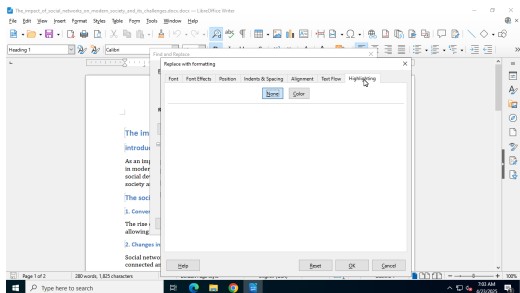

*Thought:* Selecting red color for highlighting

**Action**

pyautogui.click(733, 219, button='left')

## Step 19

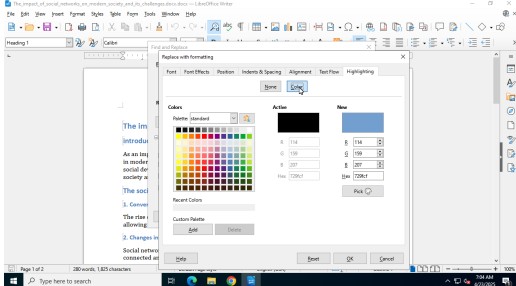

*Thought:* Selecting standard red color shade

**Action**

pyautogui.click(464, 344, button='left')

## Step 20

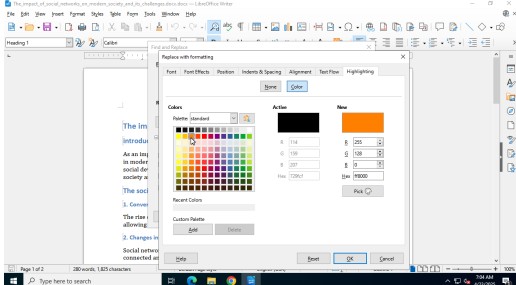

*Thought:* Correcting selection to red shade.

**Action**

pyautogui.click(473, 363, button='left')

## Step 21

**Action**

import pyautogui import time pyautogui.click(859, 645)

### Step 16

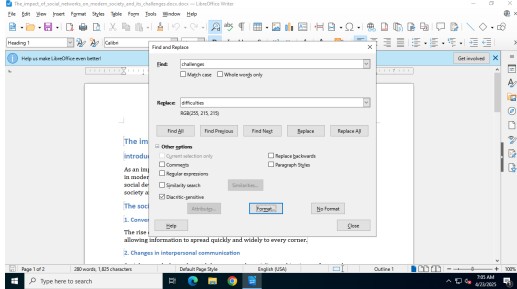

*Thought:* Planning formatting steps The user wants to know the next step after seeing a screenshot of the LibreOffice Find and Replace dialog. The task involves changing the font color of the replacement text to red. First, I'll click the Format button in the dialog. It should be located at the bottom center, around coordinates (628, 520) based on my estimate from the screenshot. After that, I'll navigate to Font Effects and choose the red color for the text. I'll also make sure to wait for the dialog to respond after the click.

**Action**

import pyautogui import time pyautogui.click(633, 520)

### Step 17

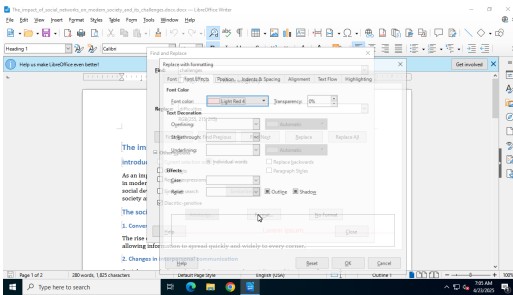

*Thought:* Clarifying Find and Replace Steps I see that we need to replace "challenges" with "difficulties" in LibreOffice Writer, and highlight the changes in red. First, I'll use the Find & Replace feature—put "challenges" in the Find box and "difficulties" in the Replace box. I should set the font color to light red in the formatting options. Once that's set, I can click "Replace All." Afterward, I'll close the character formatting dialog to complete the process.Confirming Formatting Changes I need to wrap this process up effectively. Instead of just hitting 'Enter,' it's safer to click 'OK' in the Format window since pressing 'Enter' might initiate an unwanted action. The OK button is located near the bottom center of the window, approximately at coordinates (850, 640). I'll confirm my click there, and once that window closes, I can proceed to click 'Replace All' in the Find & Replace tool. Let's finalize this plan!

**Action**

import pyautogui import time pyautogui.click(857, 641)

### Step 18

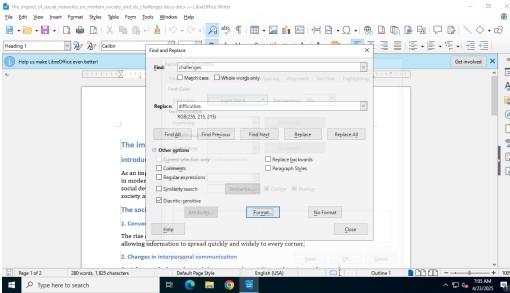

*Thought:* Approximating button coordinates I'm looking at the screenshot size of 1280x720, focusing on the dialog positioned around 361. The Format button is near the bottom right of the dialog. I'll estimate its coordinates: the button seems to be highlighted just above the bottom area. For the vertical positioning, the dialog top appears to be around y=112, while the Format button might be around y=480 to y=515 and x around 650. So, my next action is to left-click the Format button to apply the font formatting.

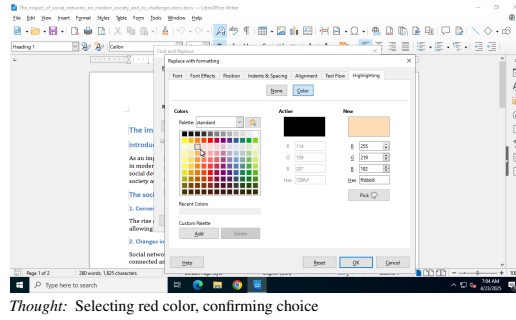

*Thought:* Selecting red color, confirming choice

**Action**

pyautogui.click(472, 335, button='left')

### Step 22

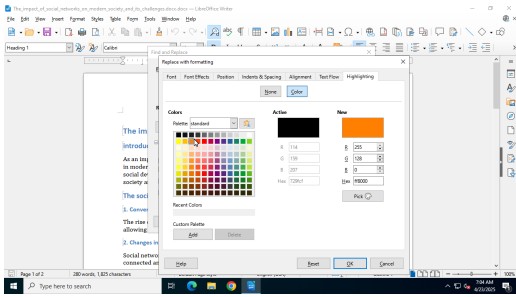

*Thought:* Changing color selection to red.

**Action**

pyautogui.click(483, 366, button='left')

### Step 23

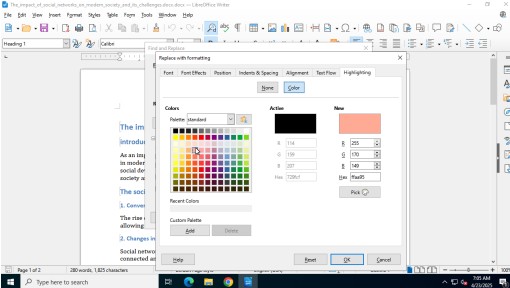

*Thought:* Selecting red paint instead of orange

**Action**

pyautogui.click(473, 327, button='left')

### Step 24

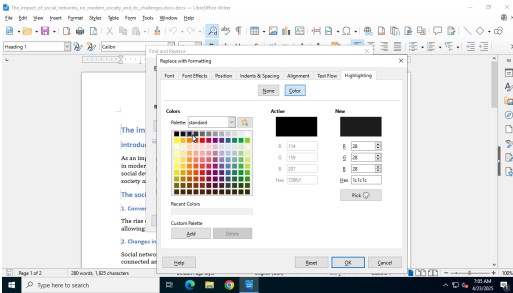

*Thought:* Selecting red shade, confirming choice

**Action**

pyautogui.click(500, 326, button='left')

### Step 25

**Action**

import pyautogui import time pyautogui.click(640, 512)

## Step 19

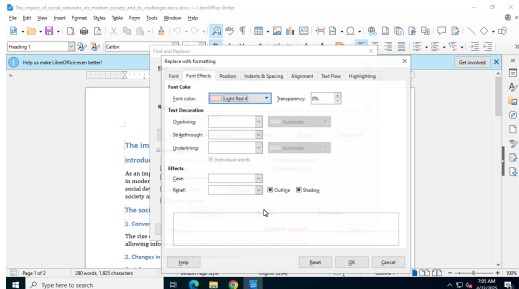

*Thought:* Analyzing text replacement process The user has a screenshot from LibreOffice Writer, focusing on the Find and Replace dialog, with the Font Effects tab showing a Light Red 4 font color. They're aiming to replace "challenges" with "difficulties" while highlighting the new text in red. The next steps involve confirming the formatting by clicking "OK" in the Replace with formatting window, then setting up the Find and Replace dialog to replace all instances. Checking the coordinates at the bottom right, I should approximate them within the given screenshot resolution.Confirming formatting action The Replace with formatting window is centered, and I see the OK button at the bottom right. My next step is clear: I'll click the OK button to confirm the changes. This will finalize the formatting for the text replacement. It's a straightforward action, but important for ensuring everything is set up correctly for the user's needs. Once I click OK, I can proceed to execute the replacement.

**Action**

import pyautogui import time pyautogui.click(859, 642)

## Step 20

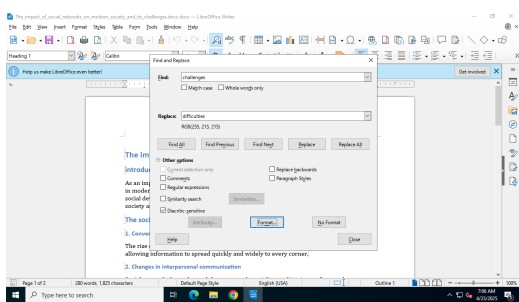

*Thought:* Executing text replacement in LibreOffice The user wants to change all instances of "challenges" to "difficulties" in their document, and they need that text highlighted in red. Since the document is already open in LibreOffice Writer, I see that the Find & Replace window is prepared with the correct terms. I need to click the Format button, choose the red font color in the Font Effects tab, and then hit "Replace All" to apply these changes. After that, I'll quickly check for the highlighted text.Choosing text formatting options I realize that the phrase "highlighted in red" could mean either font color or background color. Since the user mentioned highlighting, I'll opt for the font color red to avoid confusion. LibreOffice allows me to set this through the Format menu, so I'll focus on that. Next, I'll click the Format button, which is at approximately (626, 520) based on my screenshot estimation. This should help me apply the red font color to the replaced text effectively.

**Action**

import pyautogui import time pyautogui.click(626, 520)

## Step 21

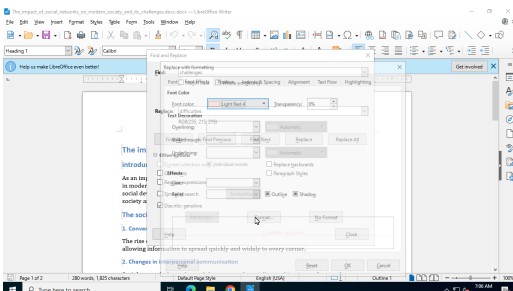

**Action**

pyautogui.click(482, 365, button='left')

## Step 26

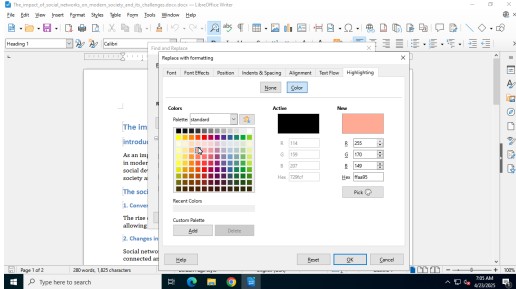

*Thought:* Confirming user's preferred red shade

**Action**

pyautogui.screenshot()

## Step 27

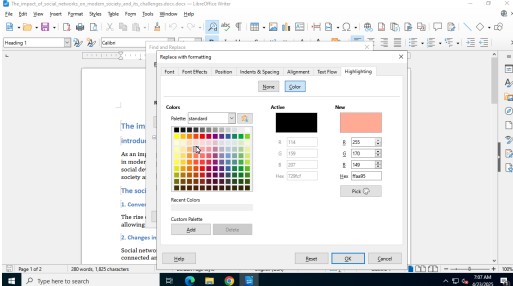

*Thought:* Hovering over red shades, confirming selection

**Action**

pyautogui.moveTo(464, 344)

## Step 28

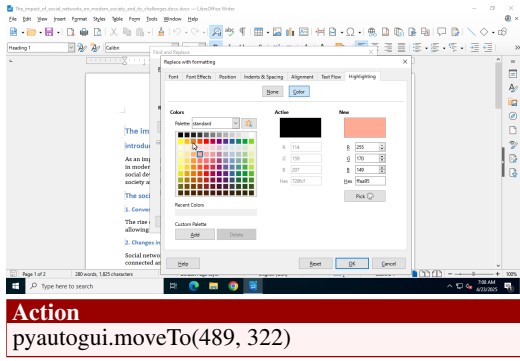

**Action**

pyautogui.moveTo(489, 322)

## Step 29

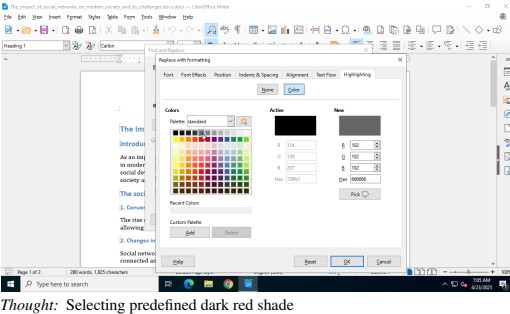

*Thought:* Selecting predefined dark red shade

**Action**

import pyautogui import time pyautogui.click(859, 333)

**Step 22**

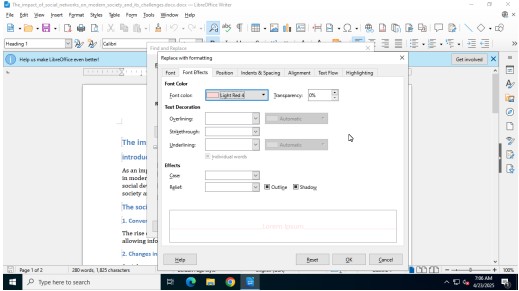

*Thought:* Analyzing find and replace steps I need to set up the Find & Replace dialog to find "challenges" and replace it with "difficulties," maybe considering case sensitivity since there might be plurals. Afterward, I want to confirm formatting by clicking OK in the small window, which has the previously selected Light Red 4 format. I'll also identify the approximate coordinates for the OK button — it seems to be around (857, 644) relative to the screenshot. So, I'll need to click that. Approximating button coordinates I'm estimating the coordinates for the OK button in the small window. The left border seems to be around 360 px and the right border maybe 880 px, making the window width about 500 px. So, I'm thinking the OK button is likely around (805, 646). Alternatively, checking the bottom row of buttons could help. It seems like the Reset button is near x 740, the Cancel button at x 880, so the OK button could be around x 800 and y 645. I'll use these coordinates for clicking.

**Action**

import pyautogui import time pyautogui.click(800, 645)

**Step 23**

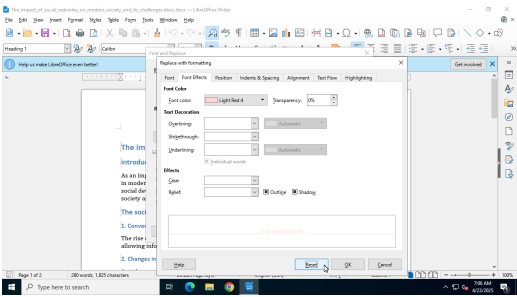

*Thought:* Planning text replacement I need to replace every instance of "challenges" with "difficulties" in LibreOffice Writer and highlight the changes in light red. I see a "Find and Replace" dialog with a formatting pane. Next, I'll confirm the font color to light red. After that, I'll ensure the "Find" field has "challenges" and the "Replace" field has "difficulties." It's essential to select "Replace all" and then click OK to apply the formatting. Finally, I'll proceed with the replacement.

**Action**

import pyautogui import time pyautogui.click(866, 644)

**Step 24**

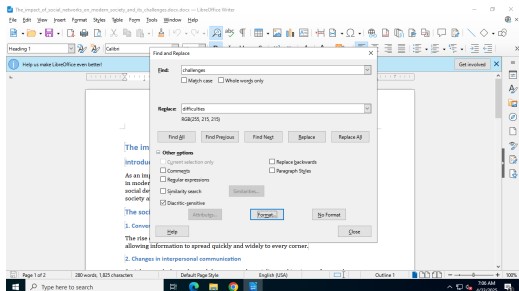

*Thought:* Planning text replacement steps I've got LibreOffice Writer open with the Find and Replace dialog ready. The "Find" field has "challenges" and the "Replace" field has "difficulties." The next step is to highlight the changed text in red. I see I need to click the Format button under Replace, choose Font Effects, and set the font color to red. I'll need to click this button at the bottom center of the dialog, which I've noted has a blue focus border. The approximate coordinates are (614, 520).

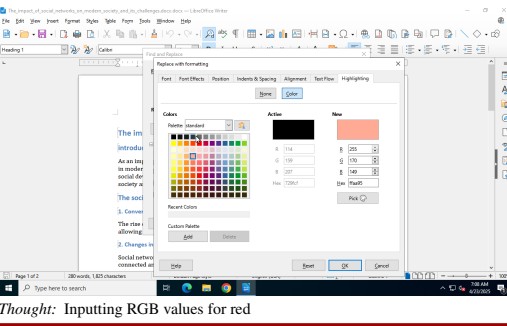

*Thought:* Inputting RGB values for red

**Action**

pyautogui.click(799, 353, button='left')

**Step 30**

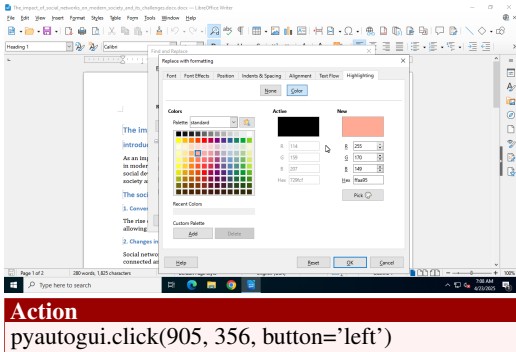

**Action**

pyautogui.click(905, 356, button='left')

**Step 31**

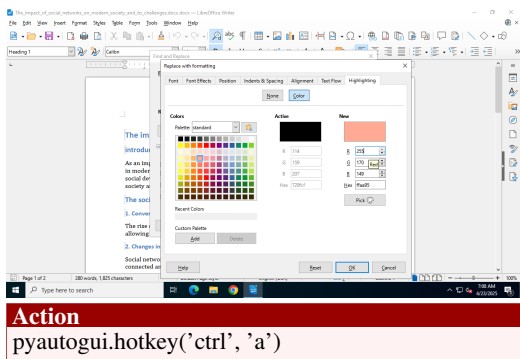

**Action**

pyautogui.hotkey('ctrl', 'a')

**Step 32**

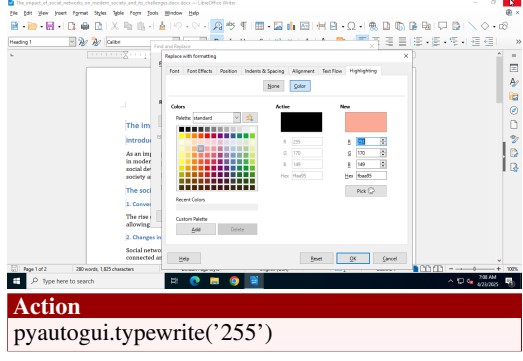

**Action**

pyautogui.typewrite('255')

**Step 33**

**Action**

import pyautogui import time pyautogui.click(640, 520)

### Step 25

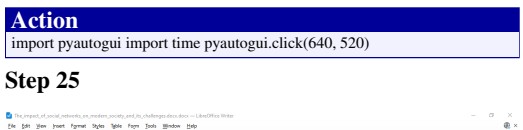

**Action**

import pyautogui import time pyautogui.click(640, 521)

### Step 26

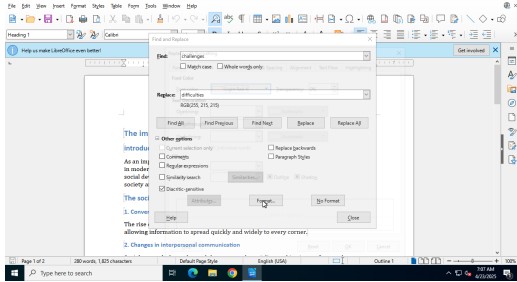

*Thought:* Detailing find and replace steps I need to describe the next step for replacing "challenges" with "difficulties" and applying red color formatting in LibreOffice Writer. The screenshot indicates that the "Replace with formatting" dialog is open, and it appears to have multiple category tabs like Font, Font Effects, and Position. The Font Effects tab is likely active, showing the "Font Color" section, which indicates that the chosen color is currently Light Red 4. Now, I need to set the font color to red. Explaining text replacement steps We need to change the replaced text to red since the user's requirement is to highlight it that way. In the "Replace with formatting" dialog, we should confirm the color selection. The next step is to click OK to apply the Light Red 4 color. After that, in the main Find & Replace dialog, we'll enter "challenges" as the text to find and "difficulties" as the text to replace it with. We should ensure the search is case-insensitive and then click Replace All. Let's put this into clear steps for the user!

**Action**

import pyautogui import time pyautogui.click(865, 644)

### Step 27

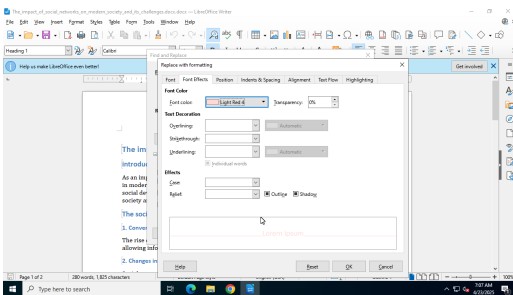

**Action**

import pyautogui import time pyautogui.click(650, 519)

### Step 28

**Action**

pyautogui.click(903, 381, button='left')

### Step 34

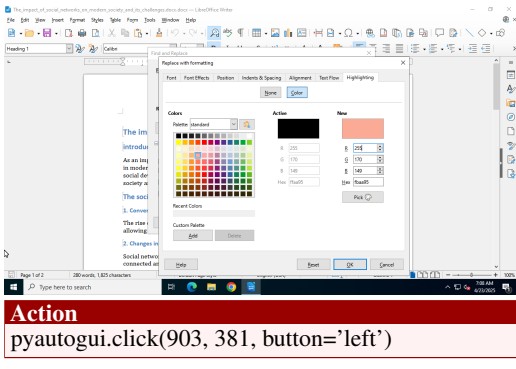

**Action**

pyautogui.hotkey('ctrl', 'a')

### Step 35

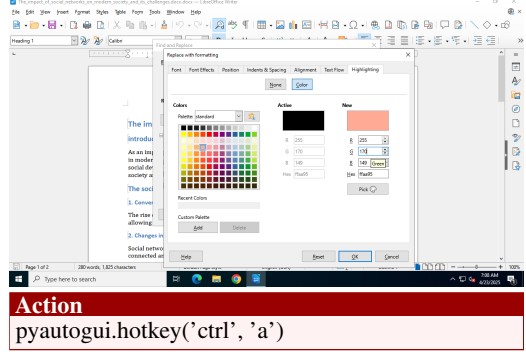

**Action**

pyautogui.typewrite('0')

### Step 36

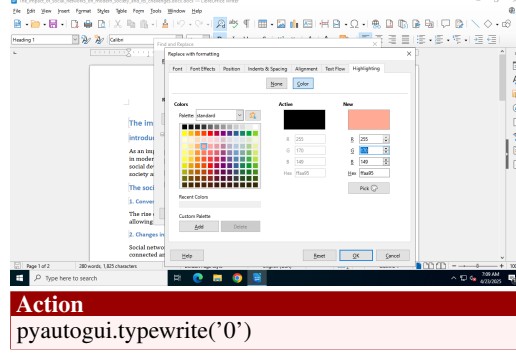

**Action**

pyautogui.click(907, 411, button='left')

### Step 37

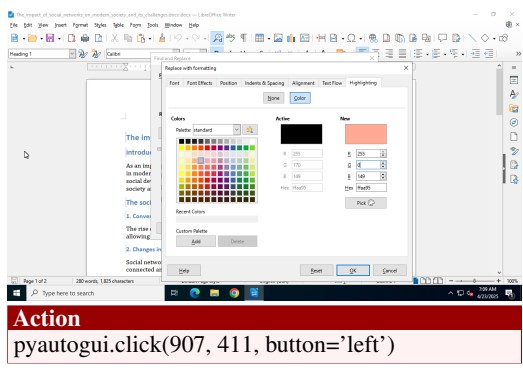

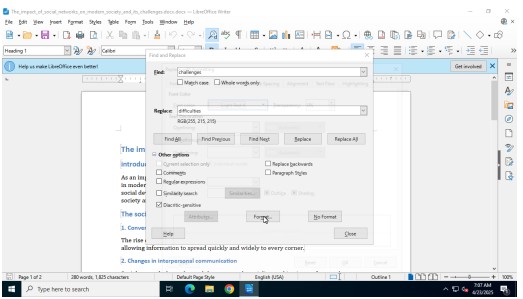

*Thought:* Configuring find and replace I need to replace all instances of "challenges" with "difficulties" and highlight the changes in red. The user specified to find "challenges" and replace with "difficulties." In LibreOffice Writer, I can set the font color to red during the replacement. I see the "Format..." button in the Replace options, so I'll click that to open formatting options. Then I'll choose Font Effects, set the font color to red, and finally click "Replace All."Planning mouse click coordinates I need to plan to use a mouse click around the coordinates (613, 523) to access the Format button under the Replace field. I'm estimating the right spot as about 610x520 based on the screenshot. It seems like specifying a left click at that Format button will help me proceed more accurately. Hopefully, that'll get me to the next step without any hiccups!

**Action**

import pyautogui import time pyautogui.click(642, 520)

## Step 29

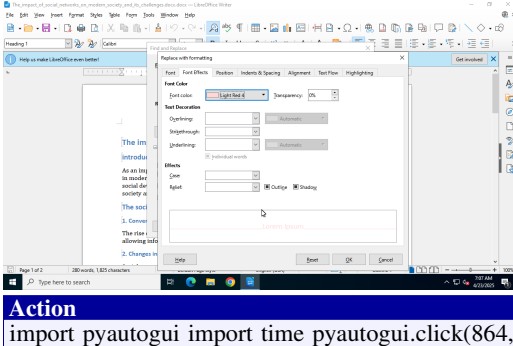

**Action**

import pyautogui import time pyautogui.click(864, 645)

## Step 30

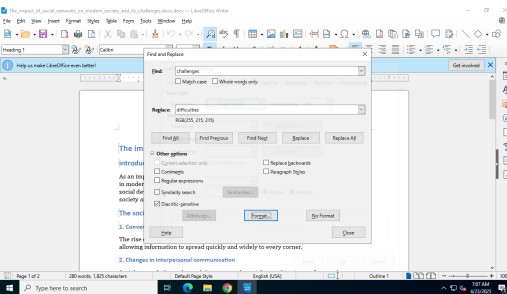

*Thought:* Adjusting text formatting The user is working in LibreOffice Writer and wants to replace "challenges" with "difficulties," highlighting the change in red. I see that there's a Format button for setting font color in the Find and Replace dialog. It's important to ensure we select the correct color. The screenshot shows a light pink color instead of red, which could mean the user picked the wrong shade earlier. Let's go ahead and open the Format option again to select red (#FF0000) for the text.Preparing to click The next step is to click the Format button, which is currently highlighted. Clicking this button will open the Character Format dialog. I'll need to simulate a left-click at the approximate coordinates of the button. Based on the screenshot, I estimate the coordinates to be around (640, 513) because the button is near the bottom center of the dialog. I'll prepare the tool call for a left-click at these coordinates.

**Action**

import pyautogui import time pyautogui.click(640, 513)

## Step 31

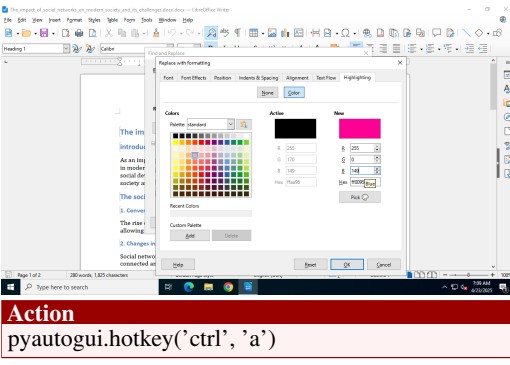

**Action**

pyautogui.hotkey('ctrl', 'a')

## Step 38

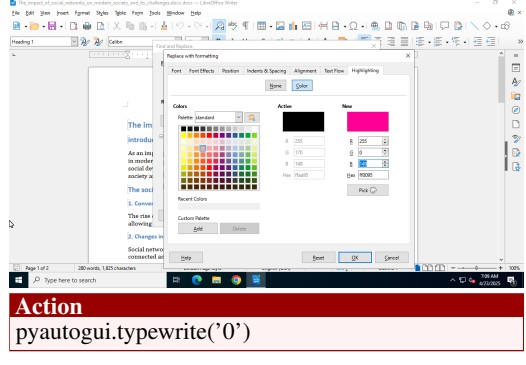

**Action**

pyautogui.typewrite('0')

## Step 39

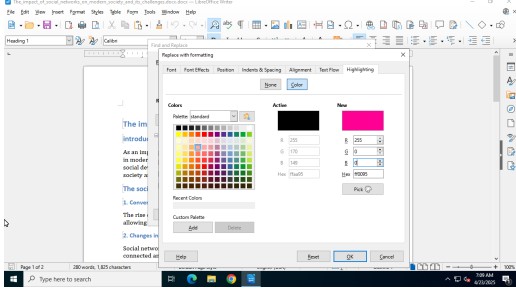

*Thought:* Applying red color, clicking "OK"

**Action**

pyautogui.click(875, 644, button='left')

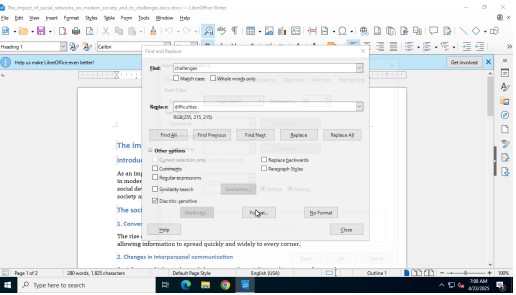

*Thought:* Setting replacement formatting The user wants to replace "challenges" with "difficulties" and highlight the changes in red, so I'm opening Find and Replace in LibreOffice Writer. I've entered "challenges" as the text to find and "difficulties" as the replacement. Next, I have to set the formatting to highlight the replacements in red. The "Format..." button is available to do this. I'll click on it to set the font color red for the replacement text, focusing on getting the coordinates right to click correctly.

**Action**

import pyautogui import time pyautogui.click(629, 525)

## Step 32

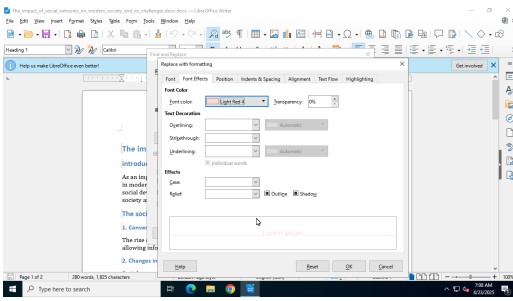

**Action**

import pyautogui import time pyautogui.click(863, 646)

## Step 33

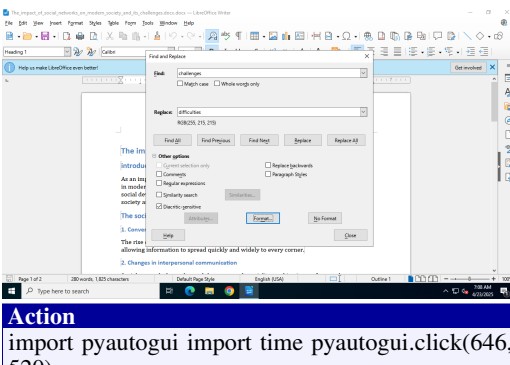

**Action**

import pyautogui import time pyautogui.click(646, 520)

## Step 34

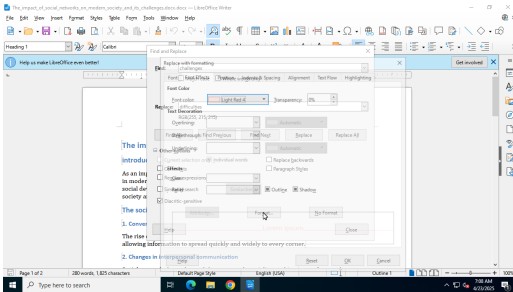

**Action**

import pyautogui import time pyautogui.click(850, 640)

### Step 35

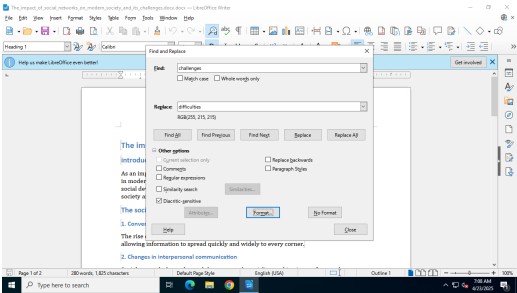

*Thought:* Updating text formatting I need to change all instances of "challenges" to "difficulties" and highlight those changes in red. I've opened the Find and Replace feature and entered the words correctly. I need to make sure to format the replacement text to be red. I'll click on the Format button next to the Replace field, and in the dialog, I'll set the font color to red. After that, I can click Replace All to finalize the changes.Planning formatting steps I've noticed that the Format button is highlighted in blue, indicating it's potentially selected but hasn't been clicked yet. I need to click that Format button, then navigate to the Font Effects tab to choose the font color red. This seems straightforward, so I'll plan my next steps, starting with that first click on the Format button. It's all about getting those settings right for the best results!

**Action**

import pyautogui import time pyautogui.click(644, 521)

### Step 36

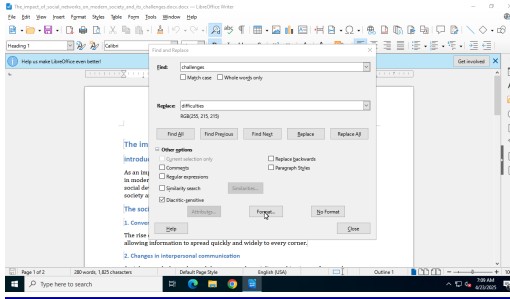

**Action**

import pyautogui import time pyautogui.click(629, 521)

### Step 37

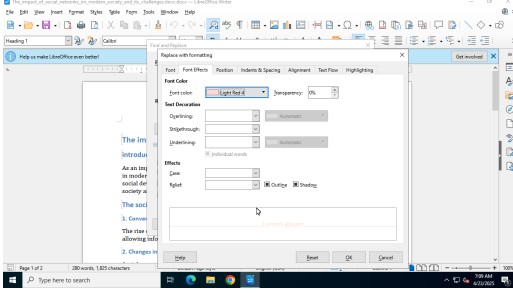

*Thought:* Handling Find and Replace I've got the LO Writer open and I need to replace "challenges" with "difficulties," highlighting the changes in red. After pressing Ctrl+H, I see the "Find and Replace" dialog with a section for formatting where I've set the font color to Light Red 4. Now, I need to confirm this and close the formatting window, then enter "challenges" in the search box and "difficulties" as the replacement term. I'll ensure the red formatting is applied before clicking OK.Confirming the action I've identified that the OK button is located at the bottom right of the sub-dialog, positioned between Reset and Cancel. I plan to click OK, estimating its coordinates to be around (866,645) based on the screenshot dimensions of 1280x720. The button is the third one from the left in the bottom row. Next, I'll proceed with the click and monitor to ensure that the action registers correctly.

**Action**

import pyautogui import time pyautogui.click(870, 645)

## Step 38

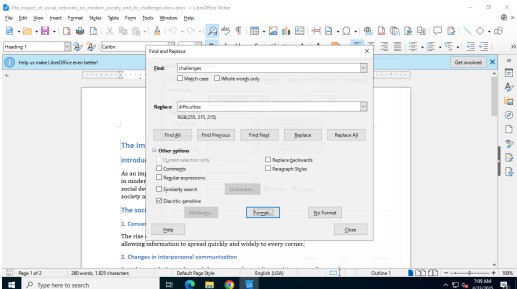

*Thought:* Determining text replacement format I need to replace all occurrences of "challenges" with "difficulties" and highlight the new text in red. I've opened the Find and Replace window. The "Find" field is already filled with "challenges", and "Replace" has "difficulties". Now, I must set the formatting to red. I'm unsure if "highlighted" means the text color should be red or the background color should be a highlight. Typically, "highlight" refers to background color, but the formatting options may clarify this. Finalizing text replacement settings The replacement section shows "difficulties" and a highlight color of RGB(255, 215, 215), which looks like a light red or pink. This suggests that the highlighting may already be set. Next, I need to click "Replace All" to execute the changes. The "Replace All" button is situated near the right side of the window. I should ensure that the background is highlighted in the correct color—perhaps adjusting it to a true red if needed. Let's use the Format option to confirm and set the color to red. Adjusting highlight settings It might be simpler to accept that the highlight is already set. Looking at the snippet, the "Replace" area shows "difficulties" with a highlight color of RGB(255, 215, 215). This seems lighter than a typical strong red. I should adjust it to a standard red color instead. My plan is to click on Format, then go to the Font Effects tab, select red, and then execute "Replace All." I'll pinpoint the coordinates for the Format button and proceed with the click.

**Action**

import pyautogui import time pyautogui.click(623, 512)

## Step 39

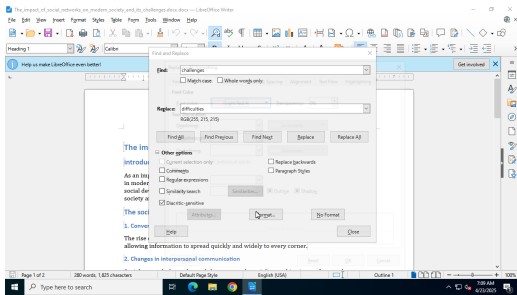

*Thought:* Changing text in document I've got a LibreOffice Writer document open, and my task is to replace "challenges" with "difficulties" while highlighting the changed text in red. I see the Find and Replace dialog is up, and I need to specify "Find: challenges" and "Replace: difficulties." For the formatting, I need to click the Format button to set the font color to red. Once that's done, I'll select Replace All. I'm checking that format button to ensure I'm clicking the right one. Setting font color in document I need to set the font color to red in the document. To do this, I'll click the Format... button under the Replace section. This action will open the Character Format menu where I can choose the red foreground color. I'll need to be sure I'm clicking the right button at the correct coordinates to ensure it opens properly. Once that's done, I can proceed with the text replacement easily.

**Action**

import pyautogui import time pyautogui.click(630, 523)

## Step 40

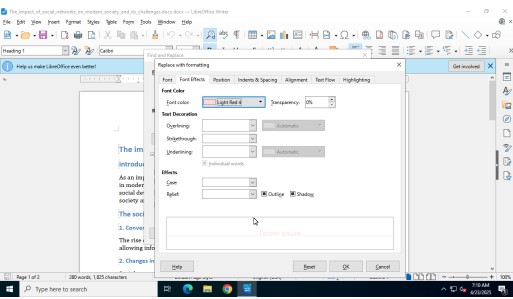

*Thought:* Configuring text replacement I'm working on replacing all instances of "challenges" with "difficulties" and changing the font color to red. I opened the Replace with Formatting window and saw that the Font Effects tab is selected, which shows Light Red 4 as the current color. Now I need to finalize the formatting by clicking OK in the dialog. After that, I'll set up the search terms to replace the text. Then I'll check "Find All" or "Replace All" for a complete update.Clicking the OK button Alright, my next action is to click the OK button at the bottom right of the window. I've checked the screenshot, and it seems that the button is situated there. I'm estimating I need to aim for the center of the OK button. I'll go ahead and do that now. This should confirm my formatting choices and allow me to move on to the next step in the process. Let's get this done!

**Action**

import pyautogui import time pyautogui.click(866, 645)

## Step 41

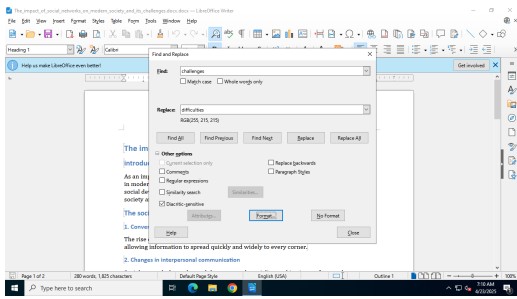

*Thought:* Setting text color in Find and Replace The user wants me to change all occurrences of "challenges" to "difficulties" and highlight the changes in red. I've opened the Find and Replace dialog, and the fields are set correctly. Now, I need to ensure that the changed text is highlighted in red. There's a 'Format...' button I need to click to set the font color. I'll locate and click this button to adjust the color. The coordinates suggest it's in the middle bottom of the window.

**Action**

import pyautogui import time pyautogui.click(630, 518)

## Step 42

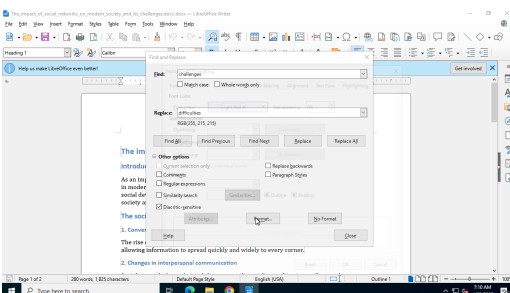

*Thought:* Analyzing Find and Replace interface I see a "Replace" label in the translucent overlay, which might be obstructing earlier clicks. Let's analyze the "Find and Replace" window more closely. At the top, there are fields for Find and Replace, followed by buttons: Find All, Find Previous, Find Next, Replace, and Replace All. The "Other options" group has checkboxes, and at the bottom, there are buttons for Attributes, Format, No Format, and Close. There's a small drop-down arrow next to the Replace field for formatting settings, similar to the format button in Writer 7.6.Estimating dialog layout I'm estimating the dialog's positioning, with the top-left around (350, 110) and the Replace field near (412, 258), making the width about 300. There's an arrow on the right, likely at (715, 258). After clicking this arrow, I expect a drop-down with "Format..." to appear; though it might be small, we could also click "Format..." from the bottom row. I plan to click the arrow at (765, 256) using a left click and then wait for the next action.

**Action**

import pyautogui import time pyautogui.click(765, 256)

## Step 43

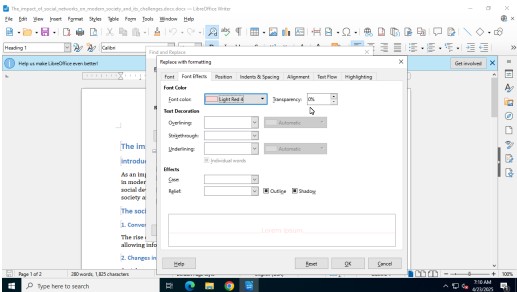

*Thought:* Analyzing the screenshot task The user provided a screenshot and wants me to help with a task involving replacing "challenges" with "difficulties" and highlighting the changes in red. They've opened the Find and Replace dialog in LibreOffice Writer, which shows the formatting options ready to go. The next step needs to be confirming the search for "challenges" and inputting "difficulties" as the replacement term. I'll make sure to apply the Light Red 4 color for the changes!Confirming formatting choices Now that the formatting dialog is open, the immediate next step is to click OK to confirm the color selection. After that, I need to return to the Find & Replace dialog and set the search and replace terms. So, first, I'll click the OK button at the bottom right of the "Replace with formatting" dialog. I'm estimating the coordinates for the click, focusing on that area. Once that's done, I'll move forward with the task.

| Action |
|---|
| import pyautogui import time pyautogui.click(864, 643) |

## Step 44

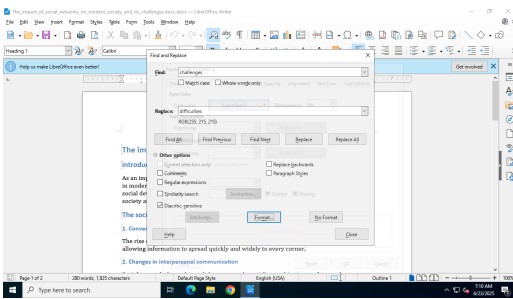

*Thought:* Adjusting text in LibreOffice The user is working in LibreOffice Writer and needs to replace "challenges" with "difficulties," highlighting the changes in red. I've opened the find and replace window with the terms already filled in. I'll need to click the Format button to set the character font color to red by navigating through the options. However, I'm wondering if the request is for a highlight color instead of just changing the font color. Since highlighting typically means changing the background, setting the font color to red feels like the simpler choice.

| Action |
|---|
| import pyautogui import time pyautogui.click(615, 518) |

## Step 45

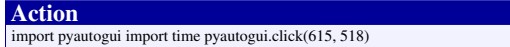

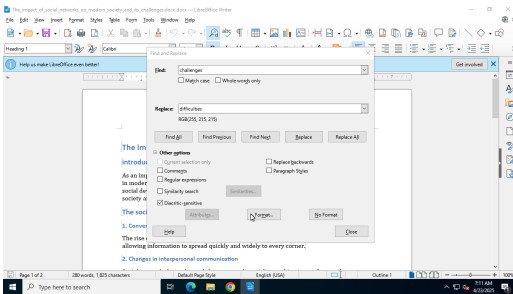

*Thought:* Clarifying text replacement I realize I need to replace all instances of "challenges" with "difficulties" and highlight them in red. I opened the Find & Replace dialog and noticed the "Find" field has "challenges" while the "Replace" field shows "difficulties" in a light pink color instead of red. It seems the background color was previously set using the Format button, but I need to adjust it for proper red highlighting, specifically with the RGB value of 255,0,0 to achieve that.Setting highlight color I need to set the highlight color for the replaced text to red. There's an option in LibreOffice for either Font Color or Highlighting. The user wants the text to be highlighted in red, which usually implies a background color. However, I'm also considering that they might just prefer the font itself to be red. Since the manual suggests highlight is typically yellow, I'm interpreting "highlight in red" as background highlight. So, I'll adjust the Replace field format

to ensure it reflects the change properly by selecting the red highlight option. Time to click on the Format button!Estimating dialog position I'm trying to approximate the location of the dialog box. It appears to be centered horizontally at around 580, 518. By estimating, it seems the dialog is mid-screen, starting around x 400 px and has a width of about 400 px. The Format button likely sits near the bottom center at around 540, 300. After looking at the screenshot dimensions, I'll set my click at roughly (600, 510) to target the Format button effectively. Time to click!

**Action**

import pyautogui import time pyautogui.click(598, 520)

### Step 46

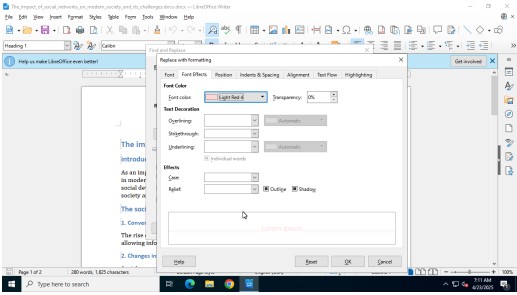

*Thought:* Executing find and replace I'm currently working in LibreOffice Writer. The user's request is to replace all instances of "challenges" with "difficulties" and to highlight those changes in red. I've opened the Find and Replace window and have the formatting box ready to go. It mentions "Light Red 4," which works for my purposes. I need to confirm the color by clicking the "OK" button in the formatting dialog. Once I do that, I can proceed with the replacement.

**Action**

import pyautogui import time pyautogui.click(867, 644)

### Step 47

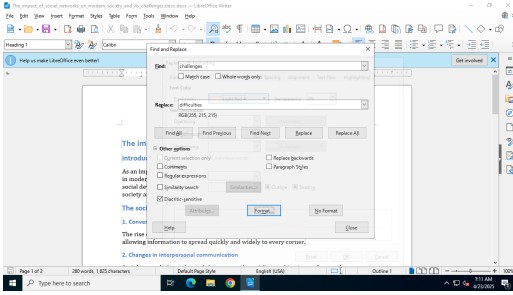

*Thought:* Planning text replacement I need to make some changes: replacing all instances of "challenges" with "difficulties" and highlighting that changed text in red. The Find and Replace dialog has the correct fields ready, with "challenges" and "difficulties" set up. Now, I need to format the replacement text to show in red. I'll click the Format... button which is currently highlighted. I'll use my mouse to do this and approximate the coordinates for clicking around (615, 517). Let's do that!

**Action**

import pyautogui import time pyautogui.click(628, 515)

### Step 48

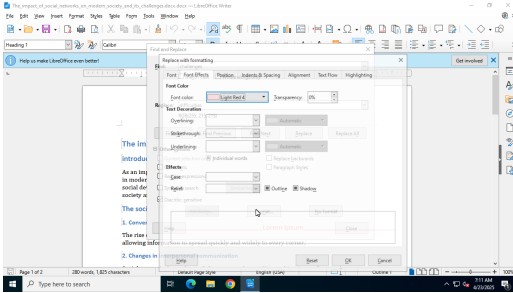

*Thought:* Planning the next steps The user has asked me to analyze a screenshot related to changing all occurrences of "challenges" to "difficulties" and highlighting the changes in red. They've opened the Find and Replace window, which shows a formatting option for the replacements. I see that in this secondary "Format" selection window, "Font Color" is set to Light Red 4, which is excellent. My next step is to confirm the replacement and apply this formatting to ensure the text is correctly updated.Confirming format changes I need to click the "OK" button on

the Format window to apply the red font color. After that, in the main Find and Replace, I'll enter "challenges" as the term to find and "difficulties" to replace it. It seems the Find and Replace window is overshadowed by the Format window, making it look faded. I can see the buttons at the bottom of the Format window, and I need to click "OK" to confirm. I'm estimating the coordinates for the button since it's not fully clear. Let's proceed!Confirming the action I see the buttons "Reset," "OK," and "Cancel" in the screenshot, and I'm estimating the coordinate for the "OK" button to be around (847, 641). My next step is to perform a left-click on that button to confirm the Light Red 4 color for the replacement text. After that, I'll regain focus on the main window. Then I should respond with the appropriate next steps. Let's keep moving forward!

**Action**

import pyautogui import time pyautogui.click(842, 641)

## Step 49

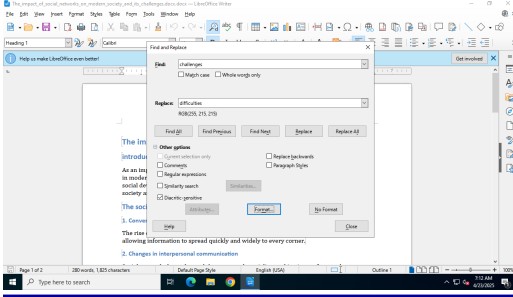

**Action**

import pyautogui import time pyautogui.click(623, 515)

## Step 50

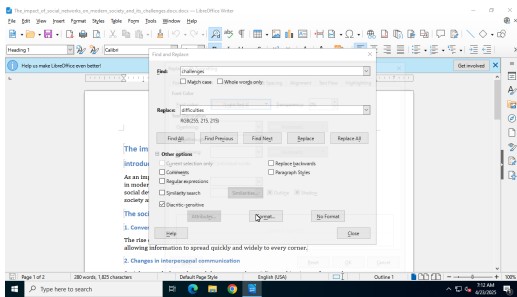

*Thought:* Checking Find and Replace setup The user wants to replace "challenges" with "difficulties" and highlight them in red. I've opened the Find and Replace tool and set it up, with "Find: challenges" and "Replace: difficulties." Now, I need to apply the highlight formatting for the replacement text. In LibreOffice Writer, there's a "Format..." button next to the Replace field that I'll need to click. I see something about RGB(255, 215, 215) in the screenshot, indicating that the light red highlight might already be selected. Let's double-check that formatting!Setting highlight formatting I see options like "Attributes..." and "Format..." under the Replace area. There's a button for "Format..." that I need to click to open the Character Format dialog for setting the highlight color to red. The user wants the changed text highlighted, which I assume means a font color of red. So, I'll click the Format button, navigate to Font Effects, select Font Color, and choose red. Then, I'll confirm and click Replace All. Let's get started by clicking that Format button!

**Action**

import pyautogui import time pyautogui.click(627, 522)

