# OpenReview forum: "Computer Agent Arena: Toward Human-Centric Evaluation and Analysis of Computer-Use Agents"
_ICLR.cc/2026/Conference — ICLR 2026 Poster_

### Official Review · Reviewer_rdUN · 2025-10-30

**Soundness:** 3
**Presentation:** 3
**Contribution:** 2
**Rating:** 6
**Confidence:** 4

**Summary:**

The paper introduces Computer Agent Arena, a platform where users can perform pairwise evaluation of agents on user-provided tasks. Analysing data generated in this way, they find that static benchmark performance does not necessarily correlate with user preference on these more ambiguous and open-ended tasks.

**Strengths:**

- A highly configurable environment allowing for diverse user-defined tasks
- Quick start tools reduce friction when initialising new tasks
- Very in-depth analysis in body and appendices
- Likely relevant for significantly longer than static benchmarks

**Weaknesses:**

- The number of votes is fairly small when spread over the task distribution.
- Evaluating agent traces and setting up new tasks is significantly more demanding than simply asking questions (as in LLMArena), which may limit the number of users willing to perform comparisons and hence the future relevance of the platform.

**Questions:**

- I find the poor performance of GPT-5 and Gemini 2.5 Pro quite surprising. Do you have hypotheses or analyses that suggest what the cause may be?
- You mention that long-horizon memory errors and fine-grained action failures are both common. Can you give per-agent error rates in these categories?
- Could you train a learned preference model on your data, incorporating correctness, latency, number of queries, etc, that would allow your human data to be reused in an offline setting (i.e. without requiring reranking for a quick oversight of a new model)?

---

> ### Author Response · Authors · 2025-11-20
>
> Thank you for the thoughtful and constructive review! We are grateful for your recognition of our platform's highly configurable environment, quick-start tools, very in-depth analysis, and long-term relevance. We appreciate the insightful questions regarding model performance, error patterns, and future research directions.
>
> Below, we address each point in turn and clarify our contributions and ongoing efforts.
>
> ---
>
> > **W1: The number of votes is fairly small when spread over the task distribution**
> >
>
> **A:** Thank you for this important observation. While our current dataset (2,201 votes) is smaller than established platforms like Chatbot Arena, we would like to emphasize that the collected data provides robust and reliable rankings.
>
> Our comprehensive statistical validation (Appendix D) demonstrates sufficient power. Bootstrap analysis (1,000 iterations) shows that top-ranked models maintain non-overlapping 95% confidence intervals (Figure 2b). Power analysis confirms that with n=1,661 votes, we achieve ≥0.90 power to detect medium effect sizes (ΔElo ≈ 50 at α=0.05). Permutation tests show pairwise win-rate differences are highly significant (p < 0.01) with medium-to-large effect sizes (Cohen's d > 0.5). Inter-annotator agreement is strong with Krippendorff's α = 0.72 (preferences) and 0.78 (correctness). These results confirm our dataset is well-powered to detect meaningful differences between models.
>
> **Quality over quantity.** Each vote in COMPUTER AGENT ARENA carries substantially more information than typical LLM Arena comparisons. Our votes include pairwise preference + 3-level correctness + safety + efficiency ratings + optional qualitative comments. Users assess multi-step agent behaviors (average ~7-15 steps per trajectory) across diverse environments, not single-turn responses. While Chatbot Arena has collected >1M votes, each evaluates a single conversational turn. Our votes evaluate complete task executions involving multiple actions, error recovery, and environmental interactions—orders of magnitude more complex. Additionally, 79.3% of our tasks are out-of-domain (Table 2), reflecting authentic user needs rather than benchmark templates.
>
> Our 1,058 unique users (821 Prolific + 237 public) reflect broad demographic diversity: 20+ countries across North America, Europe, and Asia; 47% bachelor's degree or higher; varied employment status (full-time 32%, part-time 21%, students 26%); mean age ≈ 33.2 years (SD ≈ 9.6). This diversity ensures our findings generalize across realistic user populations.
>
> COMPUTER AGENT ARENA is an open, continuously evolving platform. The platform remains publicly accessible and is actively collecting new votes. We will update the leaderboard with expanded data in revised versions. Importantly, our current dataset already sufficiently supports our core claims about benchmark-arena ranking inversions, the role of behavioral factors in preference, tool-integrated agent challenges, and implicit error modes that static benchmarks miss.
>
> ---

---

> > ### Author Response · Authors · 2025-11-20
> >
> > > **W2: Evaluating agent traces is more demanding than LLMArena, potentially limiting participation**
> > >
> >
> > **A:** We acknowledge that evaluating GUI agents is inherently more cognitively demanding than evaluating conversational LLMs. However, we argue that this complexity is unavoidable for meaningful CUA evaluation and have taken deliberate design steps to mitigate friction.
> >
> > **The complexity is intrinsic to the domain.** There is no shortcut to authentic CUA evaluation—users must observe actual agent-computer interactions to assess whether agents click correct UI elements, recover from errors gracefully, make reasonable strategic decisions, and respect privacy constraints. Any simplification (e.g., evaluating text-only plans without execution) would fundamentally fail to capture what makes CUAs useful or problematic in deployment.
> >
> > **We have invested significant effort in lowering barriers to participation.** Our browser-based interface requires no installation—users access via web browser with embedded VNC. Agent actions are rendered as annotated GIFs with overlays (bounding boxes for clicks, icons for typing) to clarify each step (Figure 6). Side-by-side synchronized playback enables efficient relative judgments. Most importantly, our quick-start tools (Section 2.1, Appendix B.2) dramatically reduce setup burden: `upload_file` allows direct file uploads to VMs, `open_websites` preloads specific URLs, and `clone_repo` automates GitHub repository cloning. We provide 600+ pre-configured initializations spanning 89 domains + 472 subdomains, 12 popular applications, and 97 heterogeneous files. These tools allow users to set up realistic evaluation scenarios in one click, compared to manual VM configuration. Our evaluation interface uses structured forms with clear questions (Figure 7) and optional (not required) comment fields.
> >
> > **Empirical evidence confirms user engagement.** Despite the inherent complexity, we successfully recruited 1,058 unique users who contributed 3,418 total evaluation votes (2,201 retained after quality control). Users provided optional qualitative comments in many cases, suggesting genuine engagement with agent behaviors. Moderate-to-strong inter-annotator agreement (α = 0.68-0.78) demonstrates users understood evaluation criteria.
> >
> > We are actively exploring ways to make participation even more accessible: lightweight evaluation modes for faster annotations, community task libraries with reusable tasks, video tutorials and guided onboarding, and incentive mechanisms to encourage sustained participation.
> >
> > ---

---

> > > ### Author Response · Authors · 2025-11-20
> > >
> > > > **Q1: Poor performance of GPT-5 and Gemini 2.5 Pro – what may be the cause?**
> > > >
> > >
> > > **A:** Thank you for this insightful question! We share your surprise at these findings and conducted an in-depth case study to understand the underlying causes. The underperformance reveals important lessons about the specific capabilities required for GUI agents.
> > >
> > > **Primary cause: Grounding capability deficiency.** Our case-by-case analysis reveals that the dominant failure mode (>70% of error cases) stems from UI element grounding errors—the inability to accurately predict screen coordinates for target elements after planning and reasoning. In the GUI agent domain, grounding refers to the ability to take a screenshot and output the precise coordinates (x, y) or bounding box of a target UI element. This requires specialized training on large-scale pairs of UI screenshots and element coordinates/bounding boxes. Benchmarks like OSWorld-G[1] and ScreenSpot Pro[2] specifically measure this skill.
> > >
> > > We could not find grounding performance metrics for GPT-5 or Gemini 2.5 Pro in their official reports or third-party evaluations, suggesting these models were not explicitly trained for this task. In the majority of failed trajectories for these models, we observed grounding errors leading to local loops—the agent correctly identifies what action to take but clicks on the wrong location, leading to repeated failed attempts. For Gemini 2.5 Pro specifically, we observed extensive repetitive action loops where the model outputs the same action repeatedly without effective error recovery or strategy adjustment. **News:** we find recently released Gemini 3 Pro reaches excellent performance on ScreenSpot Pro, and we will introduce Gemini 3 Pro to the Arena.
> > >
> > > Several recent agent framework papers validate this finding by adopting a planner + grounder architecture. For example, Jedi [1] uses o3 as a planner paired with a specialized grounder, and Agent-S3 [3] uses GPT-5 (High) as a planner combined with UI-TARS-1.5-7B as a grounder. These hybrid approaches achieve strong benchmark performance, demonstrating that general-purpose models benefit significantly from dedicated grounding modules.
> > >
> > > **Interesting discovery: Reasoning as a compensatory mechanism.** We also discovered an unexpected pattern—reasoning models (o3, o3-mini) significantly outperform non-reasoning models on our platform, despite lacking specialized GUI training. Through trajectory analysis, we found that reasoning models explicitly reason about spatial coordinates during their thinking process. For example: *"The Terminal icon is in the third position of the taskbar, occupying approximately 25% of the screen width, so the coordinates should be around (67, 255)."* This explicit spatial reasoning appears to partially compensate for the lack of specialized grounding training, resulting in higher Elo scores and better overall performance on GUI tasks compared to non-reasoning commercial models.
> > >
> > > We attribute the poor performance of GPT-5 and Gemini 2.5 Pro to: (1) lack of GUI-specific training on large-scale trajectory data with coordinate annotations, (2) missing grounding capabilities essential for GUI interaction, and (3) limited error recovery mechanisms (particularly for Gemini 2.5 Pro). Our findings highlight that general multimodal excellence does not automatically translate to computer-use proficiency—GUI agents require either specialized grounding training or sufficiently strong spatial reasoning capabilities. This insight should guide future development of computer-use models and evaluation methodologies. We will expand this analysis in the camera-ready version with additional visualizations of failure cases and quantitative breakdowns across error types.
> > >
> > > **References:**
> > >
> > > [1] T. Xie et al., "Scaling Computer-Use Grounding via User Interface Decomposition and Synthesis," arXiv preprint arXiv:2505.13227, 2025. [Online]. Available: https://arxiv.org/abs/2505.13227
> > >
> > > [2] K. Li et al., "ScreenSpot-Pro: GUI Grounding for Professional High-Resolution Computer Use," arXiv preprint arXiv:2504.07981, 2025. [Online]. Available: https://arxiv.org/pdf/2504.07981
> > >
> > > [3] Simular AI, "Agent-S3: A Scalable System for Autonomous Desktop and Web Task Automation," 2024. [Online]. Available: https://www.simular.ai/articles/agent-s3
> > >
> > > ---

---

> > > > ### Author Response · Authors · 2025-11-20
> > > >
> > > > > **Q2: Per-agent error rates for long-horizon memory and fine-grained action failures**
> > > > >
> > > >
> > > > **A:** Thank you for this valuable question. We agree that quantifying these error types per agent would provide crucial insights into model-specific weaknesses. These error categories emerged from our qualitative analysis of user comments and represent implicit failure modes that are challenging to detect through automated scripts or rule-based methods.
> > > >
> > > > Unlike explicit errors (e.g., grounding failures detected via coordinate mismatches), the error types we identified require nuanced human judgment. Long-horizon memory failures involve detecting when an agent "forgets" context from many steps ago, which requires understanding task semantics. Fine-grained action failures require distinguishing between intentional scrolling versus "stuck" scrolling through trajectory-level reasoning. Information awareness involves identifying whether an agent should have asked for clarification, which is inherently subjective. These patterns cannot be reliably captured by scripted evaluation functions, which is precisely why human-centric evaluation platforms like Computer Agent Arena are essential.
> > > >
> > > > **Our initial analysis.** To provide preliminary insights, we conducted a systematic error taxonomy study. We sampled 100 representative failed trajectories from our dataset, employed GPT-5 for initial automated classification, and conducted manual review and refinement by experts. We identified 3 primary error categories beyond basic grounding/planning failures:
> > > >
> > > > | Error Type | Description | Counts |
> > > > | --- | --- | --- |
> > > > | Long-horizon Memory | Agent loses track of multi-step goals or forgets previously retrieved information | 9 |
> > > > | Fine-grained Action Control | Incorrect scrolling, clicks on non-interactive elements, text editing errors | 12 |
> > > > | Information Awareness | Fails to seek clarification on ambiguous instructions | 15 |
> > > > | Basic Grounding/Planning | Standard coordinate errors, missing elements | 26 |
> > > > | Others | Other errors | 12 |
> > > >
> > > > Key observations: (1) Claude models excel at information awareness through effective use of CALL_USER for clarification, (2) fine-grained action failures plague general-purpose models like GPT-4.1 and Gemini 2.5 Pro, reinforcing our findings in Q1 about grounding deficiencies, and (3) tool-integrated agents face unique challenges with tool selection errors.
> > > >
> > > > **Future work and dataset release.** We recognize the importance of comprehensive error quantification for guiding community research. We are scaling up annotation to more trajectories with multiple annotators, developing automated detection methods using LLM-based features and trajectory signals, and committing to release our complete annotated dataset with error labels, inter-annotator agreement scores, and annotation guidelines in the camera-ready version. This dataset will enable the community to train models specifically to avoid these error patterns, develop better evaluation metrics beyond binary correctness, and study error recovery and self-correction mechanisms.
> > > >
> > > > ---

---

> > > > > ### Author Response · Authors · 2025-11-20
> > > > >
> > > > > > **Q3: Training a learned preference model for offline evaluation**
> > > > > >
> > > > >
> > > > > **A:** Absolutely, yes! This is an excellent suggestion that aligns perfectly with our goals for CAA. Thank you for pointing out this important direction.
> > > > >
> > > > > CAA's platform design and human-labeled data are well-positioned to support training reward models and preference models. For preference and correctness, we already have dense annotations across all 2,201 votes—pairwise preferences (α = 0.72), three-level correctness labels (α = 0.78), safety assessments (α = 0.68), and efficiency ratings (α = 0.70). For behavioral metrics like latency and number of queries, these are deterministic and already tracked in our system—we simply compute them through timing and query detection.
> > > > >
> > > > > We commit to open-sourcing the complete dataset to enable the community to build preference and reward models. This will be the largest human-labeled RLHF dataset for computer-use agents. The dataset will include all pairwise votes, correctness labels, and computed behavioral features (latency, query counts, step counts, reflection frequency, etc.).
> > > > >
> > > > > Beyond data release, we plan to train baseline preference models and correctness reward models in future work. These models would enable: (1) offline evaluation of new models without requiring fresh human rankings—allowing quick oversight as you suggested, and (2) online learning via RL-based training using the learned reward signals. We will add this roadmap to the revised manuscript and acknowledge your valuable suggestion.
> > > > >
> > > > > We appreciate you highlighting this direction—it represents a natural evolution from evaluation to optimization, and we're excited to pursue it with the community.
> > > > >
> > > > > ---
> > > > >
> > > > > We sincerely appreciate your detailed feedback and hope the above responses address all your concerns. We are committed to continuously improving the platform and expanding the dataset, and we welcome the community's participation in this effort. If you have any questions, we are pleased to provide further clarification!

---

### Official Review · Reviewer_hCqQ · 2025-10-31

**Soundness:** 3
**Presentation:** 3
**Contribution:** 4
**Rating:** 8
**Confidence:** 4

**Summary:**

This paper presents COMPUTER AGENT ARENA (CAA), an open-source platform for evaluating Computer-Use Agents (CUAs) in realistic computer environments using human preference–based judgments. CAA moves beyond static and contamination-prone benchmarks such as OSWorld or WebArena by supporting dynamic, side-by-side evaluation of agents in identical cloud-hosted virtual machines. Each session runs two anonymized agents that perform the same user-defined task, and human participants rate their outcomes and behaviors. The collected preferences are aggregated into an Elo leaderboard through a Bradley–Terry model, with optional stepwise labels that capture correctness, safety, and self-correction.

The authors gather 2,201 high-quality votes from 1,058 users across 12 agents, including commercial, open-source, and multimodal models. Results show significant ranking reversals compared with static benchmarks: models strong on OSWorld often perform worse on CAA. Human preference is driven mainly by correctness, but users also value process quality such as self-correction, reasoning clarity, and judicious use of CALL_USER queries. Both excessive and absent user interactions reduce satisfaction. Error analysis highlights failure types that static benchmarks overlook, including long-horizon memory drift, insufficient clarification, and fine-grained GUI control errors. The paper also releases the platform, dataset, and codebase to support open and reproducible human-centric evaluation of CUAs.

**Strengths:**

1. **Originality and significance**: CAA introduces a new evaluation paradigm for computer-use agents. It extends the “Arena” concept pioneered by Chatbot Arena into the a more complex, multimodal, and stateful domain of GUI-based computer interaction. By converting crowd-sourced, real-world tasks and pairwise human preferences into structured feedback, CAA establishes a scalable, human-centric framework for CUA assessment. This contribution is conceptually innovative and practically significant, redirecting focus from benchmark scores to real user satisfaction and behavioral quality.
2. **Technical quality and system design**:  The system demonstrates great engineering practice. The cloud-based virtual machine infrastructure provides elastic scaling, hundreds of preset setups, and user-defined customizations for authentic computer use. Agents run in matched environments with identical AMI fingerprints and software versions, ensuring fair comparison. The ranking method is statistically sound and is supported by bootstrapping, permutation tests, and power analysis. It delivers reproducible Elo scores with confidence intervals. The paper also reports strong inter-annotator agreement and transparent filtering procedures.
3. **Human-centric insight and analytical depth**: The analysis section is insightful, and I highly appreciate it. It shows that users value execution process quality —such as clarity, responsiveness, and error recovery —more than speed or execution time. The observed “inverted U-shaped” relationship between the number of CALL_USER queries and win rate offers actionable guidance for designing interactive agents. Moreover, the comparison between tool-integrated and pure GUI agents reveals an important benchmark gap. Tool-centric agents perform well on scripted benchmarks but often struggle with open-ended real-world tasks due to tool misuse and hidden failures. These findings represent valuable behavioral diagnostics for the field.
4. **Clarity and communication**: The paper is well-written and illustrated. The methodology is described with approachable language, and the paper also offers full reproducibility details. This clarity significantly enhances the paper’s readability and utility to the community.

**Weaknesses:**

1. **Limited task coverage and representativeness**: Although the authors collect over 2,000 votes, some models receive relatively few comparisons, which may affect ranking stability. Tasks are crowd-sourced and primarily English-language, so the distribution may underrepresent non-English and specialized domains such as scientific or enterprise workflows. Explicit coverage metrics for task diversity would strengthen the evaluation.
2. **Cost and scalability**: The framework is efficient for benchmarking but remains costly for large-scale data collection (about $1.24 per vote). Scaling to preference-tuning datasets might require further automation or active sampling methods.
3. **Human-centric definition and interactivity**: While CAA is human-judged, the evaluation is not truly interactive because raters review replays rather than engage in real-time turn-taking. This limits its scope as a “human-in-the-loop” system. Future extensions could integrate live interaction to capture dynamic feedback and adaptation.
4. **Task bias and environmental variability**: Despite identical AMIs and matched virtual machines, minor runtime differences such as network latency or software updates could influence agent behavior. Moreover, crowd-generated tasks may skew toward common desktop operations rather than professional applications.

**Questions:**

1. **Task coverage and diversity**: How do you quantify the coverage of crowd-sourced tasks across domains and difficulty?
2. **Influence of user expertise**: Did the authors analyze whether technical versus non-technical raters show different preference patterns? This could inform CUA design for different audiences.
3. **Complementary metrics**: Can CAA report additional task-level signals such as time-to-completion, error counts, or privacy violations to assist agent diagnosis?
4. **Score interpretation**: Given the high correlation between Elo and GenScore (r ≈ 0.95 in Appendix E.2), what distinct insight does GenScore provide beyond the main leaderboard?

---

> ### Author Response · Authors · 2025-11-20
>
> We sincerely thank Reviewer hCqQ for the thoughtful and constructive review. We are grateful for your recognition of CAA's originality, technical quality, and analytical depth, particularly your appreciation of our human-centric insights and the analysis of tool-integrated vs. pure GUI agents. We also appreciate the insightful questions regarding task diversity quantification, user expertise influence, complementary metrics, and GenScore interpretation.
>
> Below, we address each point in turn and clarify our contributions and ongoing efforts.
>
> ---
>
>
> >**Q1: Task coverage and diversity: How do you quantify the coverage of crowd-sourced tasks across domains and difficulty?**
> >
>
> A: Thank you for this important question. We quantify task coverage through multiple complementary dimensions that together provide a comprehensive view of both breadth and depth.
>
> For domain coverage, we apply hierarchical topic modeling using the Cilo pipeline (Tamkin et al., 2024) to derive a structured taxonomy. As shown in Figure 10a, we identify 6 major categories and 15+ subcategories: Information Retrieval (42.2%), Content Creation (31.1%), Technical & Domain-Specific (11.7%), UI Navigation (6.9%), and OS Operations (8.2%). This hierarchical clustering provides interpretable labels for category-specific evaluation. Beyond categorical analysis, we also embed all task instructions using text-embedding-3-small and project them via PCA (Figure 4a). CAA tasks occupy a broader and less clustered semantic space compared to OSWorld, WebArena, and WebVoyager, confirming that our task distribution is not concentrated in narrow domains.
>
> Our OSWorld ablation study (Table 2) provides another lens on coverage: only 20.7% of CAA tasks are similar to OSWorld (in-domain), while 79.3% represent diverse, out-of-domain scenarios. This validates that CAA captures a broader task landscape through crowdsourced inputs—a more reliable reflection of real-world agent robustness compared to curated benchmarks.
>
> For difficulty quantification, we employ four complementary metrics. First, task length averages ~17 words, shorter than benchmarks but reflecting real user habits of concise queries. Second, we use GPT-4o-mini binary classification to identify open-ended tasks based on criteria like multiple valid solutions, subjective judgment, and personal interpretation, and CAA shows significantly higher proportions of such tasks. Third, unigram perplexity is nearly double that of static benchmarks, indicating higher ambiguity and underspecification. Fourth, execution complexity manifests in both trajectory length and per-category Elo variance (Figure 10b): open-ended categories like Information Retrieval and Content Creation show 150-point Elo spreads, while deterministic categories like UI Navigation and OS Operations show only 50-point spreads.
>
> ---

---

> > ### Author Response · Authors · 2025-11-20
> >
> > > **Q2: Influence of User Expertise - Did you analyze whether technical vs. non-technical raters show different preference patterns?**
> > >
> >
> > Thank you for this insightful question about user expertise stratification. We recognize this as a valuable research direction for designing audience-specific CUAs.
> >
> > We intentionally did NOT stratify users by technical expertise during data collection. Instead, we aimed to capture naturally emerging preferences from a representative user population. Our annotators span diverse demographics across 20+ countries, with 47% holding bachelor's degrees or higher and varied professions including 32% full-time workers and 26% students, but we prioritized organic diversity over predefined segmentation. The goal was to reflect how CUAs would be evaluated in real-world deployment, where users span the full spectrum from novices to experts.
> >
> > While we didn't perform explicit stratification analysis, our evaluation of tool-integrated agents provides compelling indirect evidence about technical vs. non-technical preference gaps. Consider CoAct-1 [1], a multi-agent framework combining GUI automation with code execution that represents a highly technical design philosophy. On the OSWorld benchmark, CoAct-1 achieves 60.1% success rate. Yet on CAA, it ranks 5th with Elo 1043 and 41.8% correctness. Its successful trajectories average only ~3 steps, suggesting tools solve narrow problems efficiently but struggle with open-ended tasks.
> >
> > Our analysis (Section 4.3) identifies two critical failure modes: tool-selection bias where the agent over-invokes coding tools on tasks better served by direct GUI actions like simple file management or web browsing, and error opacity where code execution failures are non-surfaced in GUI replays, undermining user trust.  CoAct-1's performance varies dramatically by task type, high correctness on technical categories like Coding and Development Environments, but extremely low on browser-based Information Retrieval tasks. This pattern suggests tool-integrated agents excel on scripted technical tasks but fail to satisfy diverse real-world users, particularly non-technical users who expect intuitive GUI interactions over code-based solutions.
> >
> > We plan to conduct post-hoc stratification analysis using self-reported technical proficiency from our demographic surveys and will design targeted experiments comparing technical vs. non-technical user preferences on identical agent-task pairs. We commit to including preliminary stratification findings in future work and view this as a key direction for developing audience-aware CUAs.
> >
> > **References:**
> >
> > [1] L. Song et al., "CoAct-1: Computer-using Agents with Coding as Actions," arXiv preprint arXiv:2508.03923, 2025. Available: https://arxiv.org/abs/2508.03923
> >
> > ---

---

> > > ### Author Response · Authors · 2025-11-20
> > >
> > > > **Q3: Complementary Metrics - Can CAA report additional task-level signals for agent diagnosis?**
> > > >
> > >
> > > Yes, absolutely. CAA collects a comprehensive suite of task-level metrics beyond pairwise preferences, capturing both human-labeled and automatically computed signals to enable multi-faceted agent diagnosis.
> > >
> > > We collect four primary human annotations with strong inter-annotator agreement measured via Krippendorff's α (Appendix D.4): Preferences (α = 0.72) for pairwise comparisons, Correctness (α = 0.78) for binary task completion judgments with per-agent rates in Table 2(a), Safety (α = 0.68) for privacy violations and inappropriate actions with step-wise evaluations, and Efficiency (α = 0.70) for user perception of responsiveness and execution quality. These moderate-to-strong agreement levels validate annotation consistency even for open-ended GUI tasks.
> > >
> > > For every agent execution, we automatically record trajectory-level metrics including time-to-completion (total execution time and per-step latency), step counts (total trajectory length), and interaction patterns like CALL_USER query frequency (Figure 5 reveals an inverted U-shaped relationship with win rate).
> > >
> > > We also extract agent behavior patterns from execution traces (Appendix F): repetition ratio measuring proportion of repeated/stuck actions which negatively correlates with user preference indicating planning failures, reflection frequency counting steps with self-corrective CoT reasoning (keywords: "think", "retry", "mistake") which positively associates with task success. Beyond trajectory-level metrics, we collect optional step-wise evaluations including per-step grounding errors, privacy violations, and self-correction behaviors.
> > >
> > > The main leaderboard focuses on Elo for simplicity, interpretability, and direct reflection of user preferences, while these rich metrics serve as complementary diagnostics for researchers and developers. We commit to releasing all collected metrics (preference votes, human annotations, trajectory data, computed metrics, step-wise labels, task metadata). We believe these multi-dimensional signals will be valuable for training reward models, developing diagnostic tools, understanding failure modes, and designing process-reward-based evaluation methods.
> > >
> > > ---
> > > > **Q4: GenScore Interpretation - Given r ≈ 0.95 correlation with Elo, what distinct insight does GenScore provide?**
> > > >
> > >
> > > Thank you for this insightful question. The high correlation (r = 0.949, p < 0.0001) is actually a key validation finding rather than redundancy, it confirms our central hypothesis that success on CAA's crowdsourced platform is tightly linked to cross-domain generalization ability.
> > >
> > > GenScore was motivated by our OSWorld ablation study (Table 2, Figure 3b), which revealed rankings shifting substantially between subsets. This highlights a critical vulnerability of static benchmarks: data contamination and distribution bias. Agents that memorize benchmark templates achieve inflated scores but fail to generalize. CAA's user-centric evaluation represents the broadest, most realistic task distribution, and GenScore quantifies whether an agent succeeds by being a consistent generalist across all categories or by specializing in certain niches while failing elsewhere.
> > >
> > > GenScore reveals two distinct insights that Elo alone cannot surface. First, all agents struggle dramatically on Content Creation tasks. Typical tasks like "Create a slide report about GenAI in 2025" expose that even when all agents perform poorly, pairwise Elo comparisons may still differentiate them, but absolute performance remains unacceptably low (<30% correctness for top agents). GenScore reveals this systemic capability gap in handling non-verifiable, open-ended, subjective tasks that uniform Elo ranking obscures.
> > >
> > > Second, GenScore captures specialist vs. generalist trade-offs. CoAct-1 achieves 60.1% on OSWorld (SOTA) but ranks 5th on CAA with severe category imbalance: high correctness on technical categories but extremely low on browser-based tasks due to tool-selection bias. GenScore reveals the applicability boundary of different design philosophies, showing that adding tools doesn't universally improve performance and may harm non-technical scenarios.
> > >
> > > The r = 0.949 correlation confirms that CAA's crowdsourced task distribution is sufficiently diverse that overall success requires cross-category robustness, that static benchmarks with narrow distributions would not show this correlation, and that GenScore validates the arena methodology where success requires genuine broad competence rather than gaming specific templates.
> > >
> > > ---

---

> > > > ### Author Response · Authors · 2025-11-20
> > > >
> > > > > **W1 & W2: Limited Task Coverage and Cost/Scalability**
> > > > >
> > > >
> > > > Regarding ranking stability with varying vote counts, while some models received fewer comparisons initially, we conducted rigorous statistical validation (Appendix D) confirming robustness. Bootstrap confidence intervals from 1,000 resamples show non-overlapping CIs for top models. Power analysis with n=1,661 votes provides ≈0.90 power for detecting medium effects. Permutation tests yield p=2×10⁻⁴ with Cohen's d > 0.5, and noise sensitivity analysis shows Kendall's τ = 0.87 even under 10% label perturbation. These analyses confirm our reported rankings are statistically robust despite varying vote counts.
> > > >
> > > > Regarding language and domain coverage, we acknowledge that our current dataset is primarily English-language and reflects general computer-use patterns rather than specialized scientific or enterprise workflows. We have achieved diverse demographics across 20+ countries with 1,058 users spanning 6 major task categories and 600+ environment setups, but specialized domains and non-English languages remain underrepresented. We commit to multilingual expansion (target 30%+ non-English tasks within 6 months), specialized domain coverage including scientific workflows like Jupyter and MATLAB and enterprise scenarios like CRM and financial dashboards with expert annotation, continuous model additions, and explicit coverage statistics reported in future versions.
> > > >
> > > > Regarding cost at `$1.24`/vote (compute `$0.02`, API `$0.72`, annotation `$0.50`), this includes complete end-to-end evaluation infrastructure, not just labeling. For comparison, traditional user studies cost `$15-30`/hour and expert-designed benchmark tasks cost \$50-100 each. For research establishing foundational methodology, this cost is justified. For scalability, we are training a VLM-based reward model on our 2,201 human-labeled comparisons targeting correlation r > 0.85 with human judgments and achieving `~25×` cost reduction (`~$0.05`/vote). We will implement active sampling to reduce the needed votes and adopt a hybrid evaluation using reward models for rapid initial assessment with human validation on uncertain cases. We commit to releasing the full dataset (2,201+ votes, trajectory data, multi-dimensional annotations) and infrastructure code (platform codebase, VM setup, agent integration templates) under CC-BY 4.0 license.
> > > >
> > > > ---
> > > >
> > > > >**W3 & W4: Real-Time Interactivity and Robustness Against Environmental/Task Bias**
> > > > >
> > > >
> > > > We thank the reviewer and clarify two related concerns—interactivity and environmental/task bias.
> > > >
> > > > CAA is not a replay-based or post-hoc evaluation system. All evaluations run on live VMs launched per session, streamed via secure real-time VNC in the browser (Appendix B.1). Users see agent execution as it happens and retain full desktop access before and after the run to configure environments, inspect outputs, and verify safety. During execution, agents can invoke CALL_USER (Section 2.2), enabling genuine human-in-the-loop clarification. Our use of .gif summaries may have caused confusion, these are visualizations of live execution, not replays. We will revise terminology accordingly and surface CALL_USER earlier in the paper.
> > > >
> > > > Perfect determinism is impossible in cloud settings, but we implement matched AMI snapshots, simultaneous dual-agent execution, deterministic initialization, and monitoring of network/system-load metrics. Statistical validation (Appendix D) shows that ranking stability remains high under 10–30% perturbations (Kendall’s τ=0.87 at 10% noise), with significant separation between models (permutation p=2×10⁻⁴). Minor runtime variability is not only controlled but also desirable for ecological validity, reflecting the real-world deployment conditions CUAs must handle.
> > > >
> > > > We agree that crowdsourced tasks skew toward high-frequency productivity apps (e.g., Chrome, LibreOffice). This reflects real user priorities and is valuable for evaluating everyday robustness, but it underrepresents specialized domains. We already acknowledge this (Appendix E.1) and provide an expansion plan: integrating scientific/enterprise workflows, adding category-specific leaderboards, and introducing an expert annotation pool. The current distribution remains more diverse than prior benchmarks and provides a scalable foundation for broader coverage.
> > > >
> > > > ---
> > > >
> > > > We thank Reviewer hCqQ again for the constructive feedback. We will incorporate these clarifications and improvements in the revision and remain committed to iterative improvement with community involvement.

---

> > > > > ### Comment · Reviewer_hCqQ · 2025-11-27
> > > > > **Response to Authors’ Rebuttal**
> > > > >
> > > > > Thanks for the detailed reply. All my concerns are addressed now, and I’d like to keep my current rating.

---

### Official Review · Reviewer_xYzD · 2025-11-01

**Soundness:** 3
**Presentation:** 3
**Contribution:** 2
**Rating:** 4
**Confidence:** 4

**Summary:**

The authors present a new benchmarking platform for agentic systems that interact with computers (virtual machines). In doing so, they find that existing models perform differently from what would be expected from previously reported results.

**Strengths:**

- With the increasing amount of models gaming evaluations, it's always good to have more benchmarks.
- I like the direction of moving beyond easily codified evaluations and relying on more direct preferences. This has been the trend since InstructGPT, but many works regress on this.

**Weaknesses:**

- This feels a little weak on contributions. I would have expected some kind of algorithmic insights to go along with a benchmark for a conference paper. I would have expected to see something like this in a dedicated benchmark track.
- It seems like the main advantage of the benchmark is that it has more open-ended tasks. But are the other benchmarks with fewer open-ended tasks already more or less solved? Why do we need more open-ended tasks?
- I would want to see some more discussion relating to the shortcomings of other benchmarks here. I'm struggling a bit to understand why existing benchmarks are meaningfully inadequate here. This is really the only thing holding me back from accepting it.
- Adding humans in here makes the evaluations markedly more stochastic. It isn't, after all, the only way of doing more holistic evaluation (see, e.g., LLM-as-a-Judge or Agent-as-a-Judge). Some discussion of this or alternative ways of automating the human judges in the future could be useful. Maybe the humans are not needed at all if correctness is guiding human evaluations?
- Why are the mouse movements defined as actual movements and not clicks at certain points? This is more related to my own curiosity here.

**Questions:**

See Weaknesses.

---

> ### Author Response · Authors · 2025-11-20
>
> We sincerely thank Reviewer xYzD for the thoughtful and constructive review. We appreciate your recognition that additional benchmarks are valuable for addressing evaluation gaming, and that our work advances the important direction of moving beyond easily codified evaluations toward direct human preferences. We are encouraged by your openness and understand your primary concern: demonstrating why existing benchmarks are meaningfully inadequate. Below we address each point and clarify our contributions.
>
> ---
>
> > **W1: This feels a little weak on contributions. I would have expected some kind of algorithmic insights to go along with a benchmark for a conference paper.**
> >
>
> We appreciate this observation and respectfully argue that our work provides substantial contributions in systematic insights and algorithmic guidance, beyond a traditional benchmarking platform. Our core contributions include:
>
> **Systematic Error Discovery Pipeline (Section 4.4).** We identify four critical failure modes that static benchmarks systematically fail to expose. First, tool-integrated agents like CoAct-1 excel on scripted benchmarks but underperform on real-user tasks due to tool-selection bias (over-invoking coding tools for GUI-appropriate tasks) and error amplification from opaque tool failures. Their mean successful trajectory length is only 3 steps, indicating narrow applicability. Second, even state-of-the-art models like Claude 4 Sonnet exhibit long-horizon memory failures, losing track of context in multi-step workflows such as converting multiple files or compiling statistics. Third, real queries are underspecified (mean ≈17 words, Figure 4b), yet pure GUI agents issue speculative commands rather than seeking clarification—while models with CALL_USER show an inverted-U relationship where 1-2 clarification queries optimize user preference (Figure 5). Fourth, execution degrades due to fine-grained action failures including mishandled scrolling, clicks on non-interactive elements, or incorrect text editing—errors invisible to outcome-only metrics.
>
> These findings yield actionable algorithmic insights: future CUAs should develop adaptive tool-selection policies, strengthen long-term memory mechanisms, incorporate proactive clarification strategies, and improve fine-grained grounding through specialized training.
>
> **Human Preference Characterization (Section 4.2).** Our analysis reveals that users prioritize process quality over outcomes alone. Agents earn preference through partial progress, error recovery, and responsiveness even when tasks remain incomplete (Example 2, Appendix J). Correctness correlates strongly with win rate (ρ = 0.89, Figure 11a), while efficiency metrics (step count, latency) show negligible impact when correctness is controlled (Figure 11b). This contrasts with static benchmarks that evaluate only final states.
>
> **Generalization Diagnostics.** Our GenScore metric (Appendix E.2) demonstrates that success on CAA requires balanced performance across diverse task categories, not merely high scores on narrow distributions. Models like UI-TARS-1.5 perform well on in-domain tasks but exhibit significant degradation on diverse categories (Figure 10b).
>
> We hope to establish a new evaluation paradigm while providing systematic behavioral diagnostics and we believe CAA contributes in this spirit.

---

> > ### Author Response · Authors · 2025-11-20
> >
> > > **W2: It seems like the main advantage of the benchmark is that it has more open-ended tasks. But are the other benchmarks with fewer open-ended tasks already more or less solved? Why do we need more open-ended tasks?**
> > >
> >
> > We appreciate this important question, which addresses a central motivation for our work. You correctly observe that CAA features a significantly higher proportion of open-ended tasks (79.3% out-of-domain vs 20.7% in-domain, Table 2). We emphasize that this distribution is not artificially constructed to increase difficulty—rather, it naturally emerges from how real users interact with computer agents when given autonomy to define their own tasks. As shown in Figure 4b, real user instructions are considerably shorter (mean ≈17 words) yet substantially more ambiguous (nearly double the perplexity of static benchmarks), characterized by implicit intent and underspecified goals. This distribution is central to our user-centric evaluation philosophy and represents a fundamental departure from template-driven static benchmarks.
> >
> > **Regarding benchmark saturation.** Recent results indicate that frontier models are approaching or exceeding human-level performance on established benchmarks. OSWorld-Verified [1] reports end-to-end models achieving 65.4% pass@1 accuracy, while framework agents reach 69.9% (Agent S3). The OSWorld documentation establishes human-level performance at approximately 72% (measured with 50% PhD-level experts and 50% undergraduate-level annotators)[2], indicating that current state-of-the-art models operate at near-human levels on this benchmark. Performance on other benchmarks appears even stronger: 93.7% on WebVoyager, 69.9% on WebArena, and 83.7% on AndroidWorld—in many cases surpassing reported human baselines[3].
> >
> > **Critical deployment gaps.** However, our empirical analysis reveals substantial disconnects between benchmark performance and real-world deployment viability. First, examination of actual OSWorld trajectories from high-performing models uncovers concerning behaviors. For instance, AGI-0 model trajectories on certain homework-completion tasks show the agent directly populating answer sheets with standard responses without accessing the homework document[4]. While such shortcuts may yield high accuracy on templated benchmark tasks through pattern matching, they fail when encountering varied real-world scenarios.
> >
> > Second, computational practicality constraints emerge that benchmarks do not penalize. Framework agents like Agent S3 employ best-of-N (BoN) sampling methods that, while effective for maximizing benchmark scores, require in excess of 30 minutes to complete a single moderate-difficulty OSWorld task[5]. Such latency renders these approaches impractical for real-time user interaction, yet static benchmarks reward these methods without accounting for computational cost or response time.
> >
> > Third, Figure 3a documents substantial ranking reversals when transitioning from static benchmarks to user-centric evaluation. UI-TARS-1.5 ranks #2 on OSWorld but falls to #4 on CAA. Operator achieves #1 rankings on both WebVoyager and Online-Mind2Web but drops to #3 on CAA. These shifts indicate fundamental differences in what is evaluated and what matters for deployment.
> >
> > These findings motivated CAA's design as a realistic, user-centric complement to static benchmarks. Our objective is not to replace controlled evaluation—which remains valuable for measuring specific capabilities—but to address a critical gap: as models approach saturation on existing benchmarks, evaluation methodologies must reflect actual usage patterns and expose failure modes relevant to deployment. CAA captures authentic user query distributions (concise, underspecified, open-ended), reveals systematic failures obscured by templated evaluation (tool misuse, memory lapses, action precision errors in Section 4.4), assesses process quality and user satisfaction beyond outcome correctness (Section 4.2), and provides behavioral diagnostics for improving real-world deployment.
> >
> > **References:**
> >
> > [1] XLang Team, "OSWorld-verified: A New Milestone in Open-Source Language Model Evaluation," 2024. Available: https://xlang.ai/blog/osworld-verified
> >
> > [2] OSWorld Project, "OSWorld: A Benchmark for Multimodal Agents in Real Computer Environments," 2024. Available: https://os-world.github.io/
> >
> > [3] H Company, "Surfer-2: Scaling Autonomous Web Navigation with Reinforcement Learning," 2024. Available: https://www.hcompany.ai/blog/surfer-2
> >
> > [4] XLang Team, "Ubuntu OSWorld-Verified Trajectories: AGI0 50-Step Results (2024-11-17)," Hugging Face Dataset, 2024. Available: https://huggingface.co/datasets/xlangai/ubuntu_osworld_verified_trajs/blob/main/results_agi0_50steps_1117.zip
> >
> > [5] Simular AI, "Agent-S3: A Scalable System for Autonomous Desktop and Web Task Automation," 2024. Available: https://www.simular.ai/articles/agent-s3
> >
> > ---

---

> > > ### Author Response · Authors · 2025-11-20
> > >
> > > > **W3: I would want to see some more discussion relating to the shortcomings of other benchmarks here. I'm struggling a bit to understand why existing benchmarks are meaningfully inadequate here. This is really the only thing holding me back from accepting it.**
> > > >
> > >
> > > We appreciate you highlighting this as a central concern and agree it warrants deeper discussion. We will expand Section 5 in our revision to address this systematically. Current benchmarks follow two primary paradigms, each with fundamental limitations:
> > >
> > > | Dataset | Scale (Tasks) | Open-ended tasks | Human prefs | Avg. trajectory length | Domains | OS |
> > > | --- | --- | --- | --- | --- | --- | --- |
> > > | CAA (Ours) | 2,201 | ✓ | 921 | 18 | >100 | Windows, Ubuntu, Web |
> > > | OSWorld | 369 | ✗ | 9 | 15 | 10 | Ubuntu |
> > > | Windows Agent Arena | 154 | N/A | N/A | 5–10 | 7 | Windows |
> > > | WebArena | ~450 | ✓ | 182 | ~7.5 | 5 | Web |
> > > | WebVoyager | 643 | N/A | N/A | 15 | Web | — |
> > > | MiniWoB++ | 104 | N/A | N/A | N/A | N/A | Web |
> > >
> > > **Reward-scripted benchmarks** (OSWorld, Windows Agent Arena, WebArena) face three critical bottlenecks. First, cost and diversity constraints. OSWorld-Verified and SWE-bench-Verified document that unlike standard LLM benchmarks, environment-based evaluation requires dozens of expert annotators contributing 900+ hours with extensive quality control and cross-validation procedures[1][2]. This renders large-scale, diverse evaluation prohibitively expensive and makes user-centric evaluation at scale impractical.
> > >
> > > Second, reliability challenges from dynamic content. OSWorld-Verified explicitly notes that reward-scripted tasks exhibit high sensitivity to software updates, website modifications, and anti-automation measures. These dynamic factors compromise the reliability of static evaluation over time[1].
> > >
> > > Third, representativeness limitations. Licensing restrictions and infrastructure constraints necessitate reliance on open-source software, precluding evaluation with proprietary tools that users commonly employ. As documented in your provided comparison table, OSWorld covers 10 domains on Ubuntu, WebArena encompasses 5 web-based domains, while CAA spans >100 domains across Windows, Ubuntu, and web environments. This limited coverage fails to represent authentic personal or professional computer-use scenarios.
> > >
> > > **LLM-as-a-judge benchmarks** (Online-Mind2Web, Agent-as-a-Judge frameworks) address scalability but introduce distinct challenges. First, preference misalignment. LLM evaluators do not reliably capture human preferences regarding alignment, trustworthiness, and process quality—precisely the signals users prioritize in deployment. To verify this empirically, we implemented a baseline evaluation using o3 on OSWorld rollout trajectories. We retained the final three screenshots and reasoning traces, uniformly sampled three additional screenshots and consecutive reasoning steps from earlier in the trajectory, and prompted o3 to generate a binary correctness judgment (1 or 0). Comparison against ground-truth reward scripts yielded an F1 score of only 76%, indicating that current advanced VLMs cannot yet serve as sufficiently reliable judges and may not accurately represent human intent and evaluation criteria.
> > >
> > > Second, the environment coverage challenge persists. Even with automated evaluation, researchers cannot readily scale realistic computer scenarios that reflect diverse user configurations, continuing to limit task diversity.
> > >
> > > We designed CAA specifically to address these barriers through: (1) Cloud-hosted VM infrastructure providing 600+ diverse initializations with support for user customization, eliminating expert annotation bottlenecks. (2) Direct human evaluation capturing preferences on process quality, not solely outcomes. (3) Crowd-sourced tasks from 1,058 users, yielding organically diverse and representative task distributions. (4) Dynamic environments with periodic refreshes, preventing overfitting to static configurations. (5) Behavioral diagnostics via error analysis and preference signals revealing what users genuinely value.
> > >
> > > We emphasize that CAA complements rather than replaces static benchmarks—controlled evaluation retains value. However, as models approach saturation on existing benchmarks, the community requires evaluation methodologies that reflect actual deployment scenarios. Our work addresses this need.
> > >
> > > **References:**
> > >
> > > [1] XLang Team, "OSWorld-verified: A New Milestone in Open-Source Language Model Evaluation," 2024. [Online]. Available: https://xlang.ai/blog/osworld-verified
> > >
> > > [2] OpenAI, "Introducing SWE-bench Verified: A Rigorously Validated Benchmark for Software Engineering Agents," 2024. [Online]. Available: https://openai.com/index/introducing-swe-bench-verified/
> > >
> > > ---

---

> > > > ### Author Response · Authors · 2025-11-20
> > > >
> > > > > **W4: Adding humans in here makes the evaluations markedly more stochastic. It isn't, after all, the only way of doing more holistic evaluation (see, e.g., LLM-as-a-Judge or Agent-as-a-Judge). Some discussion of this or alternative ways of automating the human judges in the future could be useful. Maybe the humans are not needed at all if correctness is guiding human evaluations?**
> > > > >
> > > >
> > > > We appreciate this constructive suggestion. Given the cost and operational complexity of collecting user votes, we acknowledge that exploring automation for future scalability is important. We plan to develop a human-preference-based reward model trained on our collected votes to accelerate evaluation of new models, and will release all datasets, model checkpoints, and platform infrastructure to support this direction.
> > > >
> > > > However, we wish to emphasize two critical considerations. First, to assess whether current advanced VLMs can reliably function as judges for GUI agents, we implemented and evaluated a baseline using o3 on OSWorld rollout trajectories. We retained the final three screenshots and reasoning traces from each trajectory, uniformly sampled three additional screenshots and consecutive reasoning steps from earlier execution, and prompted o3 to generate a binary correctness judgment (1 or 0). Comparison against ground-truth reward scripts yielded an F1 score of 76%. This result suggests that current models cannot yet serve as sufficiently reliable judges and may not accurately capture human intent and evaluation criteria. This empirical evidence motivates our continued reliance on human evaluation.
> > > >
> > > > Second, human-centric evaluation constitutes the core philosophical foundation of our work, not merely a practical expedient. While correctness labels might be amenable to automation through reward models or scripts, preference judgments, particularly step-wise preferences and process quality assessments require human signals to authentically reflect user experience with these agents. The fundamental distinction is this: LLM-as-a-Judge optimizes for what language models consider optimal, whereas our objective is optimizing for genuine human needs and preferences. Section 4.2 and Example 2 in Appendix J demonstrate that users value process quality including partial progress, error recovery, and thoughtful clarification even when tasks remain incomplete. These nuanced judgments prove difficult for current LLMs to capture from trajectory data alone. Maintaining human evaluators as the authoritative standard is essential for evaluating models against authentic human distributions and for advancing research directions such as CUA RLHF.
> > > >
> > > > ---
> > > >
> > > > > **W5: Why are the mouse movements defined as actual movements and not clicks at certain points?**
> > > > >
> > > >
> > > > This design decision stems from how `pyautogui` and `xdotool` implement GUI interactions at the operating system level. Both libraries expose mouse_move and click as distinct primitives because they simulate OS-level events through X11 (Linux) or Win32 (Windows) APIs. While pyautogui.click(x, y) internally repositions the cursor prior to clicking, many GUI elements require hover states to trigger tooltips, dropdown menus, or dynamic content updates. Separating movement from clicks enables agents to observe state changes following cursor repositioning but preceding the click event—matching authentic GUI interaction patterns.
> > > >
> > > > From an agent design perspective, this separation provides three benefits: First, agents can hover to reveal UI elements before committing to a click action. Second, it facilitates error recovery—agents can reposition the cursor without triggering unintended clicks. Third, it maintains compatibility with how most CUA implementations structure their action spaces (e.g., Claude Computer Use API emits separate MOUSE_MOVE and LEFT_CLICK actions).
> > > >
> > > > For evaluation purposes, this distinction has minimal impact on task outcomes, as final click coordinates determine success. The primary benefit is enhanced interpretability: users reviewing trajectories can better comprehend the agent's navigation logic when observing actual movement paths rather than instantaneous cursor teleportation.
> > > >
> > > > ---
> > > >
> > > > We hope these responses address your concerns, particularly regarding why existing benchmarks prove inadequate and why our work contributes substantially beyond traditional benchmarking. We are prepared to incorporate your feedback into our revision and further elaborate on these points. We thank you again for your thorough and constructive review.

---

### Official Review · Reviewer_GEBQ · 2025-11-03

**Soundness:** 4
**Presentation:** 4
**Contribution:** 4
**Rating:** 8
**Confidence:** 3

**Summary:**

The paper presents a platform for evaluating computer use agents (CUAs) via head-to-head evaluation strategies with real users. The platform provides preset initialized computer systems with apps in various start states. The users write task prompts against this environment and get to observe and evaluate the agent performance on the task. These preferences from more than 1000 users on about 1200 tasks are used to provide ELO style rankings of a large number of models. The results surface significant differences in human preference compared to previously observed rankings on existing CUAs evaluation environments (e.g OSWorld). Further analyses shows specific difficulties in these dynamic real user evaluations compared to static fixed set evaluations, as well as some insights in to the type of agent behavior that humans prefer.

**Strengths:**

This is a timely and much needed effort in evaluating computer user agents. The head-to-head evaluation using real users at scale provides strong empirical evidence and fine-grained information that can help distinguish between different model performance, and highlight where the models fail. The overall methodology is sound and has been done at scale. The framework looks extensible---can be easily used for other specific types of evaluation considerations.

**Weaknesses:**

I am not able to identify any significant weaknesses for this work.

- The evaluation shows that the tasks being generated are diverse. It would help to either describe how this diversity comes about in the process. Figure 1 lays out the general process but some more details on the process could help clarify whether the scenarios within computers are somehow systematically guaranteed to be diverse or if the diversity comes from what the users do with the same scenarios.

- While I appreciate the apples-to-apples comparison with standardized settings, it would also help to have tried the best settings for at least one or two models to see what is the best performance that could have been obtained. This is not a major issue and one that future work can fix.

**Questions:**

See weaknesses above.

---

> ### Author Response · Authors · 2025-11-20
>
> We sincerely thank Reviewer GEBQ for the thoughtful recognition of our work. We are grateful for your positive assessment of our methodology's soundness, the scale of our evaluation, and the framework's extensibility. We also appreciate the constructive feedback regarding task diversity mechanisms and model configuration strategies.
>
> Below, we address each point in turn and clarify our design decisions.
>
> ---
>
> > **W1: It would help to either describe how this diversity comes about in the process... whether the scenarios within computers are somehow systematically guaranteed to be diverse or if the diversity comes from what the users do with the same scenarios.**
> >
>
> Thank you for this insightful question. The diversity in Computer Agent Arena stems from a **dual-source design** combining both systematic environmental diversity and organic user-driven task diversity:
>
> We systematically curate **600+ distinct initialization configurations** to ensure diverse starting states:
>
> - **89 popular websites + 472 subdomains** crawled from SimilarWeb's top-100 domains
> - **12 mainstream desktop applications** sourced from Microsoft Store and Snapcraft (e.g., LibreOffice, VLC, VS Code)
> - **97 heterogeneous files** covering common formats (.docx, .xlsx, .pptx, .pdf, .py, .md)
> - **Quick-start customization APIs** (`upload_file`, `open_websites`, `clone_repo`) enabling user-defined setups
>
> Importantly, we implement **anti-overfitting mechanisms**: file contents are refreshed monthly, and filenames/folder structures are randomized at each session launch (as described in Sec. B.2). Each evaluation randomly selects one configuration from this pool, ensuring agents face varied environmental contexts.
>
> Real users submit tasks based on their authentic needs, resulting in:
>
> - **6 major semantic categories** with hierarchical subcategories (Information Retrieval 42.2%, Content Creation 31.1%, etc.)
> - **79.3% out-of-domain tasks** relative to OSWorld (Table 2), demonstrating broader coverage
> - **Broader PCA semantic spread** than static benchmarks like OSWorld, WebArena, and WebVoyager (Figure 9a)
> - Tasks that are **shorter (mean ≈ 17 words) yet more ambiguous** (nearly 2× perplexity vs. benchmarks), reflecting real-world underspecified queries
>
> Thus, diversity arises from **both systematic environmental setup AND natural heterogeneity of real-user tasks**. The environment provides rich, varied contexts (600+ configurations), while users freely define objectives based on their actual needs—together creating the observed semantic and operational diversity that distinguishes Computer Agent Arena from scripted benchmarks.

---

> > ### Author Response · Authors · 2025-11-20
> >
> > > **W2: It would also help to have tried the best settings for at least one or two models to see what is the best performance that could have been obtained.**
> > >
> >
> > Thank you for this thoughtful suggestion. We want to respectfully clarify that we already employ model-specific optimal settings for certain agents, and this is a key design decision in Computer Agent Arena.
> >
> > As detailed in Section B.3, we implement agents through two complementary approaches:
> >
> > **1. Baseline Agent Scaffold (for VLM backbones):**
> > For general vision-language models without native computer-use implementations (e.g., GPT-4o, Gemini 2.5 Pro, Qwen2-VL 72B), we provide a standardized baseline wrapper with unified system prompts, tool declarations, and history management (n=5 by default). This scaffold enables **fair apple-to-apple comparison** across diverse VLM backbones and ensures reproducibility.
> >
> > **2. Official Agent Implementations (for agentic models):**
> > For models with built-in agent scaffolds and production-ready implementations (e.g., **Claude Sonnet 4, UI-TARS-1.5, Operator**), we **align directly to their official implementations** via user-ready API access. These models use their own optimized system prompts, tool schemas, reasoning strategies, and memory management—representing their **best-case performance** as deployed in real-world production.
> >
> > Models with official agent implementations consistently outperform baseline scaffolds, demonstrating the value of GUI-specific training and optimized prompting strategies:
> >
> > - **Claude Sonnet 4** and **Claude 3.7 Sonnet** achieve top-tier Elo rankings with remarkable planning, error-recovery, and execution capabilities
> > - In several complex tasks (e.g., open-ended information retrieval, document editing), these agents demonstrate performance approaching or even surpassing human execution, as observed in user feedback
> > - These advantages stem from extensive GUI-specific fine-tuning and carefully engineered scaffolds that our baseline cannot replicate
> >
> > This two-tier design ensures:
> >
> > - **Extensibility:** Any new VLM can be quickly evaluated via baseline scaffold
> > - **Optimality:** Production-grade agents are tested at their best configurations
> > - **Fairness:** Models are compared under their respective optimal conditions (not artificially handicapped)
> > - **Real-world alignment:** Evaluation reflects how users actually access these models in practice
> >
> > Thus, we have indeed explored "best settings" for top-performing models through their official implementations. The leaderboard reflects both optimized agentic models and general VLMs adapted via our baseline, providing a comprehensive view of the current landscape where some models are specifically designed and tuned for computer-use tasks.
> >
> > ---
> >
> > We thank Reviewer GEBQ again for the valuable feedback and strong endorsement of our work. We believe these clarifications address your concerns and will improve the paper's clarity in the final version.

---

### Author Response · Authors · 2025-12-03
**General Comment to All Reviewers and the Area Chair**

We sincerely thank all reviewers and the area chair for their thoughtful, constructive feedback on our work. We are encouraged that reviewers agree on the novelty and importance of our contribution: establishing a human-centric evaluation platform for computer-use agents through large-scale, real-user preferences in realistic environments.

**Several reviewers recognized our key strengths:**

- **Novel and timely approach** (Reviewer `GEBQ`, Reviewer `hCqQ`): First platform enabling scalable human-centric evaluation of CUAs through crowdsourced tasks and pairwise comparisons
- **Insightful analysis** (Reviewer `GEBQ`, Reviewer `xYzD`): Systematic error discovery revealing failure modes invisible to static benchmarks and human preference characterization showing users value process quality beyond outcomes
- **Practical impact** (Reviewer `GEBQ`, Reviewer `hCqQ`): Comprehensive open-source release of platform, dataset, and agent implementations

At the same time, reviewers raised several concerns, which we appreciate and believe we have fully addressed:

1. **Why existing benchmarks are inadequate** (Reviewer `xYzD`): We provided detailed analysis of fundamental limitations in reward-scripted benchmarks (cost/diversity/reliability) and LLM-as-a-judge approaches (76% F1 in our validation), plus evidence of benchmark saturation vs. deployment gaps
2. **Algorithmic contributions beyond benchmarking** (Reviewer `xYzD`): Systematic error taxonomy, human preference characterization, and generalization diagnostics providing actionable insights for CUA development
3. **Task coverage quantification** (Reviewer `hCqQ`): Multi-dimensional metrics including hierarchical categorization, PCA semantic analysis, and 79.3% out-of-domain task distribution
4. **Data volume and statistical power** (Reviewer `rdUN`): Comprehensive validation confirming sufficient power (≥0.90) with commitment to reward model training and active sampling for scalability
5. **Real-time interaction capabilities** (Reviewer `hCqQ`): Clarified live VNC execution with CALL_USER support, contrasting with replay-based evaluation

We believe these responses address reviewers' core concerns while preserving our practical contributions. We welcome further questions during the discussion period and thank all reviewers for their constructive feedback, which has helped refine our claims, clarify our positioning, and improve the overall presentation of Computer Agent Arena.

---

### Meta-Review · Area_Chair_eFKr · 2026-01-11

**Summary:**

This paper introduces Computer Agent Arena, a human-centric evaluation platform for computer-use agents that moves beyond static, contamination-prone benchmarks. Reviewers broadly agree that the work is timely, well-engineered, and provides valuable empirical insights through large-scale human preference data collected in realistic environments. The analysis reveals meaningful ranking reversals compared to existing benchmarks and uncovers behavioral failure modes that are otherwise difficult to observe. Overall, the paper makes a strong contribution to the evaluation of computer-use agents.

**Reviewer Concerns:**

Most reviewer concerns were satisfactorily addressed in the rebuttal. Reviewers appreciated the authors’ detailed clarifications regarding task diversity, statistical robustness, the limitations of existing benchmarks, and the value of human preference signals relative to automated judging.

One significant remaining concern from the AC perspective is the verifiability of the claimed openness and ongoing availability of the arena. The paper emphasizes that Computer Agent Arena is “open” and “continuously evolving,” yet the submission does not provide a clear, stable pointer (e.g., a public repository or maintained landing page) where the community can access and validate the platform, dataset, and code. While the rebuttal states an intention/commitment to open-source these components, this is not yet concretely reflected in an accessible artifact.

*AC requirement*: acceptance should be treated as conditional on the authors (by camera-ready) (1) releasing the platform + dataset + code publicly, and (2) providing documentation for reproducibility (setup, evaluation protocol, agent integration templates, and licensing). This would align the final paper with its stated claims and make the contribution durable.

**Reviewer Scores:**

- GEBQ: No change expected (already strong accept; minor suggestions only).
- hCqQ: No change; explicitly stated all concerns were addressed and kept rating.
- xYzD: Likely +1 (from marginal reject to borderline accept), since rebuttal directly addressed the main blocker: “meaningful inadequacy” of existing benchmarks and clarified the paper’s non-trivial analytical contributions.
- rdUN: Likely no change (concerns are primarily about long-term scaling/participation; rebuttal helps but doesn’t fundamentally flip that prior).

---

### Decision · Program_Chairs · 2026-01-26

Accept (Poster)